# Climate effects on archaic human habitats and species successions

Axel Timmermann[1,2 ✉], Kyung-Sook Yun[1,2], Pasquale Raia[3], Jiaoyang Ruan[1,2], Alessandro Mondanaro[4], Elke Zeller[1,2], Christoph Zollikofer[5], Marcia Ponce de León[6], Danielle Lemmon[1,2], Matteo Willeit[7] & Andrey Ganopolski[7]

It has long been believed that climate shifts during the last 2 million years had a pivotal role in the evolution of our genus *Homo*[1–3]. However, given the limited number of representative palaeo-climate datasets from regions of anthropological interest, it has remained challenging to quantify this linkage. Here, we use an unprecedented transient Pleistocene coupled general circulation model simulation in combination with an extensive compilation of fossil and archaeological records to study the spatiotemporal habitat suitability for five hominin species over the past 2 million years. We show that astronomically forced changes in temperature, rainfall and terrestrial net primary production had a major impact on the observed distributions of these species. During the Early Pleistocene, hominins settled primarily in environments with weak orbital-scale climate variability. This behaviour changed substantially after the mid-Pleistocene transition, when archaic humans became global wanderers who adapted to a wide range of spatial climatic gradients. Analysis of the simulated hominin habitat overlap from approximately 300–400 thousand years ago further suggests that antiphased climate disruptions in southern Africa and Eurasia contributed to the evolutionary transformation of *Homo heidelbergensis* populations into *Homo sapiens* and Neanderthals, respectively. Our robust numerical simulations of climate-induced habitat changes provide a framework to test hypotheses on our human origin.

During the past 5 million years (Ma), a gradual transition in climate conditions has occurred from the warmer and wetter Pliocene (5.3–2.6 Ma) to the colder and drier Pleistocene (2.6–0.011 Ma). During this time, tropical savannahs and open grasslands expanded in central–eastern Africa[4], which, according to the savannah hypothesis[5] and variants thereof[6], contributed to the early evolution of our human ancestors. Milankovitch cycles in solar insolation and climate (Extended Data Figs. 1–3), particularly the eccentricity-modulated precessional cycle (Extended Data Fig. 1a), further created multiple human migration corridors from sub-Saharan Africa into northern Africa, the Arabian Peninsula and Eurasia[7–10]. The existence of these corridors is well supported by fossil, archaeological[9] and genetic[11] evidence. A possible effect of astronomical forcings on early hominin evolution has been suggested in the context of the variability selection hypothesis[3,12,13], which posits that early hominin evolution, selection and speciation were influenced by alternating periods of high and low variability in climate and resources.

To better quantify the impact of spatially heterogenous orbital-scale climate variability[14] (Extended Data Fig. 4) on human evolutionary transitions, we conducted an unprecedented transient global coupled general circulation model (CGCM) simulation covering the global climate history of the last 2 Ma (henceforth referred to as the 2Ma simulation).

2Ma is based on the state-of-the-art Community Earth System Model version 1.2 (CESM1.2) in 3.75° × 3.75° horizontal resolution forced with ice-sheet distribution and elevation as well as $CO_2$ evolution obtained from another transient intermediate-complexity model simulation[15] and astronomical insolation changes[16] (Methods). 2Ma, which uses an orbital acceleration[17] factor of 5, reproduces key palaeo-climate records such as tropical sea surface temperatures, Antarctic temperatures, the eastern African hydroclimate and the East Asian summer monsoon in close agreement with palaeo-reconstructions (Extended Data Figs. 1 and 2), which supports the realism of our CGCM-based simulation. Glacial–interglacial variability is characterized by a global mean temperature amplitude of approximately 2–3 °C (5–6 °C) during the Early (Late) Pleistocene (Extended Data Fig. 2a), which is consistent with estimates from an Earth system model of intermediate complexity[15] and palaeo-climate data constraints[18–20].

To quantify the relationship between climate and the presence of hominin species, we built a climate envelope model (CEM; Methods and Extended Data Fig. 5). This CEM was derived from an extended version of a previously published species database (SDB)[21,22] composed of geochronologically constrained hominin fossils and archaeological layers containing lithic industries (Fig. 1a–e and Methods) and topographically downscaled (1° × 1° grid) 1,000-year averaged data of climate

[1]Center for Climate Physics, Institute for Basic Science, Busan, South Korea. [2]Pusan National University, Busan, South Korea. [3]DiSTAR, Università di Napoli Federico II, Monte Sant'Angelo, Naples, Italy. [4]DST, Università degli Studi di Firenze, Florence, Italy. [5]Anthropological Institute, University of Zurich, Zurich, Switzerland. [6]Department of Informatics, University of Zurich, Zurich, Switzerland. [7]Potsdam Institute for Climate Impact Research, Potsdam, Germany. ✉e-mail: timmermann@pusan.ac.kr

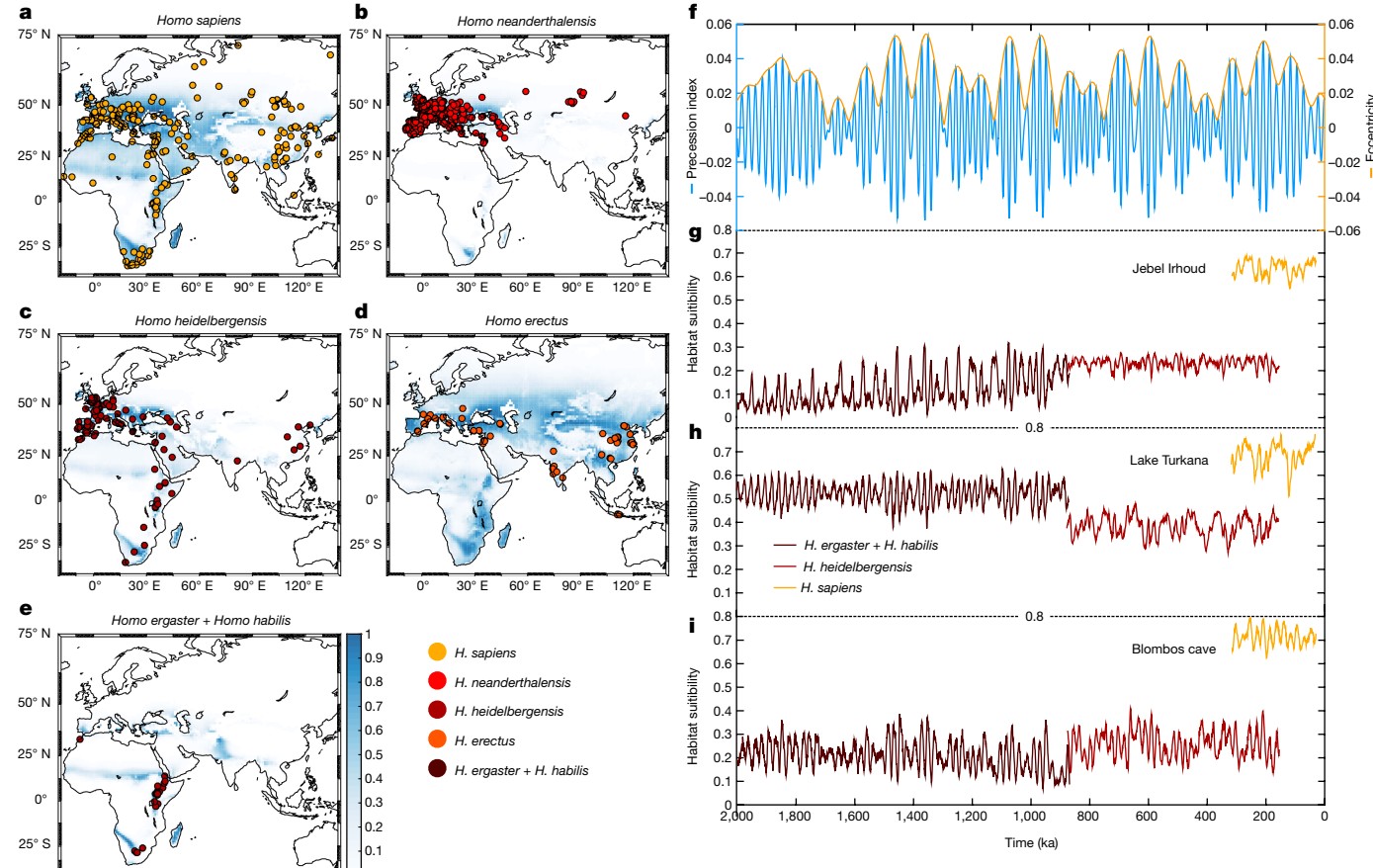

**Fig. 1 | Hominin species habitat suitability. a–e**, African–Eurasian species distribution calculated from a Mahalanobis distance model using four-dimensional climate envelope data of topographically downscaled temperature, precipitation and NPP changes simulated by 2Ma (Methods) and the locations and ages of fossil and archaeological sites (Supplementary Table 1). The time-averaged habitat suitability (blue to white shading) covering the period of respective hominin presence can be interpreted in terms of probability (Methods), with values ranging from 0 (habitat unsuitable) to

1 (habitat extremely suitable). Coloured circles represent the locations of fossils and/or archaeological artefacts associated with the five hominin groups. **f–i**, Time series for precession (blue) and eccentricity (**f**) and simulated regional habitat suitability at selected sites of archaeological interest for *H. habilis* and *H. ergaster* (treated jointly as early African *Homo*), *H. heidelbergensis* and *H. sapiens* (**g–i**). The centre locations of a 4° × 4° average area include Jebel Irhoud (34° N, 4° W), a region near Lake Turkana (0° N, 34° E) and Blombos cave (34° S, 21° E).

variables from 2Ma, which are relevant for human survival. These factors included annual mean precipitation, temperature, yearly minimum precipitation and net primary productivity (NPP; Methods). The 3,245 data entries of the extended SDB (Supplementary Table 1) contain information about location, age, age uncertainty and hypothesized species, selected among early African *Homo* (combining *Homo habilis* and *Homo ergaster* as one group), Eurasian *Homo erectus*, *Homo heidelbergensis*, *Homo neanderthalensis* and *Homo sapiens*. The spatiotemporal climate fields of the 2Ma simulation (see Extended Data Figs. 3 and 4 for select locations) were extracted for the species-presence locations and ages in the SDB and were then statistically aggregated as a CEM. Subsequently, using the Mahalanobis metric[23,24] and the spatiotemporal climate evolution in 2Ma, we derived a habitat suitability model (HSM; Methods) for each species, which quantifies the probability of finding fossil and/or archaeological evidence of the species at a given time and geographical location.

The key goals of our study were (1) to address how past climate changes have affected archaic human habitats; (2) to test whether the current fossil and archaeological records (location and age of each hominin species) have been affected by the orbital-scale evolution of our climate system; (3) to identify common climate envelopes and therefore potential contact zones of hominin groups; and (4) to identify linkages between regional climate shifts and evolutionary diversification.

## Time-averaged habitats

To illustrate the connection between climate and the temporal and geographical extent of hominin species, we focused on habitat suitability calculated from the CEM. The simulated time-averaged maps of hominin habitat suitability (Fig. 1a–e) exhibit several interesting features. In particular, the suitable habitat for early African *Homo* (Fig. 1e) is composed of relatively narrow corridors that begin in southern Africa and run northward throughout the rift valley, straddle the Intertropical Convergence Zone and cut across southern Africa in a northwest–southeast direction. Such a limited range and high spatiotemporal heterogeneity of habitat suitability are consistent with high levels of environmental specialization and sensitivity to regional environmental perturbations, such as eccentricity-modulated precessional cycles (Fig. 1f–i). Even though we included only Eurasian fossils and artefacts for *H. erectus* in the HSM, the predicted global habitat suitability of this species is far more extensive than that of any other hominin species analysed here (Fig. 1d). This is consistent with the concept that *H. erectus* was, on an evolutionary timescale, a flexible generalist who roamed Earth for more than 1 Ma and inhabited a wide range of different environmental conditions (Extended Data Fig. 6). Even though *H. erectus* and early African *Homo* fossil records are treated as geographically disjunct (Fig. 1d, e), there is still regional overlap in their climate

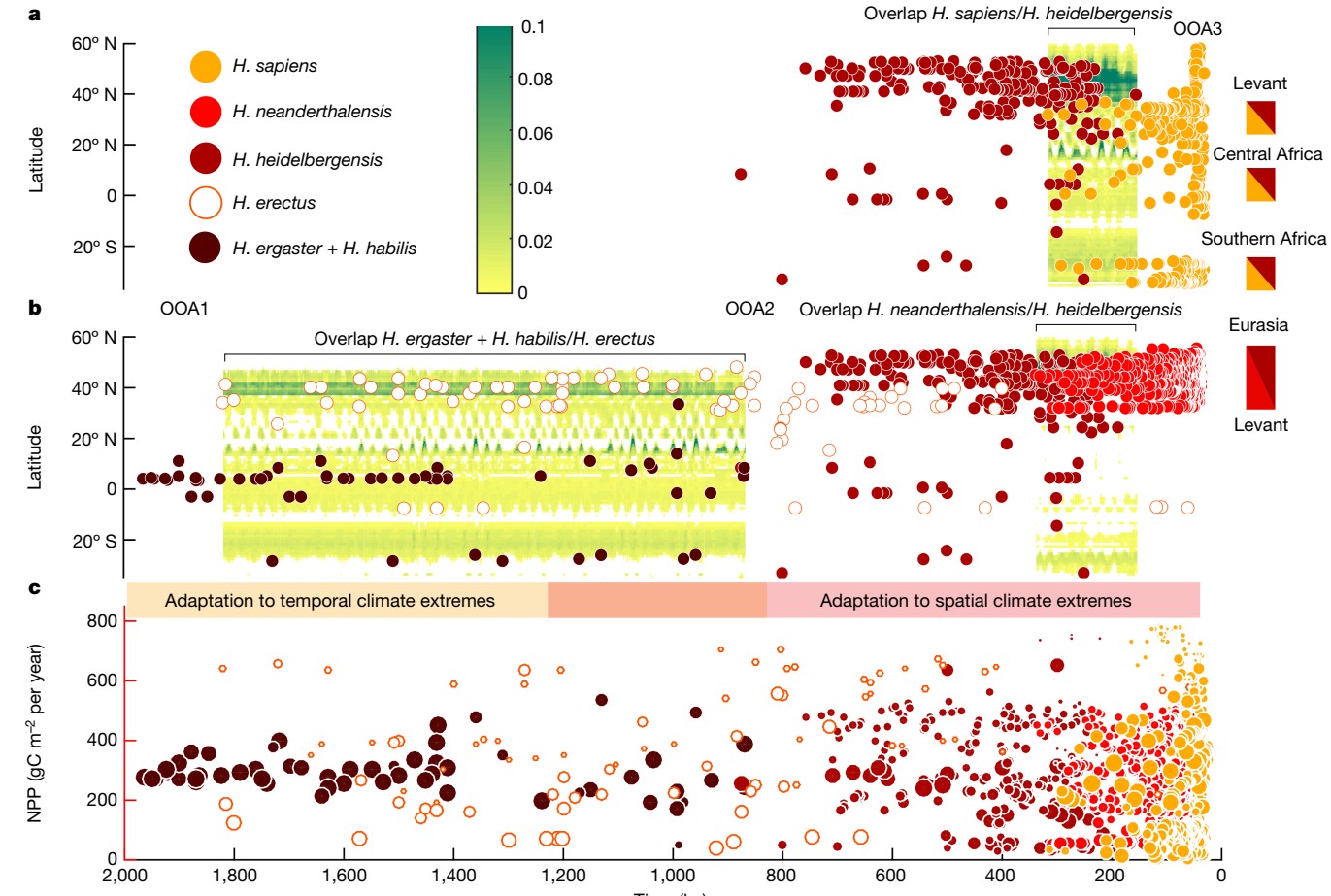

**Fig. 2 | Habitat overlap, succession and geographical distribution of fossils.**
**a**, Green shading represents a Hovmoeller (time–latitude) diagram of the zonal mean of the spatial scalar product of habitat suitability for *H. sapiens* and *H. heidelbergensis*. Circles represent the corresponding average age and latitudinal distributions of fossils and archaeological artefacts. High values of habitat overlap coinciding with joint presence of fossils indicate possible locations of hominin interaction, diversification and possibly speciation. **b**, Same as **a** but for *H. neanderthalensis* and *H. heidelbergensis* (right side) and *H. erectus* and *H. ergaster–habilis*. 'Out-of-Africa' migration periods are marked as OOA. Potential regions for gradual diversification and transformation are indicated by dual-coloured boxes. **c**, NPP (Methods) for each fossil and archaeological site (coloured circles), selected by averaging the 2Ma NPP data in a 6° × 6° vicinity around the fossil sites and for their respective fossil ages. The size of the circle represents the great-circle (Haversine) distance to a grid point in central–eastern Africa (4° N, 36° E), with larger circles indicating closer proximity to this location.

envelopes inside Africa (Fig. 2b and Extended Data Fig. 7d), which is consistent with a deeper ancestral linkage between these two groups[25]. For *H. heidelbergensis*, we observed a time-averaged habitat suitability pattern that was qualitatively similar to that of *H. neanderthalensis* (Fig. 1b, c). By comparing the climate niches of *H. sapiens* (Fig. 1a) with those of other hominin species, we determined that our own species was best equipped to cope with dry conditions (Fig. 1g and Extended Data Fig. 6c). This extended climatic tolerance of *H. sapiens* was introduced into the CEM by a group of fossils and archaeological artefacts located in northeastern Africa, the Arabian Peninsula and the Levant (Fig. 1a and Supplementary Table 1). This tolerance of dry conditions greatly enhanced the mobility of *H. sapiens*, which may have further facilitated the documented multiple-wave dispersals into Eurasia across the Sinai passage or Bab-el Mandeb strait into the Levant (Extended Data Fig. 6c) and the Arabian Peninsula[9], respectively.

## Climate impacts on species distributions

The temporal evolution of our HSM exhibits pronounced Milankovitch cycles (Figs. 1g–i and 3, Extended Data Fig. 6 and Supplementary Videos 1–5). Tropical regions are characterized mostly by precessional cycles, which are modulated by eccentricity cycles of 80–120 thousand years (ka) and 405 ka (Fig. 1f), whereas extratropical locations show a stronger component of 80–120 ka due to $CO_2$ and ice-sheet forcings (Extended Data Fig. 6b, d). Notably, regional climate changes and the resulting habitat changes were driven not only by the interplay of local forcings but also by remote effects such as eastern equatorial Pacific temperature changes, as suggested recently[26] by a synthesis of African hydroclimate proxy records and tropical sea surface temperature reconstructions.

To further test whether orbital-scale climate variability influenced the observed spatiotemporal distribution of hominin species, we recalculated the CEM for each species using the fossil and archaeological data in combination with a time-scrambled trajectory of the CESM1.2 simulation (Methods). The resulting new CEM is different from the original one in that it assigns different, randomized temporal climate states to the fossil and archaeological data while maintaining the overall regional co-variability of the climate components and the long-term mean state. By comparing the long-term mean difference in the habitat suitability projections of the null-hypothesis model with the original one, we could then ascertain whether Milankovitch cycles influenced the distribution of fossils and archaeological sites on a regional level. The results for *H. sapiens*, *H. neanderthalensis* and *H. heidelbergensis* (Extended Data Fig. 8a–c) showed statistically significant differences

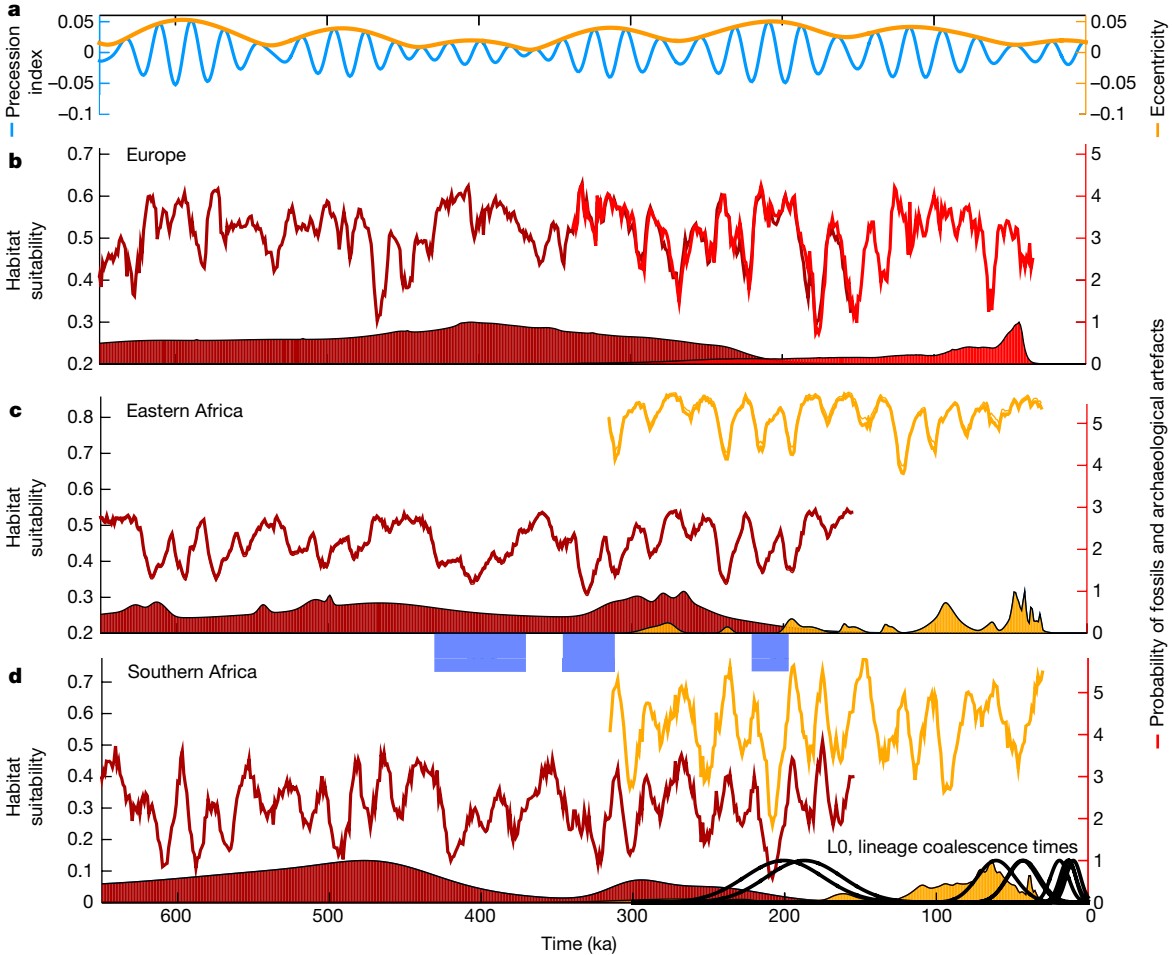

**Fig. 3 | Hominin species succession in Europe and southern Africa.**
**a**, Eccentricity (orange) and precession (blue) indices from Fig. 1f. **b**, Habitat suitability calculated from the CEM for *H. heidelbergensis* (dark red curves) and *H. neanderthalensis* (red curves) in Europe (4° × 4° average centred around 45° N, 6° E). **c**, Same as **b** but for *H. heidelbergensis* (dark red) and *H. sapiens* (orange) in eastern–central Africa (4° × 4° average centred near 5° S, 36° E). **d**, Same as **c** but for southern Africa (4° × 4° average centred near 24° S, 24° E). The shaded curves represent probability estimates of the occurrence of respective fossil and/or archaeological data obtained from the ages and age uncertainties of the fossils in the respective broader regions. The thick black curves in **d** represent the probability of the coalescence times[32] of the mitochondrial DNA lineages L0, L0d′k, L0a′b′f′g, and L0k as a genetic manifestation of deep-rooted modern human ancestry in southern Africa. Light-blue shaded bars indicate key periods of reduced habitat suitability in southern Africa. The robustness of these calculations against uncertainties in species attribution and dating of archaeological layers (Methods) is further documented in Extended Data Fig. 10.

($P < 0.05$, paired $t$-test) in the calculated habitat suitability, with values of more than 0.05 in magnitude attained when comparing the unshuffled and shuffled models over parts of Asia, Europe and Africa. This documents that the orbital-scale trajectory had an important role in determining where and when hominin species lived.

## Species successions

To identify locations where potential succession or speciation of hominin groups may have taken place, we calculated the species overlap as the co-variance of habitat suitability between the different hominin groups (Fig. 2 and Extended Data Fig. 7). We assumed that species that interacted with or emerged from each other probably shared similar regional climate envelopes, at least during their transition time.

For *H. neanderthalensis* and *H. heidelbergensis*, the highest values of niche overlap were found in Europe (Fig. 2b and Extended Data Fig. 7b), which also hosts archaeological artefacts and fossils from both species[27] (Fig. 3b) and has been regarded as the 'birthplace' of Neanderthals[28]. By comparing the zonal mean overlap for *H. sapiens* and *H. heidelbergensis* with their respective fossil and archaeological sites,

we identified two key areas with climatic conditions that were suitable for joint occupancy outside Europe: central–eastern Africa and southern Africa (Fig. 3). In addition to habitat overlap (Fig. 2), we calculated the regional habitat similarity as an indicator for potential evolutionary transitions such as baseline evolution or speciation events (Extended Data Fig. 7a). A more detailed analysis into the simulated regions of orbitally varying species overlap indicated two pronounced periods of reduced habitat suitability in southern Africa for *H. heidelbergensis* at 415–360 ka and 340–310 ka (Fig. 3d). These prolonged eras of climatic stress were further characterized by low probabilities for fossil and archaeological records in subequatorial Africa. Subsequently, from 310 ka to 200 ka, high values of habitat suitability correlated with the first evidence of *H. sapiens* in southern Africa in terms of both fossils and archaeological artefacts[29–31] (Figs. 2a and 3d) as well as presence of the earliest mitochondrial DNA lineage (L0) of southern African origin[32]. The disappearance of *H. heidelbergensis* from Africa could potentially be explained by progressive evolution of *H. heidelbergensis* into *H. sapiens*, which would be consistent with the presence of their respective fossils and archaeological artefacts at about 200–300 ka (Supplementary Table 1) and their similar values for regional habitat

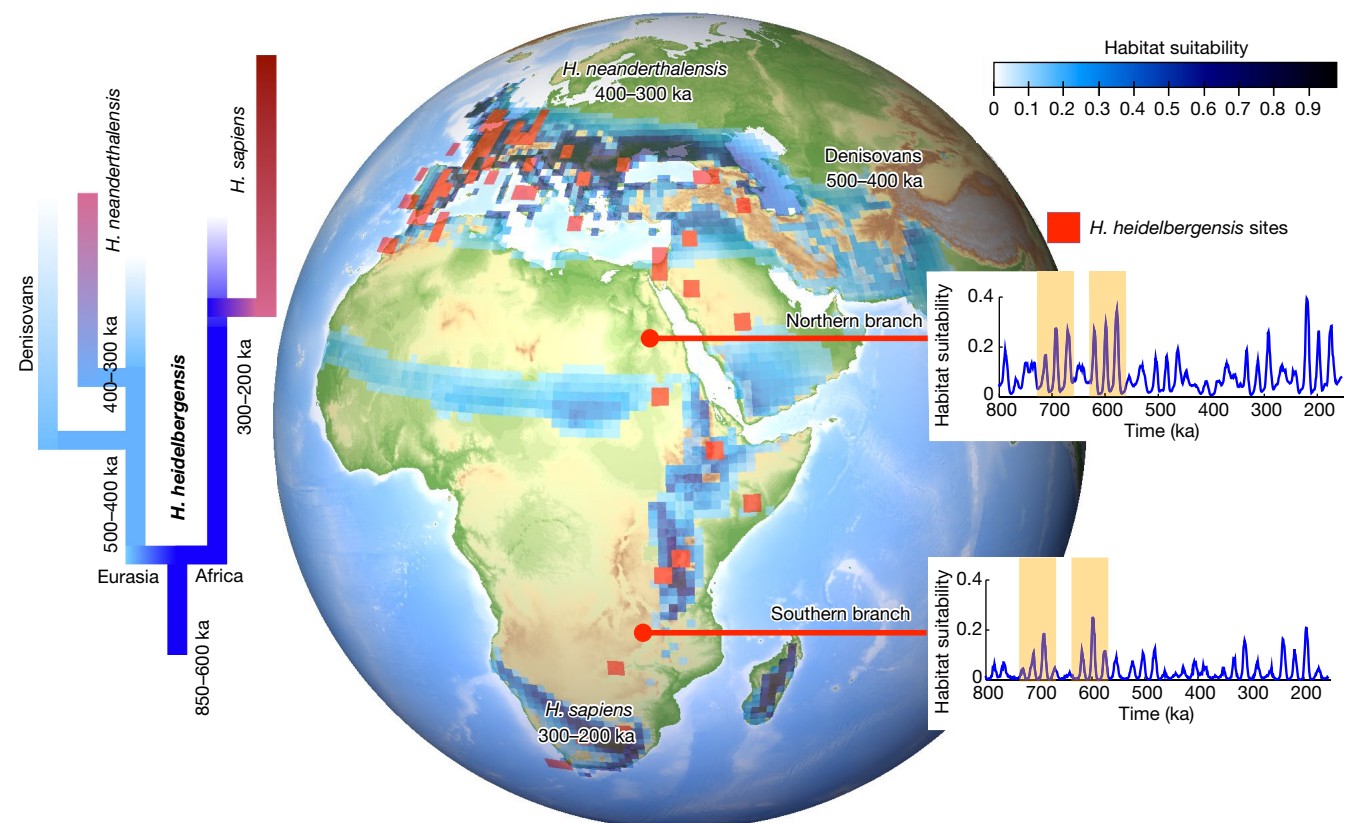

**Fig. 4 | Schematics of *H. heidelbergensis* succession.** On the basis of fossil ages, we propose a split of *H. heidelbergensis* into northern and southern branches (blue shading, habitat suitability) around 850–650 ka. The gradual transition at 300–200 ka of *H. heidelbergensis* into *H. sapiens* in southern Africa is supported by fossil and archaeological data in this region and habitat overlap estimates (Fig. 2a). The proposed divergence at 400–300 ka of *H. heidelbergensis* into Neanderthals in Europe is consistent with recent genetic estimates[34]. This scenario is also in agreement with Neanderthal whole-genome data[44] that suggest a population split between Neanderthal–Denisovan and modern human lineages between 550 and 765 ka and a divergence between Neanderthals and Denisovans around 445–473 ka. Possible eccentricity-modulated windows for early non-coastal north–south migrations occurred around 700 ka and 600 ka during periods of high eccentricity, according to the calculated HSM (see inset time series for 4° × 4° averages centred near 21° N, 31° E and 20° S, 31° E).

suitability (Fig. 3d). By contrast, a larger habitat discrepancy between *H. heidelbergensis* and *H. sapiens* (Fig. 3c) occurred in central Africa, indicating that gradual species transition or diversification is less likely to have occurred in this region than in southern Africa, at least from a climate envelope perspective. Another major climate disruption in southern Africa around 210–200 ka (Fig. 3d) during the austral summer perihelion (Fig. 3a) could have increased the regional environmental stress on *H. sapiens*, leading to dispersal and, subsequently, genetic diversification. This timing is consistent with the presence of the first known mutation event that occurred in our reconstructed common mitochondrial ancestry[32], even though considerable uncertainties in dating and methodology still exist[33]. Overall, our analysis suggests that the emergence of *H. sapiens* and the gradual disappearance of *H. heidelbergensis* in southern Africa coincided with long-term climatic anomalies during Marine Isotope Stages 11 and 9.

## Speciation and dispersal

We combined a transient Pleistocene climate model simulation with an extensive compilation of hominin fossils and archaeological artefacts to study the environmental context of hominin evolution. On the basis of the resulting HSM and palaeogenetic evidence[34,35], we propose the following scenario (Fig. 4): about 850–600 ka, *H. heidelbergensis*, which may have originated from *H. ergaster* in eastern Africa (Extended Data Fig. 7e), split into southern and northern African branches, the latter of which included northern African and Eurasian populations. The intensified dispersal into off-equatorial regions may have occurred during periods of high eccentricity around 680 ka and 580 ka, which increased habitat suitability in otherwise unhospitable regions (Fig. 4, insets). The southern branch experienced considerable climatic stress in southern Africa during Marine Isotope Stages 11 and 9, which could have accelerated either a gradual or a cladogenetic transition into *H. sapiens*[36]. The Eurasian populations of the northern branch further bifurcated around 430 ka, possibly giving rise to Denisovans, which populated parts of central and eastern Asia. Inside central Europe, *H. heidelbergensis*, which experienced strong local climatic stress due to eccentricity-modulated ice-age cycles (Fig. 3b), gradually evolved into *H. neanderthalensis* between 400 ka and 300 ka. Side branches to northwestern Africa, back-propagation, multiple dispersals[37], interbreeding[38] and subsequent speciation[39] may have further complicated the picture.

Recent studies have suggested that the sequence of hominin speciation events and the long-term positive trend in brain size may have been linked to past climatic shifts in Africa[40]. Our analysis supports the notion of strong Milankovitch cycles in early hominin habitat suitability in central Africa (Fig. 1h). Moreover, during the early Pleistocene (2–1 Ma), early African *Homo* populations occupied two main habitats: one in central–eastern Africa and the other in southern Africa (Fig. 2b). On average, these groups preferred geographical regions that were characterized by relatively stable NPP values of 200–380 gC m$^{-2}$ per

year (Fig. 2c). Within Africa, early African *Homo* populations adapted mostly to local orbital-scale variations in climate and NPP (Fig. 1g–i and Extended Data Figs. 3, 4 and 9), as reflected also in their habitat suitability. After the mid-Pleistocene transition and with the emergence of *H. heidelbergensis* between approximately 885 and 865 ka, the dynamics again changed remarkably. *H. heidelbergensis* began to migrate into Eurasia and other regions, encountering along their journey a much wider spatial range of NPP, from extremely low values of 20 gC m$^{-2}$ per year to values exceeding 600 gC m$^{-2}$ per year (Fig. 2c). These migrating groups crossed large spatial gradients in climate and NPP that far exceeded the temporal ranges in NPP experienced by their more stationary Early Pleistocene predecessors. This transition to global wanderers about 0.8–0.6 Ma must have required *H. heidelbergensis* to acquire new adaptation skills, which in turn also strengthened their ability to further expand their geographical range, thereby providing a strong positive migration–climate adaptation feedback. Our analysis clearly shows that *H. erectus* had already undergone such a transition from regional dweller to early global wanderer before 1.8 Ma (Fig. 2c). Together with the *H. heidelbergensis* evidence, this indicates that dispersals from Africa always involved an adaptive shift, either biological or cultural, to wider climate envelopes. Therefore, to understand hominin evolution during the Pleistocene, the full spatial and temporal complexity of the climate signal and the corresponding habitat suitability must be considered.

## Discussion

The main conclusions of our analysis are robust with respect to the existing uncertainties in species attribution, particularly for the period from 1 to 0.3 Ma, and the dating of archaeological layers, as demonstrated by key HSM calculations with four different scenarios that accounted for these factors (Methods and Extended Data Fig. 10). Although our study is based on species-stratified fossil and archaeological input data, our calculation of species overlap as HSM co-variability allowed us to treat potential species transitions and successions in human evolutionary history quantitatively and to identify their spatiotemporal characteristics. To the best of our knowledge, such research has not been reported thus far. The HSM captures regionally distributed patchworks of habitable areas in agreement with a general multiregional perspective[41] (Figs. 1 and 4). According to our CEM, southern and eastern Africa as well as the region north of the Intertropical Convergence Zone emerge as potential long-term refugia for various types of archaic humans. As the climate changed on orbital timescales, these refugia shifted geographically, creating population patterns with greater complexity. Further analysis of the pan-African connectivity of refugia in our HSM dataset, as shown in the inset in Fig. 4, will increase understanding of hominin dispersal, interbreeding and cladogenetic transitions as well as potential cultural exchanges.

In summary, we demonstrated that astronomically forced climate shifts were a key factor in driving hominin species distributions[42] and dispersal and were probably important for diversification[43].

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

## Methods

### 2Ma simulation

We conducted the 2Ma simulation with the Community Earth System Model (CESM), version 1.2, at an ocean and atmosphere resolution of approximately 3.75° × 3.75°. The model uses bathymetry of the Last Glacial Maximum and time-varying forcings of greenhouse gases[15], ice sheets[15] and astronomical insolation conditions[16]. CESM1.2 has a relatively low standard equilibrium climate sensitivity (ECS) of 2.4 °C per $CO_2$ doubling, which lies outside the likely range of estimates[45] (3.7 ± 1.2 °C) obtained with other climate model simulations conducted as part of the Coupled Model Intercomparison Project, phase 6. However, this value is within the lower range of recent estimates compiled by the Intergovernmental Panel on Climate Change sixth assessment report[46] of Working Group 1, which identifies a very likely ECS range of 2–5 °C. To obtain a more realistic response to past long-wave radiative forcings in our palaeo-climate model simulation and to implicitly capture radiative effects of other $CO_2$-correlated forcings[47] from dust, vegetation, $N_2O$ or $CH_4$, we therefore scaled the range of the applied $CO_2$ forcing[15] by a factor of 1.5. The resulting effective ECS, which includes non-$CO_2$ greenhouse gas forcings, was in our case approximately 3.8 °C. Our result is in reasonable agreement with the Coupled Model Intercomparison Project phase 6 estimate and previous palaeo-climate estimates[18,19] of 3.2 °C, which were obtained from reconstructions of the global mean surface temperature and radiative forcings covering the last 784,000 years. Amplification of the $CO_2$ forcing in CESM1.2 led to a realistic representation of the amplitude of global mean, tropical and Antarctic temperature changes (Extended Data Figs. 1b and 2a, b) and to a simulated temperature range between Last Glacial Maximum and Late Holocene conditions of approximately 5.9 °C. This result is in close agreement with recent palaeo-proxy-based estimates[20]. Similar to previous long-term transient climate model simulations conducted with Earth system models of intermediate complexity[7,48], the CESM1.2 simulations use an orbital acceleration factor of 5, which means that the 2-million-year orbital history is squeezed into 400,000 model years in CESM. The complete model trajectory is based on 21 individual chunks that were run in parallel, with each covering at least one interglacial–glacial cycle (Supplementary Table 2). Moreover, each chunk overlaps with the next chunk so that the issue of initial conditions and spin-up time can be evaluated properly. The final climate trajectory is obtained by combining the individual chunks and by using sliding linear interpolation in the chunk-overlap periods. The model simulation has been evaluated against numerous palaeo-proxy-based data (Fig. 1 and Extended Data Fig. 1). Unlike other Earth system models[49–51], the 2Ma simulation conducted with CESM1.2 does not generate strong internal millennial-scale variability such as that shown by Dansgaard–Oeschger cycles. The CESM1.2 data are provided on the climate data server of the Institute for Basic Science (IBS) Center for Climate Physics at https://climatedata.ibs.re.kr.

### Topographic downscaling

The T31 spectral resolution of the 2Ma CESM1.2 simulation (approximately 3.75° × 3.75° horizontal resolution) is too coarse to properly capture important topographic barriers, which may have affected the dispersal and distribution of archaic humans. We applied simple downscaling of the simulated monthly surface temperatures $T_s(t)$ onto a 1° × 1° horizontal grid by accounting for the difference in height $\Delta h(t)$ between the ETOPO5 topographic dataset and the orographic forcing of the 2Ma experiment. The lapse rate-corrected temperatures were then calculated as $T^*_s(t) = T_s(t) - g\Delta h(t)$, where $g$ represents a constant average lapse rate of $g = 6$ °C per 1,000 m. The simulated rainfall $p(t)$ was downscaled onto the high-resolution topography by accounting for temperature-dependent moisture availability through the Clausius–Clapeyron equation as $p^*(t) = p(t)e^{[17.625T^*_s/(T^*_s + 243.04) - 17.625T_s/(T_s + 243.04)]}$.

### A posteriori calculation of NPP

2Ma uses fixed plant functional types but a prognostic leaf area index. Therefore, we calculated the NPP a posteriori (Extended Data Figs. 5 and 9) using a simple empirical relationship among temperature, precipitation and tree fraction. The topographically downscaled temperature $T^*_s$ (in degrees Celsius) and precipitation $p^*$ (in millimetres per year) of the 2Ma simulation were used at every grid point to calculate the tree fraction[52] as $\tau = 0.95\{1 - e^{[-\beta(T^*_s - T_m)]}\}p^{*\alpha}/(p^{*\alpha} + f)$, with the additional term $f = be^{[\gamma(T^*_s - T_m)]}$, and the parameters $\beta = 0.45$, $\alpha = 3$, $b = 2.6 \times 10^6$, $\gamma = 0.155$ and $T_m = -15$ °C; $\tau$ is capped between 0 and 1. Subsequently, the downscaled NPP can be calculated from an empirical model[53] as $N^* = \{6,116[1 - e^{(-0.0000605p^*)}](1 - \tau) + \tau \min(FP, FT)\}f(CO_2)$, where the minimum (min) is taken over the mathematical terms $FP = 0.551p^{*1.055}/e^{(0.000306p^*)}$ and $FT = 2,540/[1 + e^{(1.584 - 0.0622T^*_s)}]$. The function $f(CO_2) = [1 + 0.4\ln(CO_2/280)/\ln(2)]$ captures the bulk effect of $CO_2$ fertilization of plants[54] in the same way as the CLIMBER Earth system of intermediate complexity, and its time evolution is obtained from the transient $CO_2$ forcing of CESM1.2.

### Extended dataset of archaeological and fossil hominin data

The SDM for the *Homo* genus was derived from a recent compilation of archaeological and fossil data[21]. The original data compilation published in 2020 (ref. [21]) included 2,754 radiometric age estimates for fossil hominin occurrences, each accompanied by the confidence interval around the estimate, the fossil site name and the archaeological layer within the site (where available) from which the dated sample was derived, the geographical coordinates of the site and the possible attribution to one or more than one *Homo* species. Confident attributions to a single species generated a core record, whereas instances with multiple attributions formed an extended record. Six different species were recognized: *H. habilis*, *H. ergaster*, *H. erectus*, *H. heidelbergensis*, *H. neanderthalensis* and *H. sapiens*. The updated record, as shown in Supplementary Table 1, contains 3,245 data entries restricted to the temporal age interval of 2 Ma–30 ka; those from Australia and the Americas were excluded. Further, we combined *H. habilis* and *H. ergaster* into a single African Oldowan toolmaker species. Each occurrence is attributed to a given species depending on the presence of unambiguous anatomical remains, either singly or in connection to a specific lithic tool industry. This helped to guide identification if this was not otherwise possible from the bones and teeth alone (398 entries, 12%), the age limits of the individual species or the stone tool industry. For example, an occurrence in Africa older than the first appearance of *H. sapiens* at Jebel Irhoud[55] yet younger than the first appearance of *H. heidelbergensis* at Melka Kunture[56] is attributed to *H. heidelbergensis*. Moreover, French Mousterian stone tools have been unambiguously assigned to *H. neanderthalensis*, whereas Aurignacian tools were attributed to *H. sapiens*. When these criteria were applied, the core record included 94.5% of the attributions, 48.5% of which refer to *H. neanderthalensis* and 37.5% of which refer to *H. sapiens*. Where neither of these criteria was met (in the original compilation, the SDM acknowledges attribution uncertainty), we accounted for this by testing the stability of our results with respect to different versions of the SDM (Extended Data Fig. 10). For example, transitional industries (for example, the Levantine Mousterian or Lincombian–Ranisian–Jerzmanowician industries) received multiple attributions because they fit either *H. sapiens* or *H. neanderthalensis* in terms of toolmaker identity[57,58]. A detailed explanation of this approach is provided as supplementary material for ref. [21] (https://ars.els-cdn.com/content/image/1-s2.0-S259033222 0304760-mmc1.pdf).

A second source of uncertainty stems from dating. Although approximately 50% of the entries refer to the [14]C method (>90% of which are based on accelerator mass spectrometry), other dating methods such as electron spin resonance (14% of the sample), thermoluminescence (12%) and optically stimulated luminescence (12%) are less precise than

radiocarbon dating. Nonetheless, multiple datings are present for individual fossil sites, even within a single stratigraphic layer at the site. To account for uncertainties in species attribution and age, we ran our analyses according to the four different approaches described below.

1. Multiple dates, tier 1. Only the core record, which excludes entries with uncertain species attributions, and all age estimates available for each archaeological layer are used. Multiple age estimates per layer are possible, and the age uncertainty for each is included in our analysis. This subdivision includes 3,060 data entries. Although the main analysis in our study is based on this case, we need to consider possible sampling biases due to the higher weights given to archaeological layers with multiple dates (Figs. 1–4 and Extended Data Fig. 10).

2. Multiple dates, tier 2. The extended record, in which ambiguous species attributions are treated by randomly choosing among the possible candidate species, is used along with multiple age estimates (including uncertainties) per layer. This subdivision includes 3,245 (all) data entries (Extended Data Fig. 10).

3. Single date, tier 1. Multiple age estimates for a single archaeological layer are combined in this approach to provide a minimum and maximum age for the layer. Each archaeological layer has only one entry, thereby eliminating possible sampling biases in the estimation of our CEM. This subdivision includes 1,567 data entries (Extended Data Fig. 10).

4. Single date, tier 2. Age estimates for archaeological layers are treated as those in the single date, tier 1 category except that the extended record rather than the core record is used. This subdivision includes 1,652 data entries (Extended Data Fig. 10).

We acknowledge that our species subdivisions may be controversial and that these do not necessarily require constancy of morphology, habitat and behaviour. However, even though some species attributions such as *H. heidelbergensis* could be questioned, we remain confident that the majority of the record presents little challenge considering that 86% of the core data belong to the well-defined, widely accepted *H. neanderthalensis* or *H. sapiens* record and tool-making traditions. Thus, even though some species attributions might be considered invalid, widely accepted constraints are used. Clearly, to the best of current knowledge, 500,000-year-old remains in Africa can belong to neither *H. sapiens* nor H. *habilis*[59], irrespective of whether the name *H. heidelbergensis* is considered appropriate. To further reduce uncertainties, we tested the robustness of our main findings with four alternative scenarios (Extended Data Fig. 10) for species attribution and dating and excluding uncertain and poorly dated species (for example, *Homo floresiensis*, *Homo naledi*, *Homo bodoensis*, *Homo longi* and Denisovans), which are restricted to too few fossil sites for which no climatic variability can possibly be ascertained or do not currently include any other locality or remains in their definition. The final species assignments used in our study should be interpreted here as plausible working hypotheses.

## Mahalanobis CEM

To derive the CEM (Extended Data Fig. 5) that best characterizes the habitable conditions for hominins, we chose four key climatic variables: annual mean temperature and precipitation ($T^*_{am}$ and $P^*_{am}$, respectively), annual minimum precipitation ($P^*_{min}$) and terrestrial NPP ($N^*$). Obtained as 1,000-year downscaled averages (1° × 1° horizontal resolution), these variables, which relate to physiological constraints for hominin survival and the availability of food resources, are combined as a four-dimensional climate environment vector $\mathbf{C}(t) = (T^*_{am}, P^*_{am}, P^*_{min}, N^*)$ with 2,000 values in time ($t$) corresponding to 1,000-year (200-year) orbital (model) means from the 2Ma simulation. The fossil and archaeological data entries for the five individual hominin groups are described in the previous section. Although our main analysis focuses on the multiple date, tier 1 case (Methods and Supplementary Table 1), the robustness of our results was tested against other ways of treating species and age model uncertainties (Extended Data Fig. 10). The data entries are represented by their longitude $\lambda_{j,i}$ and latitude $\varphi_{j,i}$ coordinates and the respective average age $t_{j,i}$ and age uncertainties $\Delta t_{j,i}$ with $i = 1, ..., 5$ corresponding to the five hominin groups. We defined the fossil state vector as $\mathbf{z}^i = (\lambda_{1,i}, \varphi_{1,i}, t_{1,i}, \Delta t_{1,i}, ..., \lambda_{n,i}, \varphi_{n,i}, t_{n,i}, \Delta t_{n,i})$ with $n_i$ representing the total number of fossils in each group during the past 2 million years. We then built the matrix $D$ from the four-dimensional climate data subsampled at the fossil data sites and the corresponding nearest ages. Age uncertainties were considered through a Monte Carlo sampling method, which expanded the length of the overall data vector. We obtained $D^i$ (4 × $N_i$ matrix for each $i = 1, ..., 5$) for each hominin group as $D^i = (T^*_{am}(\mathbf{z}^i), P^*_{am}(\mathbf{z}^i), P^*_{min}(\mathbf{z}^i), N^*(\mathbf{z}^i))$. We then calculated the Mahalanobis squared distance model[23] for each group using $\zeta_i^2(D^i) = (D^i - <D^i>)^T S^{-1} (D^i - <D^i>)$, where $<...>$ represents the sample mean value and $S^{-1}$ is the inverse co-variance matrix obtained from the data. The Mahalabonis squared distances $\zeta_i$ were then transformed into a cumulative chi-squared distribution $\chi^2_{CDF}$ in the four-dimensional climate data space $C$. When using 4 degrees of freedom[23], the corresponding probability $H(C) = 1 - \chi^2_{CDF}(C)$ represents the likelihood of finding a fossil for a specific quadruplet within the four-dimensional climate data space in the HSM. We interpreted $H$ as a probability, which we refer to as habitat suitability. Given the temporal evolution of $C$ for every grid point of the downscaled 1° × 1° data over the last 2 million years, we were able to calculate the spatiotemporal habitat suitability for each downscaled grid point $(x,y,t)$ in the model as $H(x,y,t) = H(T^*_{am}(x,y,t), P^*_{am}(x,y,t), P^*_{min}(x,y,t), N^*(x,y,t))$. The stability of the HSM was tested by using different dimensionalities and combinations of climate parameters such as annual mean and seasonal range of temperature and precipitation and annual mean and minimum values of temperature and precipitation. The key conclusions of our study remained essentially unchanged. Moreover, we tested the stability of our results against the omission of hominin sites with ambivalent original species attributions (multiple date, tier 2) and different treatment of archaeological ages (single date, tiers 1 and 2). The calculated $H(x,y,t)$ was qualitatively very similar for the four different cases (Extended Data Fig. 10). Therefore, our main conclusions remain robust with respect to uncertainties in species attribution and archaeological layer dating.

## Random climate trajectory

To address the question of whether the actual climate trajectory influenced the distribution of fossil and archaeological data, we developed a CEM and HSM in which fossil data were assigned to randomly chosen climate states from the CESM1.2 simulation under the constraint that the climate range selected must overlap with the total fossil age range of the respective species. We randomized the time variability of the four-dimensional climate data vector (annual mean temperature, annual mean precipitation, minimum precipitation and NPP) while keeping the co-variability among the climate vector components, as well as the mean state, invariant. The original HSM ($H$), which is based on the real trajectory of CESM1.2, and the model ($H_{scr}$) that we trained from a scrambled trajectory were then compared. The time-averaged differences between the models for *H. sapiens*, *H. neanderthalensis* and *H. heidelbergensis* were then interpreted as an indication of whether the realistic climate evolution influenced the observed hominin distributions in space and time relative to a system that maintains its orbital climate co-variance and mean state (Extended Data Fig. 8a–c) but does not consider the exact time evolution of glacial–interglacial and orbital cycles. The time-averaged difference between $H(x,y,t)$ in the original HSM and $H_{scr}(x,y,t)$ in the HSM derived from time-randomized climate data was then tested at each grid point using a paired $t$-test.

## Reporting summary

Further information on research design is available in the Nature Research Reporting Summary linked to this paper.

## Data availability

The CESM1.2 data and the calculated hominin habitat suitability data are available on the climate data server at https://climatedata.ibs.re.kr. The database of hominin remains and artefacts used here is provided in Supplementary Table 1. The maps in Fig. 1 and Extended Data Figs. 7, 8 and 10 were generated using M_Map: a mapping package for MATLAB, version 1.4m, available at http://www.eoas.ubc.ca/~rich/map.html. The map in Fig. 4 was generated using the software Paraview, freely available at https://www.paraview.org.

## Code availability

The MATLAB codes used to generate Figs. 1–3 will be shared on the climate data server at https://climatedata.ibs.re.kr. The CESM1.2 code is available at https://www.cesm.ucar.edu/models/cesm1.2/.

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

**Acknowledgements** A.T., K.-S.Y., D.L. and E.Z. received funding from IBS under IBS-R028-D1. M.W. acknowledges funding from DFG GA 1202/2-1, BMBF PalMod. C.Z. was supported by the Swiss Platform for Advanced Scientific Computing, grant F-1516 74105-03-01. The 2Ma CESM1.2 simulations were conducted on the ICCP/IBS supercomputer Aleph, a Cray XC50-LC system.

**Author contributions** A.T. designed the study, developed and coded the HSM, downscaled the CESM1.2 data, prepared the figures and wrote the initial draft of the manuscript. K.-S.Y. conducted the CESM1.2 2Ma simulation. P.R. and A.M. prepared the extended version of the fossil and archaeological dataset. All other authors contributed to the interpretation of the data and the writing of the manuscript.

**Competing interests** The authors declare no competing interests.

**Additional information**
**Correspondence and requests for materials** should be addressed to Axel Timmermann.

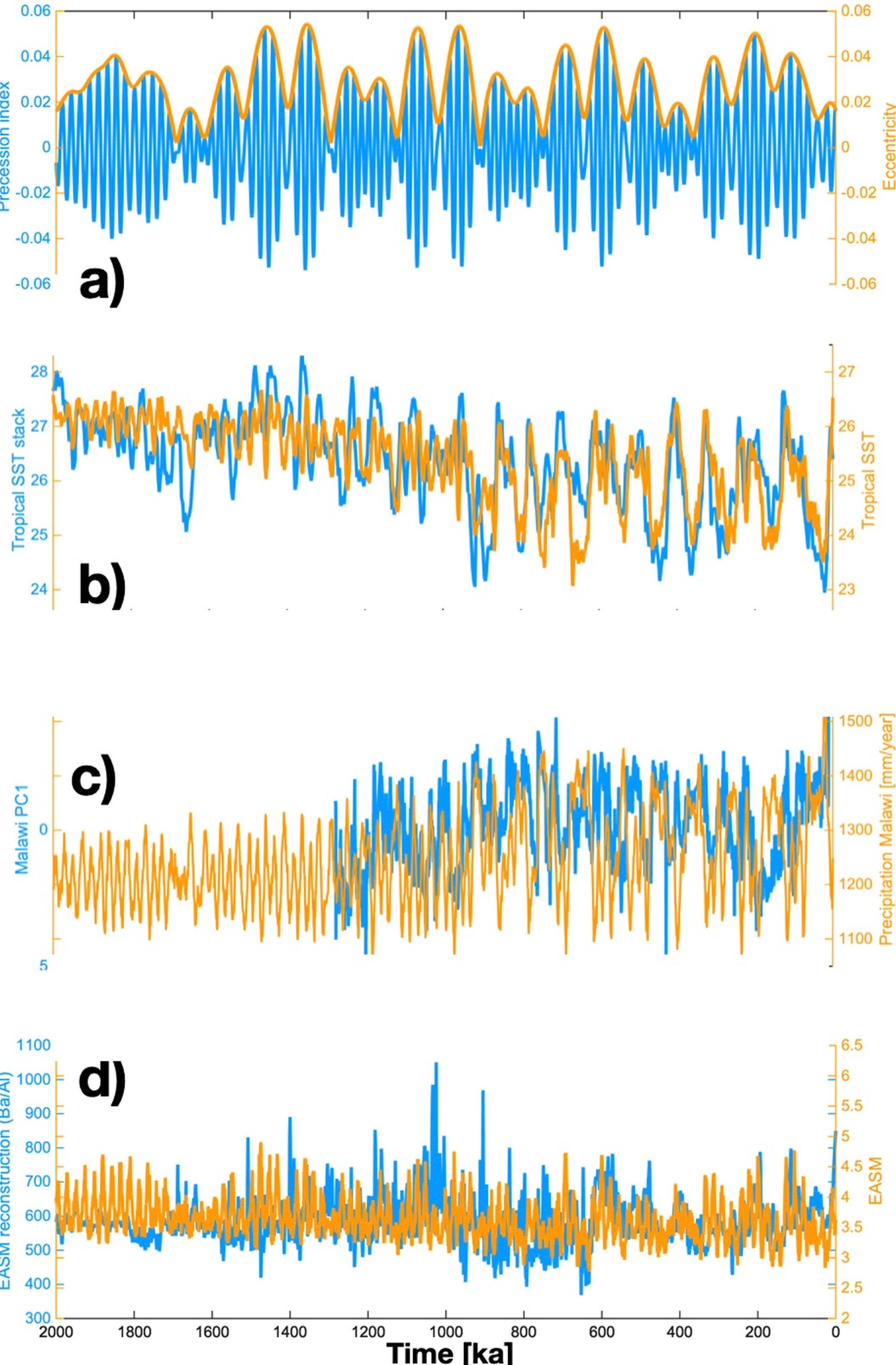

**Extended Data Fig. 1 | Model/data comparison.** a) precession index (blue) over last 2 million years and eccentricity timeseries (orange); h) reconstructed[60] (blue) and simulated (orange, CESM1.2) tropical sea surface temperatures; i) hydroclimate reconstruction from Lake Malawi[61] (blue, southeastern Africa) and simulated precipitation (orange); j) geochemical proxy (Ba/Al ratio) representing variations of the East Asian Summer Monsoon[62] (EASM) (blue) and simulated rainfall from CESM1.2 (orange) for the location 116.2°E, 19.3°N (mm/day).

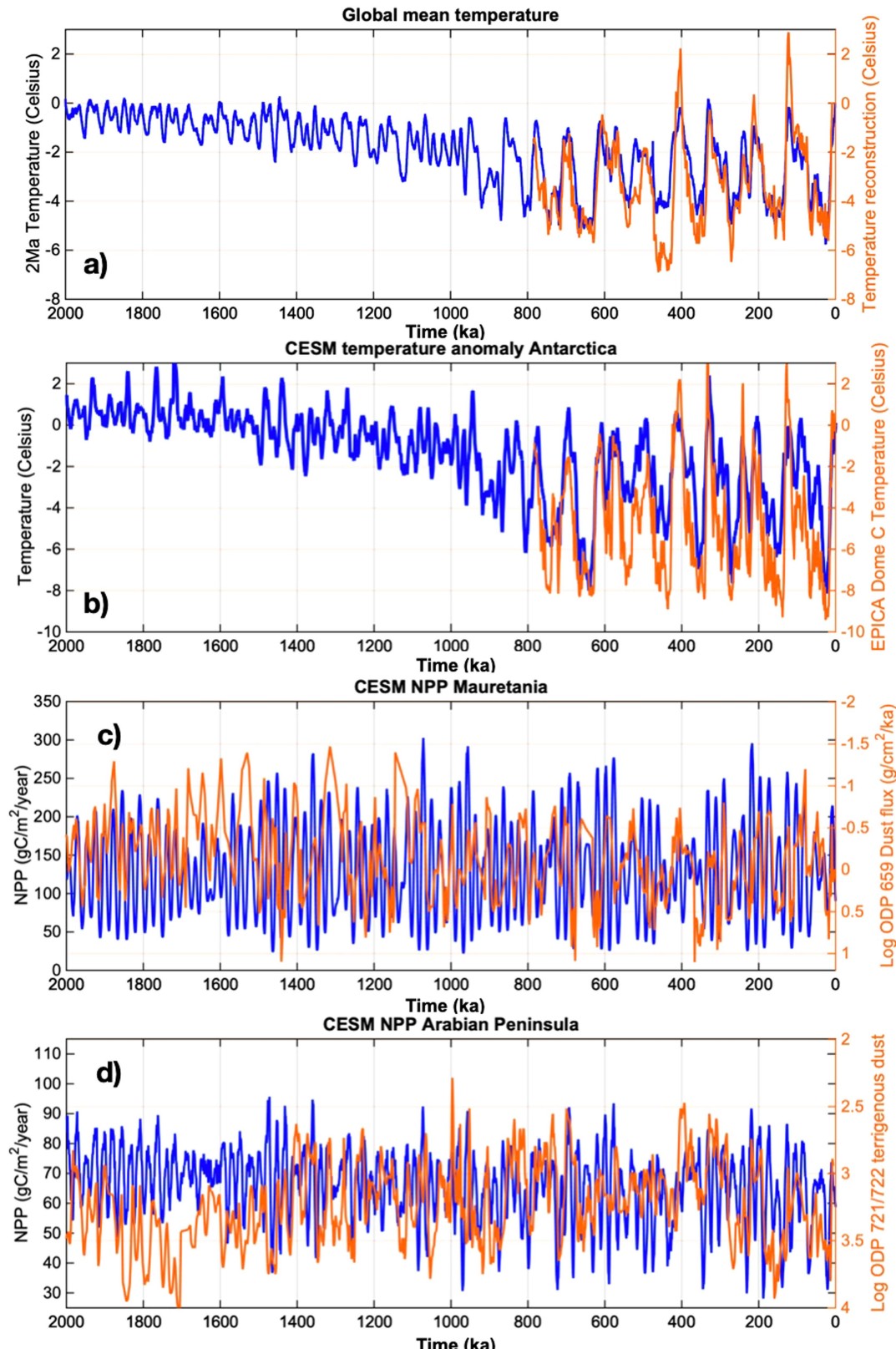

**Extended Data Fig. 2 | Model/data comparison. a)** simulated global temperatures anomalies (relative to pre-industrial conditions) (blue) and palaeo-climate reconstruction[18]; **b)** simulated regional temperature in Antarctica (5°x5° degree average over Dome C location) (blue) with deuterium based temperature reconstruction from EPICA Dome C[63]; **c)** calculated net primary production for northwestern Sahara (blue) with logarithm of dust flux from marine core ODP659[64]; **d)** calculated net primary production for Arabian Peninsula (blue) with logarithm of terrigenous dust concentration from marine core ODP721/722[65].

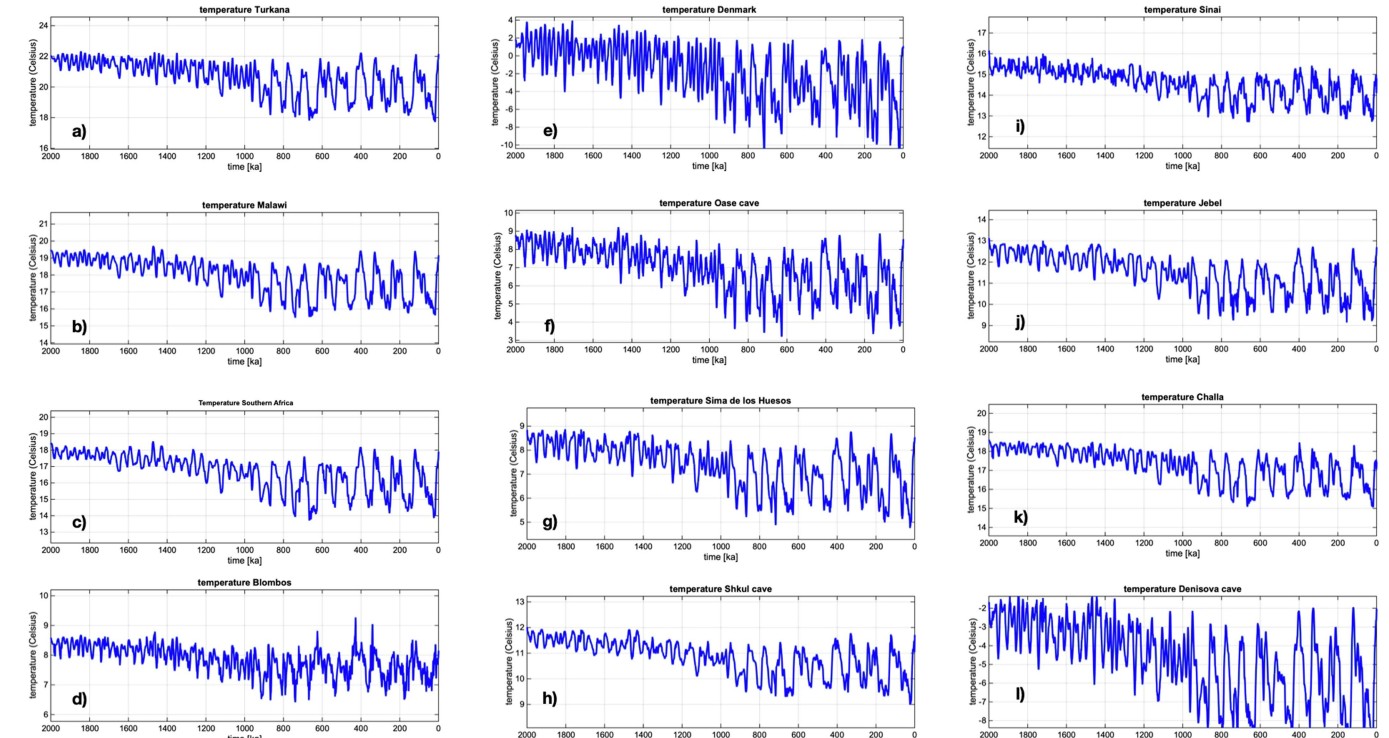

**Extended Data Fig. 3 | Surface temperature near sites relevant for interpretation of hominin evolution.** from a-l: Lake Turkana (3°N, 36°E), Lake Malawi (12°S, 34°E), Southern Africa (25°S, 25°E), Blombos cave (34°S, 21°E), Denmark (56°N, 9°E), Peçtera cu Oase (45°N, 22°E), Sima de los Huesos (42°N, 5°W), Shkul cave (33°N, 35°E), Sinai (30°N, 34°E), Jebel Irhoud (34°N, 4°W), Lake Challa (3°S, 38°E), Denisova, cave (52°N, 84°E). The timeseries represents a spatial mean of 7°x7° around the center location.

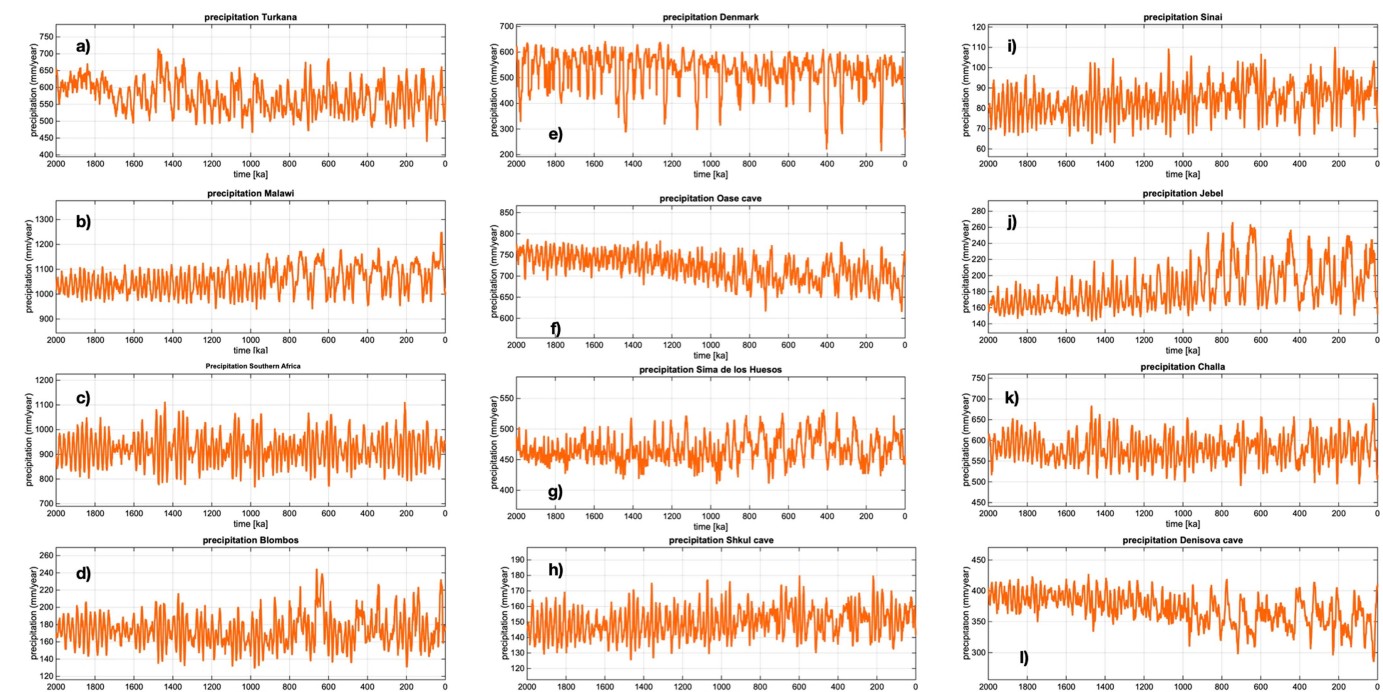

**Extended Data Fig. 4 | Precipitation near sites relevant for interpretation of hominin evolution.** from a-l: Lake Turkana (3°N, 36°E), Lake Malawi (12°S, 34°E), Southern Africa (25°S, 25°E), Blombos cave (34°S, 21°E), Denmark (56°N, 9°E), Peçtera cu Oase (45°N, 22°E), Sima de los Huesos (42°N, 5°W), Shkul cave (33°N, 35°E), Sinai (30°N, 34°E), Jebel Irhoud (34°N, 4°W), Lake Challa (3°S, 38°E), Denisova, cave (52°N, 84°E). The timeseries represents a spatial mean of 7°x7° around the center location.

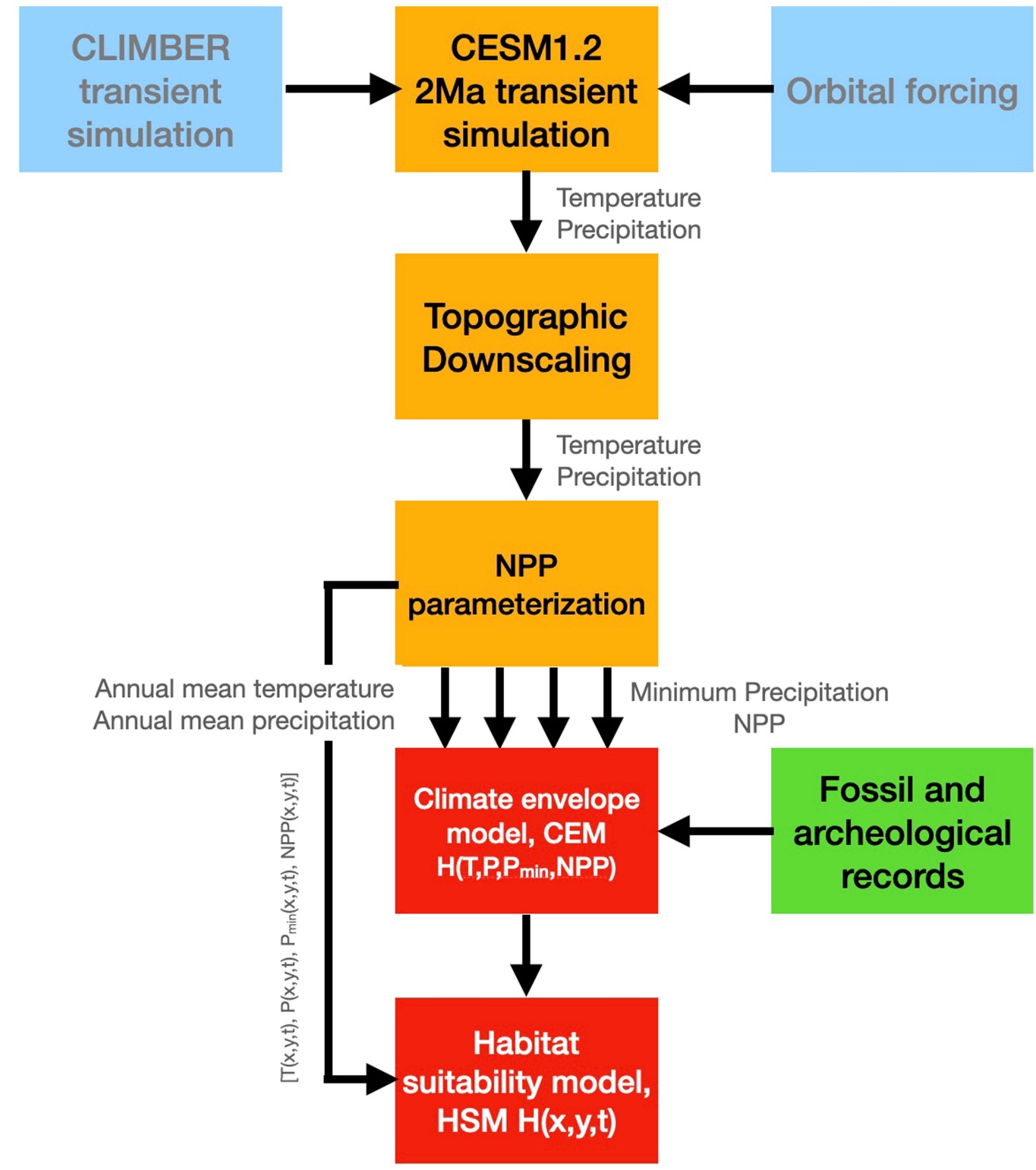

**Extended Data Fig. 5 | Schematic of Climate envelope model (CEM) and Habitat suitability model (HSM) set-up.** The orbital scale transient 2Ma simulation of CESM1.2 is conducted by using $CO_2$ and ice-sheet forcings from an intermediate complexity model simulation[15] and orbital forcing from astronomical estimates[16]. The simulated surface temperatures and precipitation from 2Ma on a ~3.75-degree horizontal grid are then downscaled to a 1x1 degree horizontal grid by including lapse-rate corrected topographic features. Net Primary Production is calculated from empirical parameterizations of the downscaled temperature and precipitation fields (Extended Data Fig. 8d–f). Using an extended fossil and archaeological database (Supplementary Table 1) in combination with the downscaled annual mean temperatures, annual mean rainfall, minimum rainfall and net primary productivity, a statistical Mahalanobis distance-based climate envelope model (CEM) is derived. The model is then forced for every land grid point on a 1x1 degree grid with the temporal evolution of the downscaled climate variables to obtain the temporal evolution of the habitat suitability (HSM) for each of the 5 hominin groups and at every grid point. The impact of resolution on key features in simulated net primary production is further illustrated in Extended Data Fig. 8d–f.

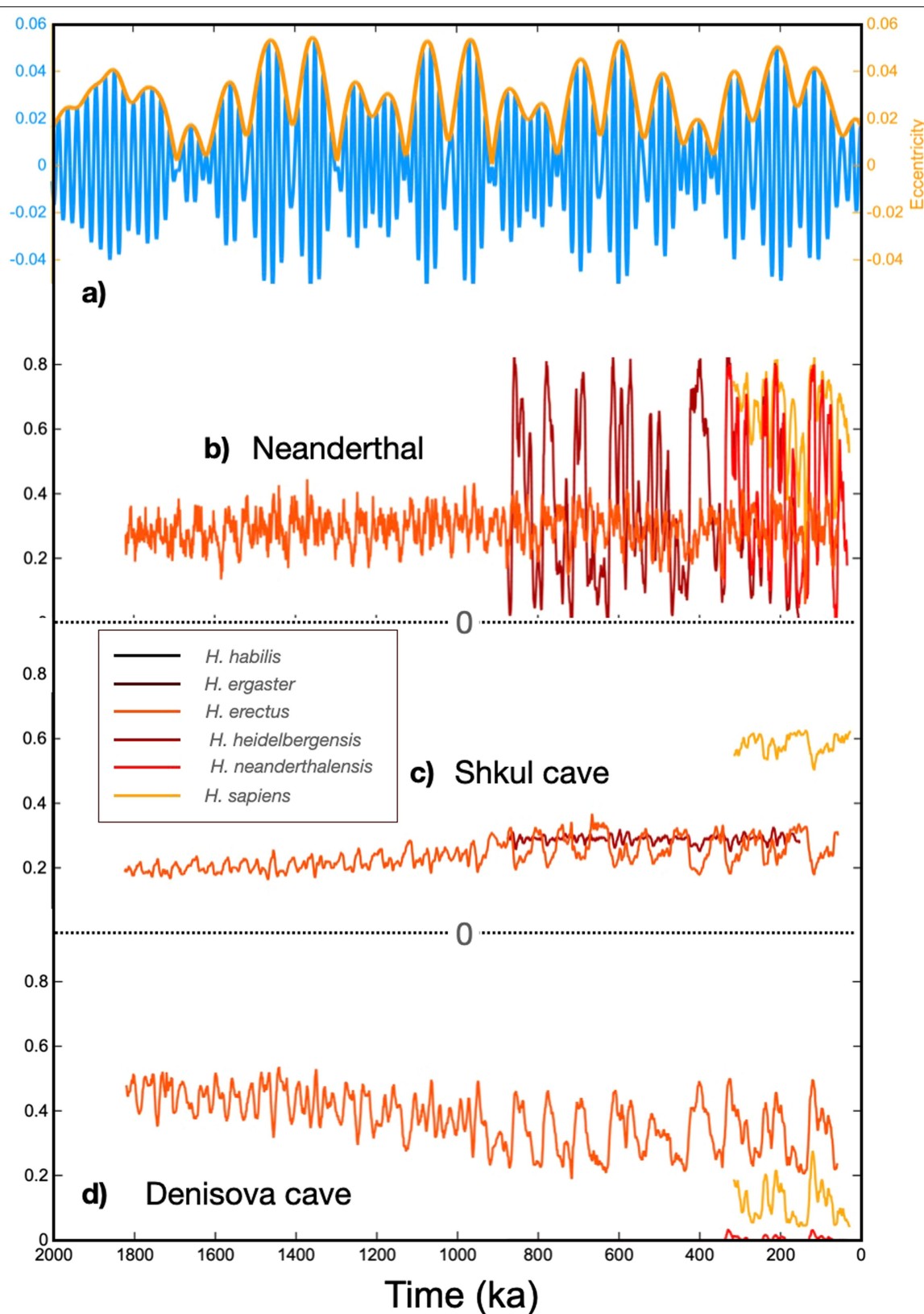

**Extended Data Fig. 6 | Orbital effects on regional habitability in Eurasia.** Timeseries for precession (blue) and eccentricity (**a**) and simulated regional habitat suitability at selected sites of archaeological interest for *Homo habilis*, *Homo ergaster*, *Homo erectus*, *Homo heidelbergensis*, *Homo neanderthalensis*, and *Homo sapiens*. The centre locations of a 5°x5° averaging area are selected as: **b**) Neanderthal (51°N, 7°E); **c**) Shkul cave (33°N, 35°E); **d**) Denisova Cave (51°N, 85°E).

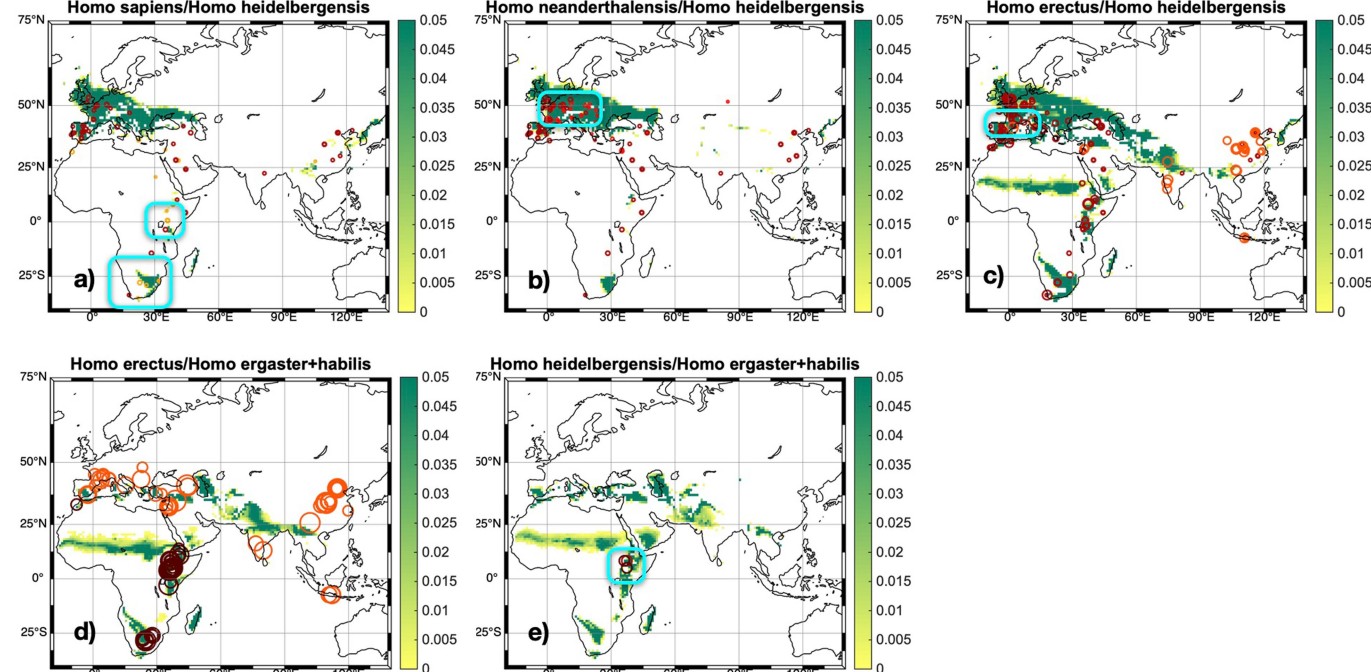

**Extended Data Fig. 7 | Potential hominin overlap regions. a)** Shading indicates the relative occurrence frequency (relative to the entire 2-million-year simulation history) of when the habitat suitability for *Homo sapiens* and *Homo heidelbergensis* at a grid point both exceed 0.3 and exhibit a difference of less than 0.1. This constraint is considered here as a measure of habitat similarity during overlap times. Circles show the respective fossil and archaeological sites from the overlap periods. Cyan boxes highlight areas, for which there are mixed fossils available and high habitat similarity. These areas are potential areas for speciation events or gradual evolutionary transformations. **b)** same as a), but for *Homo neanderthalensis* and *Homo heidelbergensis*; **c)** same as a) but for *Homo erectus* and *Homo heidelbergensis*; **d)** same as a) but for *Homo erectus* and *Homo ergaster/habilis;* **e)** same as a) but for *Homo ergaster/habilis* and *Homo heidelbergensis*. The size of the circles scales with the age of the fossils.

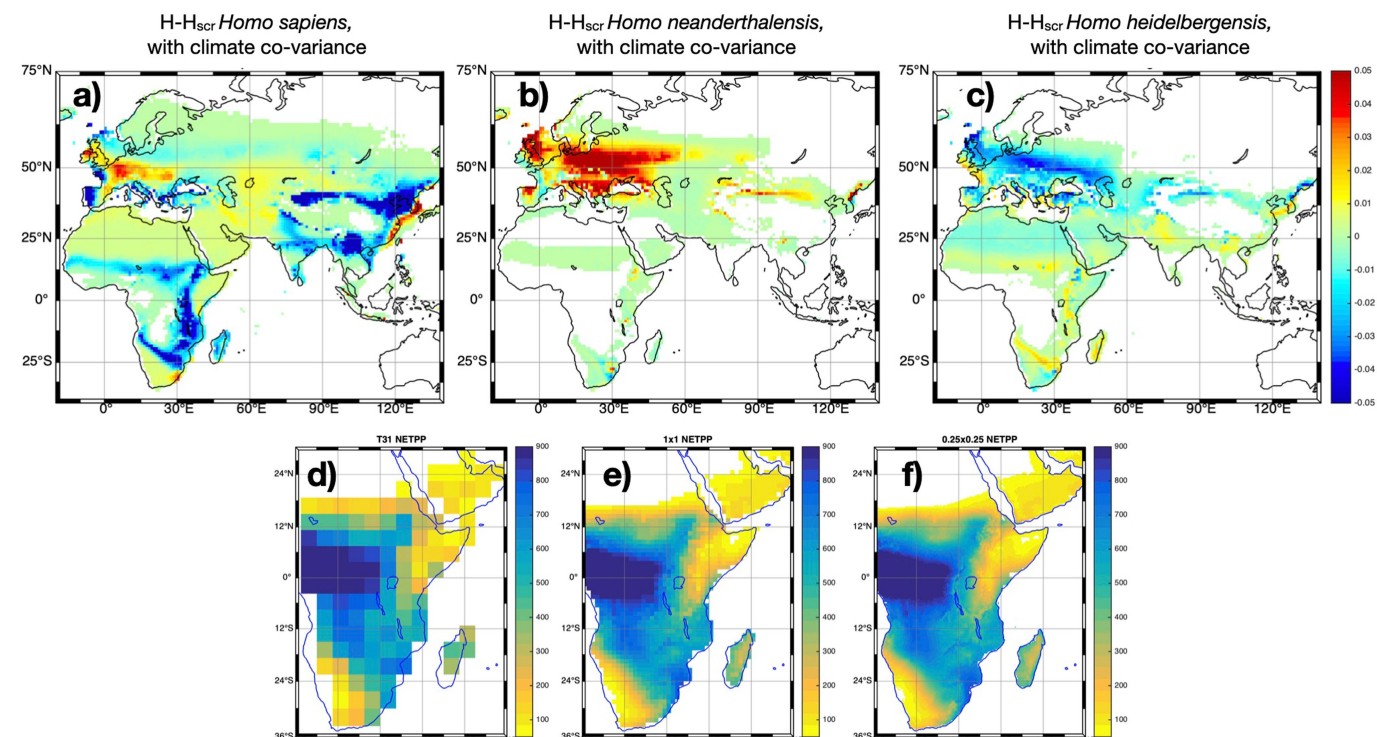

**Extended Data Fig. 8 | Climate effects on time-mean habitat suitability of Homo sapiens, Homo neanderthalensis, Homo heidelbergensis.**
**a**–**c**) difference between time-averaged habitat suitability estimated from original climate envelope model $H$ and climate envelope model $H_{scr}$ calculated from vector-randomized climate variables (Methods) for 3 hominin groups. This type of time randomization (Methods) maintains the time-mean and climate co-variance among the 4-dimensional climate input vector. Results show the effect of the original orbital-scale climate trajectory (relative to a randomized climate trajectory) on hominin habitat suitability. Only grid points with $p < 0.05$ are shown in colors based on a paired t-test using $H(x,y,t)$ and $H_{scr}(x,y,t)$. **d**–**f**) Illustration of late-Holocene altitude-corrected downscaling of

net primary production: Left: Simulated NPP ($gC/m^2/year$) for original T31 atmosphere resolution, obtained from empirical NPP model using 1000-year late Holocene average of total precipitation and surface temperatures simulated by the CESM1.2 2Ma experiment; middle: same as left, but using altitude-corrections for temperature and precipitation as downscaling onto a 1x1 degree target grid, showing the emergence of key topographic features in Africa. This resolution was chosen in the calculations of the climate envelope model; right: for illustration, same as middle but for a 0.25x0.25 degree horizontal target grid. The qualitative gain in terms of regional details from T31 to 1x1 degree resolution outweighs the additional gain going from 1x1 degree resolution to a 0.25x0.25 degree grid.

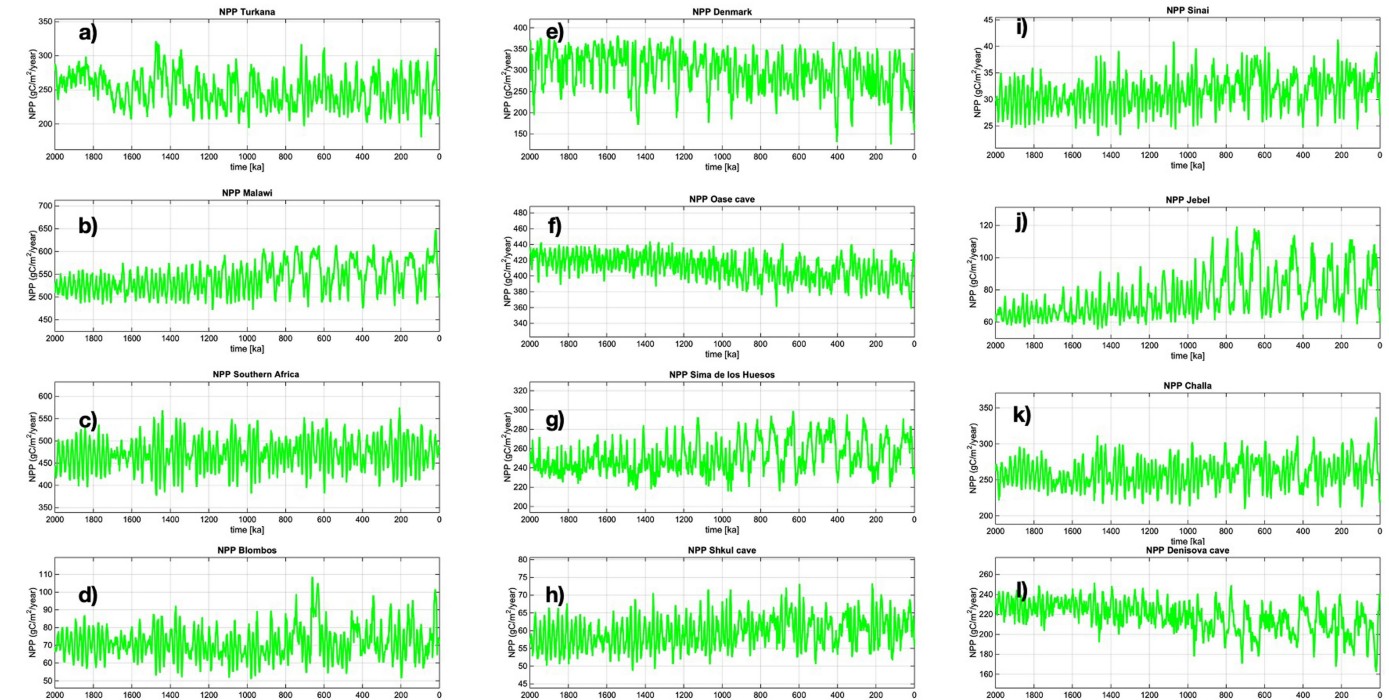

**Extended Data Fig. 9 | Net primary production near sites relevant for interpretation of hominin evolution.** from a-l: Lake Turkana (3°N, 36°E), Lake Malawi (12°S, 34°E), Southern Africa (25°S, 25°E), Blombos cave (34°S, 21°E), Denmark (56°N, 9°E), Peçtera cu Oase (45°N, 22°E), Sima de los Huesos (42°N, 5°W), Shkul cave (33°N, 35°E), Sinai (30°N, 34°E), Jebel Irhoud (34°N, 4°W), Lake Challa (3°S, 38°E), Denisova, cave (52°N, 84°E). The timeseries represents a spatial mean of 7°x7° degree around the center location.

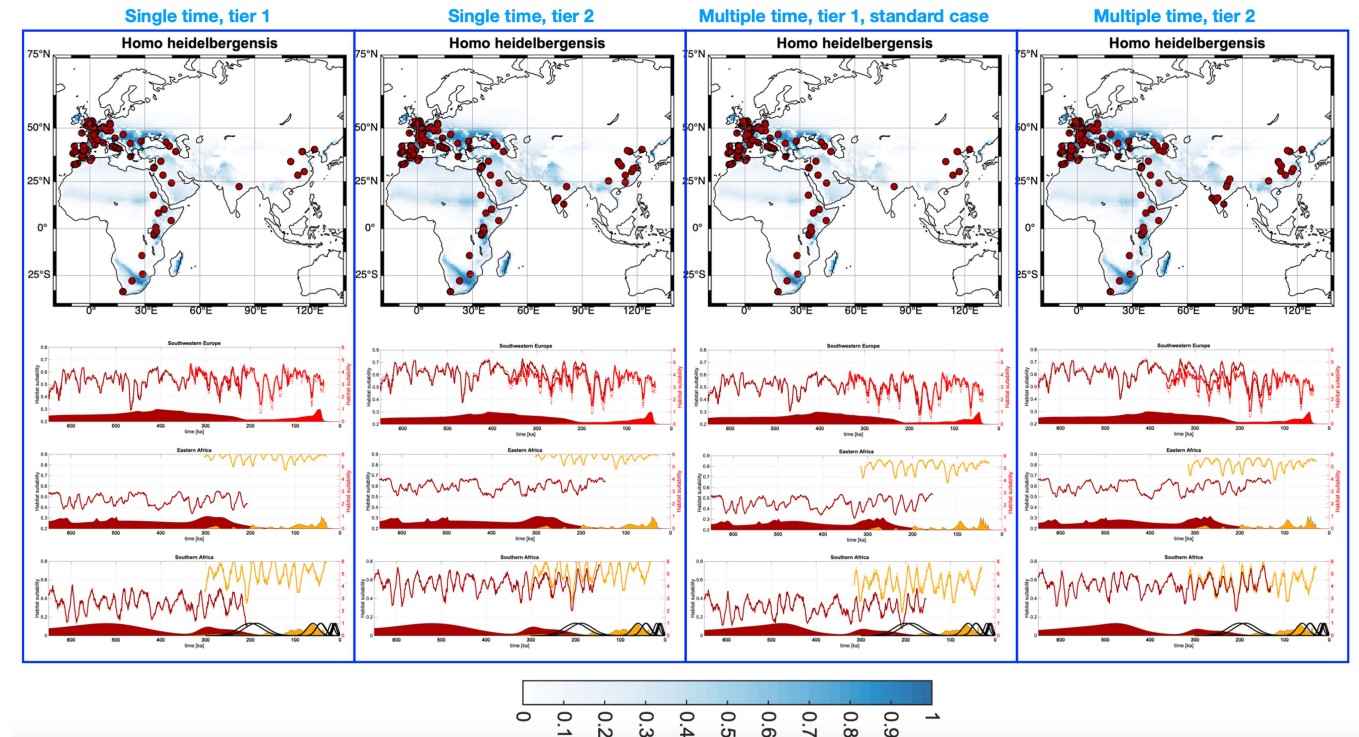

**Extended Data Fig. 10 | Robustness with respect to age and species assignment uncertainties.** Upper panels, same as for Fig. 1c, but for the 4 different cases single date-tier 1,2 and multi date-tier 1,2 (Methods). Multi-date tier 1 is the standard case used in our analysis. Lower panels, same as Fig. 3b–d, but calculated also for the 4 cases of assessing species and age uncertainties.

# Reporting Summary

## Statistics

For all statistical analyses, confirm that the following items are present in the figure legend, table legend, main text, or Methods section.

| n/a | Confirmed | |
|---|---|---|
| ☒ | ☐ | The exact sample size (*n*) for each experimental group/condition, given as a discrete number and unit of measurement |
| ☒ | ☐ | A statement on whether measurements were taken from distinct samples or whether the same sample was measured repeatedly |
| ☐ | ☒ | The statistical test(s) used AND whether they are one- or two-sided *Only common tests should be described solely by name; describe more complex techniques in the Methods section.* |
| ☒ | ☐ | A description of all covariates tested |
| ☒ | ☐ | A description of any assumptions or corrections, such as tests of normality and adjustment for multiple comparisons |
| ☒ | ☐ | A full description of the statistical parameters including central tendency (e.g. means) or other basic estimates (e.g. regression coefficient) AND variation (e.g. standard deviation) or associated estimates of uncertainty (e.g. confidence intervals) |
| ☐ | ☒ | For null hypothesis testing, the test statistic (e.g. *F*, *t*, *r*) with confidence intervals, effect sizes, degrees of freedom and *P* value noted *Give P values as exact values whenever suitable.* |
| ☒ | ☐ | For Bayesian analysis, information on the choice of priors and Markov chain Monte Carlo settings |
| ☒ | ☐ | For hierarchical and complex designs, identification of the appropriate level for tests and full reporting of outcomes |
| ☒ | ☐ | Estimates of effect sizes (e.g. Cohen's *d*, Pearson's *r*), indicating how they were calculated |

*Our web collection on statistics for biologists contains articles on many of the points above.*

## Software and code

Policy information about availability of computer code

| Data collection | The climate model simulations were conducted with the CESM1.2 model on the IBS/ICCP XC50-LC supercomputer Aleph, The computer code of the CESM1.2 model can be downloaded from https://www.cesm.ucar.edu/models/cesm1.2/. |
|---|---|
| Data analysis | The data analysis of the CESM1.2 simulations was conducted with the Climate Diagnostic Operators https://code.mpimet.mpg.de/projects/cdo/. The habitat suitability model was derived using our own computer codes which were developed for the software Matlab R2014b. Our codes will be shared with interested users on https://climatedata.ibs.re.kr. Maps in Fig. 1, Extended Data Figs. 4, 9, 10, 11 were generated in the matlab package m_map, Pawlowicz, R., 2020. "M_Map: A mapping package for MATLAB", version 1.4m, http://www.eoas.ubc.ca/~rich/map.html. The map in Fig. 4 was generated using the freely-available software Paraview https://www.paraview.org |

For manuscripts utilizing custom algorithms or software that are central to the research but not yet described in published literature, software must be made available to editors and reviewers. We strongly encourage code deposition in a community repository (e.g. GitHub). See the Nature Portfolio guidelines for submitting code & software for further information.

## Data

Policy information about availability of data

All manuscripts must include a data availability statement. This statement should provide the following information, where applicable:
- Accession codes, unique identifiers, or web links for publicly available datasets
- A description of any restrictions on data availability
- For clinical datasets or third party data, please ensure that the statement adheres to our policy

The CESM1.2 data and the calculated hominin habitat suitability data will be shared on the climate data server https://climatedata.ibs.re.kr. The database of hominin remains, and artefacts generated and used here is provided as Supplementary Table 1. There are no restriction on data usage and access.

# Field-specific reporting

Please select the one below that is the best fit for your research. If you are not sure, read the appropriate sections before making your selection.

☐ Life sciences ☐ Behavioural & social sciences ☒ Ecological, evolutionary & environmental sciences

For a reference copy of the document with all sections, see nature.com/documents/nr-reporting-summary-flat.pdf

# Ecological, evolutionary & environmental sciences study design

All studies must disclose on these points even when the disclosure is negative.

| | |
|---|---|
| Study description | Using the Mahalanobis distance approach, a new climate computer model simulation covering the past 2 million years is combined with an extensive hominin fossil and archaeological data compilation (based on previously published sources) to derive a new climate envelope model. The climate envelope model is forced with the climate model input to estimate the time evolution of habitat suitability for 6 hominin species on a spatial mesh of 1 degree x 1 degree and covering the last 2 million years. |
| Research sample | The fossil and archaeological data compilation used in this study is based on Raia, P., Mondanaro, A., Melchionna, M., Di Febbraro, M. & Diniz-Filho, J. A. F. Past Extinctions of Homo Species coincided with increased vulnerability to Climatic Change. One Earth 3, 1-11 (2020) and extended and updated with new published data sources. Our dataset includes 3246 entries for 6 hominin species during the past 2 million years, each characterized by their latitude, longitude, age and age uncertainties. All data points have been compiled from previously published studies. The climate model used in this study is the Community Earth System Model, version 1.2 in in 3.75×3.75 degree horizontal resolution. Data from this climate model are further downscaled to a 1x1 degree horizontal grid to account for orographic effects. The 6 hominin species (Homo sapiens, Homo neanderthalensis, Homo heidelbergensis, Homo erectus, Homo ergaster, Homo habilis) were chosen to capture key aspects in the evolution of the genus homo between 2 million years ago to 30,000 years ago. The CESM1.2 climate model was chosen for its numerical efficiency (300 simulation years per day on Cray XC50) and its well-documented realistic presentation of the climate system under present-day conditions. Our study focuses on the past 2 million years, due to the availability of the computer model simulation data. |
| Sampling strategy | The fossil and archaeological data were selected from the vast anthropological and archaeological literature following simple criteria: data samples must be attributable to 1 or 2 (out of the 6) species, age estimates and respective uncertainties must be provided in the original reference, age of samples must cover the time period from 2 million years ago to 30,000 years ago. Based on these criteria our extensive literature search yielded 3246 samples. |
| Data collection | The novel CESM1.2 computer model simulation for the last 2 million years was conducted by Dr. Kyungsook Yun (co-author). |
| Timing and spatial scale | The climate model simulation covers the period from 2 million years ago to early Holocene. We therefore concentrate our data analysis on the period 2 Ma - 0.03 Ma. The climate model simulation does not have any gaps. The age gaps for the fossil/archaeological samples are provided in Supplementary Data 1. Data samples were selected only for Africa and Eurasia, because Australia and the Americas are not a subject of our paper. Temporal resolution: The model simulation output was sub-sampled temporally for the analysis using monthly stratified 1000-year means. This approach is justified, because our study focuses on orbital-scale changes in climate with a minimum timescale of 21,000 years (precessional cycle). Spatial scale: 3.75×3.75 degree horizontal resolution output from the climate model is further downscaled to a 1x1 degree horizontal grid to account for orographic effects and allow for a better comparison with the hominin dataset. |
| Data exclusions | The entire climate model simulation covering the orbital history of the past 2 million years is included. Our hominin database only includes fossil and archaeological sites that have good chronological constraints. Undated specimen are excluded in our analysis. |
| Reproducibility | Our study is not based on experimental data, but on numerical simulations, which are highly reproducible within the computational accuracy on other computing platforms. |
| Randomization | To capture the effect of fossil and archaeological age uncertainties, the climate envelope model links fossil datapoints to climate model data with 100 different ages obtained from a random distribution that spans the most likely age of the hominin data and their corresponding age uncertainties. The climate niche model therefore accounts for the age uncertainties of the hominin record. To further test the effect of climate mean state, variability and exact trajectory on the climate envelope model, we have applied 2 randomization methods: a) 4 dimensional climate data vector was scrambled randomly in time while keeping the 4-dimensional climate covariance intact; b) 4-dimensional climate data vector components are scrambled randomly in time and independently from each other, which destroys the climate co-variance structure, but maintains the average climate state. |
| Blinding | Blinding is not applicable to our numerical model simulations, because the same computer code will generate the same simulation on other computing platforms, which use the same numerical precision. |

Did the study involve field work? ☐ Yes ☒ No

# Reporting for specific materials, systems and methods

We require information from authors about some types of materials, experimental systems and methods used in many studies. Here, indicate whether each material, system or method listed is relevant to your study. If you are not sure if a list item applies to your research, read the appropriate section before selecting a response.

## Materials & experimental systems

| n/a | Involved in the study |
|-----|----------------------|
| ☒ | Antibodies |
| ☒ | Eukaryotic cell lines |
| ☒ | Palaeontology and archaeology |
| ☒ | Animals and other organisms |
| ☒ | Human research participants |
| ☒ | Clinical data |
| ☒ | Dual use research of concern |

## Methods

| n/a | Involved in the study |
|-----|----------------------|
| ☒ | ChIP-seq |
| ☒ | Flow cytometry |
| ☒ | MRI-based neuroimaging |

