## [Peer Review File · Nature]

Manuscript Title: Climate effects on archaic human habitats and species successions

Reviewer Comments & Author Rebuttals

Reviewer Reports on the Initial Version:

Referees' comments:

Referee #1 (Remarks to the Author):

This paper is an attempt to examine climate shifts over the last 2 Ma and the role it played in hominin origins and dispersal. The paper uses a model simulation to examine a number of factors of relevance to environmental changes, paying attention to hominin records across Africa and Eurasia. This is a most interesting approach and I can see the merit of integrating a large amount of interdisciplinary data in this format, in order to test hypotheses and to offer new evolutionary insights.

This paper references and uses a range of climate models and simulations - CGCM, CESM, CEM, etc, that are beyond my technical capability to evaluate, and they really require expertise to examine their details. I am more familiar with the palaeoanthropological and archaeological literature, and this review is undertaken on this basis.

In order to examine the palaeoanthropological and archaeological record, the authors rely on "an extensive fossil and archaeological database", based on Reference 18, Raia et al. 2020. Raia et al. is published in the new journal, "One Earth". Given that no information is provided in the Timmermann et al. paper, I had to go to the One Earth publication. This was unfortunately very frustrating, as the paper was behind a paywall, and our institution does not subscribe to this journal. I tried several ways to access the Raia et al. paper, but I failed in my attempts. For me, this was not only frustrating, but I could not access the "extensive" database that Timmerman et al. rely on for this paper. As a reviewer, this makes it near impossible for me to evaluate the veracity of the database.

In examining the Timmermann et al. paper, I have to say I am a bit concerned about the nature of the Raia et al. database. Despite the fact that they say it is "extensive" I find the database wanting and somewhat unclear, including on the basis of the figures supplied in the Timmermann et al. paper. The authors refer to their Figure 1a-f to illustrate what I take to be all finds in the database. Yet, when I examined Fig. 1, I had all sorts of questions without the ability to cross-check the figures. The first that came to my mind was: are these really all the fossil and archaeological sites over the last 2 Ma? I was also immediately struck that this may not be comprehensively "archaeological". Is this an terminology problem with Raia et al. and do they mostly mean fossils? I have no way to evaluate this. In a closer look, I have a large number of questions. For example, why are no early African sites depicted in large parts of the continent, such as West and North Africa? For the H.

erectus maps, I am wondering about the locations - I have no idea, for example, how many sites in the Indian subcontinent are designated *H. erectus* (there are no *H. erectus* fossils in India). The *H. sapiens* distribution inset is almost impossible to see, as yellow is used on yellow, and I do not know the ages of the sites, such as the ones being used in Eastern Asia (early or late, or both?).

Taking a turn to Fig. 4, this is said to be a distribution of "southern" and "northern" *H. heidelbergensis* populations (which is unconventional, but stated as a proposal). I am not really clear on this figure and the main text does not help. This figure looks to be a rather simplistic rendition of "directions" of movement and "succession". The directionality is rather basic and problematic given the time depths involved. On succession here and in the text, I am somewhat surprised by the authors citation of *H. sapiens* as in Southern Africa at 300 ka. They seem to be unaware of key literature in the last few years on African multi-regionalism, which is not evaluated or cited. See, for example, Scerri et al. in *TRENDS*, 2018, which is fast becoming a dominant model and heavily cited, with follow up literature.

In ED Fig. 7, these problems are further demonstrated. Here for the first time the authors use "artefacts" instead of archaeological, so there must be sites in the Raia et al. database. This makes me wonder if archaeological sites are then related to species. I also have concerns about the accuracy of these "site" distributions over the last 2 Ma and I would like to see additional back up information. In addition, the figure is really hard to see with the small size and the colours used in the symbols and background colours, so this really should be improved.

As the authors can see from my commentary, so much hinges on the nature of the database. But without one, I can not say much more. I would strongly urge that a database be supplied in the supplementary information. I would also urge the authors to conduct their own evaluation of the integrity of the database (site and locality spatial distributions, type of site, dating methods and issues, etc), including commentary, supplementation perhaps, and tabled evaluation data. If the authors can provide more confidence with include a hominin fossil / archaeological site database, I for one would be much happier evaluating the points presented in the Results and Discussion and Synthesis sections.

Referee #2 (Remarks to the Author):

The Timmermann et al. manuscript uses a paleo-climate model run for the last 2 million years to simulate changes in the regional environment in which early humans evolved. They compare their results to a compilation of both the fossil and archaeological records. They are testing the hypothesis that changes in astronomic forcing significantly altered the environments of Africa causing the evolution, extinction and migrations of hominins. The model output seems to suggest that temperature, rainfall and vegetation (in terms of productivity) did in fact have an impact on human evolution.

This study builds on earlier work by this group and others. First the rich literature using proxy data to look at local and regional climate variability linking it to orbital variations – particularly eccentricity modulated precession. Second modelling work starting with the impacts of East African uplift on the

climate and vegetation of East Africa in the 1990s and worth of mention is the excellent modelling work on the environment associated with Neanderthals by the authors of this current study in QSR.

We could all go into great detail and suggest different ways of modelling, different models to use, different scaling, but I am happy that the authors have chosen an appropriate set of models and applied them in the very best way they can. Because modelling is just away to test hypothesis in a temporal-spatial way that is not possible with proxy data. There is, however, one issue which is difficult to deal with. The model outputs are downscaled to $1^{\circ} \times 1^{\circ} = 110 \text{ km}$ which is extremely good. But we know that there are huge variations in East Africa over the very short distances due to the mountainous topography. In Ethiopia you can go from cloud rainforest to arid desert in 50 km. East Africa is also dominated by lakes which are threshold mechanisms – if rainfall increases by a certain amount suddenly they fill up. There is also not a one to one relationship between rainfall, temperature and what vegetation exists. This is because of bifurcations in the climate-vegetation relation due to moisture feedback created by vegetation, particularly forests. No idea how you get round this micro/niche environment problem.

I do like that the authors have used randomised climate outputs to test their conclusions. This adds depth and makes it clearer their key findings.

I am also interested in the author's observation that 'largest density of hominin fossils is found in high NPP regions $>200 \text{ gC/m}^2/\text{year}$, with the exception of northeastern'. I think this needs unpacking and I am not sure I agree with their observation. First average NPP over 2 million years seems to go against the argument of the paper that variability or changing climate is important. Second hominins live in very specific places in East Africa – by very productive lakes so the mean may not capture this mean. Third there is also an upper limit – as early humans do not like highly productive regions such as, dense forest see Extended Data Fig. 7. What is interesting is that current fossil data still suggests all the early hominin species evolved in East Africa – the exception to the rule the authors have suggested.

I am not convinced by the *Homo heidelbergensis* discussion and suggest a comparison with recently published pan-African data may help here – by Kaboth-Bahr et al. (2021) in PNAS.

I also do not really understand the reasoning that early species were limited by temporal changes in climate while later species were spatially limited by climate change. The fundamental control on evolution is 'does an individual reproduce successfully or not' – do those traits get passed on to the next generation and provide some advantage or not. Hence why I am not sure how the temporal and spatial can be separated but do understand this may be due to the condensed writing style required by Nature.

Overall this is an interesting and stimulating piece of science. The authors try to test the hypothesis posed by many colleagues that orbital forcing changed the local/regional environment of Africa which created opportunities for evolution and extinction of hominin species. The modelling is excellent as is the comparison with the fossil record. The issues are with model resolution – but this is true of all model based studies. There are issues with many assumptions being made in the modelling all of which sound reasonable but not create the realistic output wanted (the

compounded error/assumption idea). I have struggled with the interpretation of the results which seems to jump beyond the model results and tries to explain what caused the evolution of key species.

Referee #3 (Remarks to the Author):

Review of 2021-07-10738 By Chris Brierley, UCL

This paper represents a substantial body of work. The ability to create suitability maps varying in time for 5 different hominin species appears a wonderful result. My expertise doesn't cover all the disciplines included in this manuscript. So I can't comment on whether this is a substantial advance on existing work, but the depth of the climate modelling contribution to it certainly is.

To my knowledge, no one has yet undertaken a transient simulation with the model a sophisticated CCSM3 for anything like this length of simulation. I would be very interested to see they evolution of the climate within these simulations, but I accept it isn't the focus of this particular manuscript and I hope it will come out in another paper later. The creation of a habitability metric seems to be have been done using a suitable and appropriate method – although I confess I am not an expert in this kind of model.

Items to be addressed/clarified:

I was surprised that there was no justification of the choice of the 4 input variables to this model. The CESM run allows for many interesting variables to be investigated (for example, relative strength of a 1 in 10 year drought on soil moisture). While I don't think that all such options should be investigated, the reasoning behind picking the 4 that are used should be explained.

The artificial inflating of the CO₂ forcing by 1.5x in the GCM simulation seems unjustified. The climate sensitivity of CCSM4 at T31 resolution is 2.9oC according to Shields et al (2012), so CESM1.2 should be similar. This is pretty-much exactly the central estimate according the new IPCC AR6 report. In fact, the value of 4.5oC reached by Friedrich and Timmerman (2020) is even outside the likely range from the IPCC. Clearly, there could be some non-linearity in the Pleistocene value due to the ice-sheets, but that sits more happily into a earth system sensitivity framework. There needs to be a much better justification of this, as otherwise the CESM simulation would artificially inflates the (greenhouse gas forced) climate variations seen over past 2 Ma. I note that the timeseries in EDFig 2 suggest otherwise, but how can you discount these biases as arising from downscaling (something you argue is vital) or deficiencies in the ice-sheet forcings.

It is a shame that the CESM1.2 simulation did not deploy either interactive vegetation, which would have avoided the need for the NPP approach. Isotopes would have been really useful for subsequent studies too.

I am confused by the multiple downscaling steps (as shown in ExtD Fig. 5). This figure shows that there are 3 interim steps to go from the CLIMBER simulation to the species distribution. As I

understand it, the CESM1.2 acts as an initial dynamical downscaling, which is then followed by additional statistical downscaling and an empirical NPP model. This raises some questions:

1. What is the added benefit of the CESM downscaling step? It comes at considerable computational cost, but are the conclusions the same as just doing a statistical downscaling from the CLIMBER variables?
2. Why is 1° resolution sufficient, but T31 isn't? Does this relate to the spatial range roamed by hominins present at the fossil sites? Or are the topographic differences more important – but I'd guess that the altitudinal differences between the individual sites and ETOPO1 are greater than the altitudinal differences between ETOPO1 and the T31 model. I don't see any discussion of these points in the paper.
3. Does the empirical NPP model really do a better job at capturing the temporal changes in NPP simulated by CESM1.2? I understand the argument about the prescribed PFTs, but why not just look at the NPP for the individual PFTs and if necessary recombine them. This NPP step feels really crude.

Author Rebuttals to Initial Comments:

Referees' comments:

Our answers are provided in blue.

Referee #1 (Remarks to the Author):

This paper is an attempt to examine climate shifts over the last 2 Ma and the role it played in hominin origins and dispersal. The paper uses a model simulation to examine a number of factors of relevance to environmental changes, paying attention to hominin records across Africa and Eurasia. This is a most interesting approach and I can see the merit of integrating a large amount of interdisciplinary data in this format, in order to test hypotheses and to offer new evolutionary insights. This paper references and uses a range of climate models and simulations - CGCM, CESM, CEM, etc, that are beyond my technical capability to evaluate, and they really require expertise to examine their details. I am more familiar with the palaeoanthropological and archaeological literature, and this review is undertaken on this basis.

We have further explained our modeling approach in the revised manuscript to clarify some details. The corresponding section now reads:

“To quantify the relationship between climate and the presence of archaic hominin species, we build a spatial climate envelope model (CEM) (Methods, Extended Data Fig. 3). The CEM uses a previously generated species database^{21,22} (SDB) comprised of geo-chronologically constrained hominin fossils and archaeological layers containing lithic industries (Fig. 1a-f) with topographically-downscaled (1°x1° grid) 1000-year averaged data of climate variables from 2Ma which are relevant for human survival (annual mean precipitation, temperature, yearly minimum precipitation and net primary productivity, see Methods). The data entries of the SDB contain information about location, age, age uncertainty and hypothesized species, selected from early African Homo (combining Homo habilis and Homo ergaster as one group), Eurasian Homo erectus, Homo heidelbergensis, Homo neanderthalensis, and Homo sapiens. Spatio-temporal temperature and precipitation fields of the 2Ma climate model simulation (e.g. Extended Data Figs. 4,5,6 for select locations) are extracted for the species-presence locations and ages in the SDB and then statistically aggregated as a CEM. Subsequently, using the Mahalanobis metric^{23,24} and the spatio-temporal climate evolution in 2Ma we derive a habitat suitability model (HSM, Methods) for each species, which quantifies the probability of finding fossil and/or archaeological evidence of the species at a given time and geographic location.”

Moreover, the slightly revised Methods section should provide additional information:

In order to examine the palaeoanthropological and archaeological record, the authors rely on "an extensive fossil and archaeological database", based on Reference 18, Raia et al. 2020. Raia et al. is published in the new journal, "One Earth". Given that no information is provided in the Timmermann et al. paper, I had to go to the One Earth publication. This was unfortunately very frustrating, as the paper was behind a paywall, and our institution does not subscribe to this journal. I tried several ways to access the Raia et al. paper, but I failed in my attempts. For me, this was not only frustrating, but I could not access the "extensive" database that Timmerman et al. rely on for this paper. As a reviewer, this makes it near impossible for me to evaluate the veracity of the database.

We understand the frustration of the reviewer and apologize for the lack of access to the original database and the paywall issue of the original Raia et al 2020 reference.

We are using now the slightly updated data base presented in Mondanaro et al. (2020) which can be downloaded freely from:

<https://ars.els-cdn.com/content/image/1-s2.0-S2589004220308853-mmc2.xlsx>

We also provide the original data (Raia et al. 2020, OneEarth) and the corresponding metadata to the reviewer as part of this submission to facilitate the review. In our data-sharing statement we now refer to the Mondanaro paper and the open access link mentioned above. The revised statement now reads “Data availability: The CESM1.2 data and the calculated hominin habitat suitability data will be shared on the climate data server <https://climatedata.ibs.re.kr>. The database of hominin remains and artefacts used here can be accessed through <https://ars.els-cdn.com/content/image/1-s2.0-S2589004220308853-mmc2.xlsx>”

In addition, we revised our manuscript as follows:

*“To quantify the relationship between climate and the presence of archaic hominin species, we build a spatio-temporal climate envelope model (CEM) (Methods, Extended Data Fig. 3). Our CEM uses topographically-downscaled ($1^{\circ} \times 1^{\circ}$ grid) 1000-year averaged data of climate variables from 2Ma which are relevant for human survival. Here we choose annual mean precipitation, temperature, yearly minimum precipitation and net primary productivity (Extended Data Figs. 4, 5, 6 for select locations), in combination with a previously generated database comprised of geochronologically constrained hominin fossils and archaeological layers containing lithic industries (Fig. 1a-f). The extensive database used here stratifies the data entries into spatiotemporally delimited groups which have been assigned to early African hominins (combining *Homo habilis* and *Homo ergaster* as one group), Eurasian *Homo erectus*, *Homo heidelbergensis*, *Homo neanderthalensis* or *Homo sapiens*. The CEM is based on a 4-dimensional climate envelope Mahalanobis metric^{20,21}. “*

In examining the Timmermann et al. paper, I have to say I am a bit concerned about the nature of the Raia et al. database. Despite the fact that they say it is "extensive" I find the database wanting and somewhat unclear, including on the basis of the figures supplied in the Timmermann et al. paper. The authors refer to their Figure 1a-f to illustrate what I take to be all finds in the database. Yet, when I examined Fig. 1, I had all sorts of questions without the ability to cross-check the figures. The first that came to my mind was: are these really all the fossil and archaeological sites over the last 2 Ma?

As the reviewer can now check in full using the data and metadata provided as part of this submission and the open access link given above, the 2572 database entries form, to our knowledge, the largest human paleontological and archaeological database ever assembled, which uses radiometric dates. Archeological sites with very uncertain age constraints have not been used. We update the database continuously as new literature gets published. We focus here only on the period from 2 Ma to 30 ka, thereby leaving out the more recent data from *Homo sapiens* covering the past 30 ka. In fact, Fig. 1 shows all the fossil and archaeological sites in the database, but some sites are very close to each other, that they may appear as single points. On the global maps, this gives the false (but unavoidable) impression of fewer datapoints, but if we zoom into specific regions, we see a much richer structure – see illustration below for Neanderthals, where we zoom into the area between Levant, Black Sea and Caspian Sea.

I was also immediately struck that this may not be comprehensively "archaeological". Is this an terminology problem with Raia et al. and do they mostly mean fossils? I have no way to evaluate this.

This is now detailed in the files, we submitted along with our revised manuscript to facilitate the review. The database contains a mixture archaeological artefacts and fossils, both of which assigned to species, wherever identification is plausible.

In a closer look, I have a large number of questions. For example, why are no early African sites depicted in large parts of the continent, such as West and North Africa?

Possible sites with evidence for early *Homo* presence in North Africa are Ain Boucherit Stratum AB and Ain Hanech. A possible additional occurrence is at Thomas Quarry I (Level L). Of these, the lower levels of Ain Boucherit AB are older than our 2 million years age limit considered in our study, the other is missing in the OneEarth/iScience papers. Tighenif is present in our database but classified under the *Homo heidelbergensis* attribution (i.e. not properly 'early'), the same is true for Kharga Oasis, Cap Chatelier, Sidi Abderrahmane and Salé. All of these African sites are in our compilation, but they were not assigned to early *Homo sapiens*, but rather to *Homo heidelbergensis*.

The evidence for early humans presence in West Africa is scarce. Scerri et al. 2017 documented the presence of Middle Stone Age technology in Lower Senegal Valley starting from 25 ka. Other, contemporary lithic assemblages from the Falémé Valley at the site of Fatandi V are Later Stone Age (LSA) which includes microliths and segments, and points (Lebrun et al., 2016). In Cameroon, minimum radiocarbon ages of 15.3 ka were reported in association with centripetal preferential Levallois artefacts (MacDonald, 1997). Our data focus on the >30 ka period and these samples were not included in our compilation.

Else-where in both the arid Sahelian and forested regions of West Africa, the few dated sites such as Ounjougou, Birimi and Bilma document the latest MSA at between ~40 and 30 ka. In a very recent article, Douze 2021 traced back the origin of MSA to MIS5. In regard to the Lower Paleolithic, Agai 2021 reported the discovery of Acheulean tools in Nigeria comparable or very similar to human tools found in East Africa but the author did not provide any radiometric data, which is the reason this record does not make for our database.

Agai, J. M. (2021). A Report on the History of the Acheulean Industry of Mai Idon Toro in the Central-Region of Nigeria. *Culture and History*, 1(1), p29-p29.

Bailey, S. E., Benazzi, S., Souday, C., Astorino, C., Paul, K., & Hublin, J. J. (2014). Taxonomic differences in deciduous upper second molar crown outlines of *Homo sapiens*, *Homo neanderthalensis* and *Homo erectus*. *Journal of human evolution*, 72, 1-9.

Douze, K., Lespez, L., Rasse, M., Tribolo, C., Garnier, A., Lebrun, B., ... & Huysecom, E. (2021). A West African Middle Stone Age site dated to the beginning of MIS 5: Archaeology, chronology, and paleoenvironment of the Ravin Blanc I (eastern Senegal). *Journal of Human Evolution*, 154, 102952.

Lebrun, B., Tribolo, C., Chevrier, B., Rasse, M., Lespez, L., Leplongeon, A., ... & Huysecom, É. (2016). Establishing a West African chrono-cultural framework: First luminescence dating of sedimentary formations from the Falémé Valley, Eastern Senegal. *Journal of Archaeological Science: Reports*, 7, 379-388.

MacDonald, K. C. (1997). *Western African and Southern Saharan Advanced Foragers*. Walnut Creek: Altamira (Sage).

Raynal, J. P., Alaoui, F. Z. S., Magoga, L., Mohib, A., & Zouak, M. (2002). Casablanca and the earliest occupation of North Atlantic Morocco. *Paléorivages de Casablanca. Stratigraphie et Préhistoire ancienne au Maroc atlantique*, 13(1), 65-77.

Sahnouni, M., Parés, J. M., Duval, M., Cáceres, I., Harichane, Z., Van der Made, J., ... & Semaw, S. (2018). 1.9-million- and 2.4-million-year-old artifacts and stone tool-cutmarked bones from Ain Boucherit, Algeria. *Science*, 362(6420), 1297-1301.

Scerri, E. M., Blinkhorn, J., Niang, K., Bateman, M. D., & Groucutt, H. S. (2017). Persistence of Middle Stone Age technology to the Pleistocene/Holocene transition supports a complex hominin evolutionary scenario in West Africa. *Journal of Archaeological Science: Reports*, 11, 639-646.

Smith, J. R., Giegengack, R., Schwarcz, H. P., McDonald, M. M., Kleindienst, M. R., Hawkins, A. L., & Churcher, C. S. (2004). A reconstruction of Quaternary pluvial environments and human occupations using stratigraphy and geochronology of fossil-spring tufas, Kharga Oasis, Egypt. *Geoarchaeology: An International Journal*, 19(5), 407-439.

For the *H. erectus* maps, I am wondering about the locations - I have no idea, for example, how many sites in the Indian subcontinent are designated *H. erectus* (there are no *H. erectus* fossils in India).

Yes, of course, we agree that there are no *H. erectus* fossils in India. Our database includes possible *Homo erectus* archaeological sites are Attirampakkam, Sadab, Teggihalli II, based on archaeological interpretations. Pappu and Akhilesh (JHE 2019) discuss the Acheulian findings of Attirampakkam in the context of what they define as *Homo erectus* sensu lato. Based on biochronological and archaeological perspectives *Homo erectus* attribution might be a plausible candidate. Please see below the corresponding references, which were discussed in our previous papers, but are not part of the primary references of our study here.

Akhilesh, K., Pappu, S., Rajapara, H. M., Gunnell, Y., Shukla, A. D., & Singhvi, A. K. (2018). Early Middle Palaeolithic culture in India around 385–172 ka reframes Out of Africa models. *Nature*, 554(7690), 97.

Chauhan, P. R. (2010). Comment on 'Lower and Early Middle Pleistocene Acheulian in the Indian sub-continent' by Gaillard et al.(2009)(*Quaternary International*). *Quaternary International*, 223, 248-259.

Pappu, S., & Akhilesh, K. (2019). Tools, trails and time: debating Acheulian group size at Attirampakkam, India. *Journal of human evolution*, 130, 109-125.

Szabo, B. J., McKinney, C., Dalbey, T. S., & Paddayya, K. (1990). On the age of the Acheulian culture of the Hunsgi-Baichbal Valleys, Peninsular India. *Bulletin of the Deccan College Research Institute*, 50, 317-321.

The *H. sapiens* distribution inset is almost impossible to see, as yellow is used on yellow, and I do not know the ages of the sites, such as the ones being used in Eastern Asia (early or late, or both?).

Yes, we agree. This was not a good choice of colors. We have now updated the color scheme of Fig. 1 (see below). The median age of the sites is indicated by the size of the circles. Moreover, a detailed age estimate with uncertainty ranges is provided in <https://ars.els-cdn.com/content/image/1-s2.0-S2589004220308853-mmc2.xlsx>, as mentioned in the Data availability statement.

Taking a turn to Fig. 4, this is said to be a distribution of "southern" and "northern" *H. heidelbergensis* populations (which is unconventional but stated as a proposal). I am not really clear on this figure and the main text does not help. This figure looks to be a rather simplistic rendition of "directions" of movement and "succession". The directionality is rather basic and problematic given the time depths involved.

The point is well taken. We have therefore modified the figure, omitting now the directions and instead of schematic potential migration corridors, we draw the actual computed time-mean habitat suitability (in blue contours), from our climate envelope model. The revised figure is now much

more informative, because the habitat suitability already nicely illustrates the potential migration corridors, with some areas being disconnected on average. However, in these intermittently disconnected areas there are occasional corridors that can open, when the precessional and eccentricity forcing are in the right phase to create green and habitable corridors (see orange timeseries inlays on right side). This information is further used in our updated discussion section as

“...we suggest the following scenario (Fig. 4): about 850-600 ka, *Homo heidelbergensis*, which may have originated from *Homo ergaster* in eastern Africa (Extended Data Fig. 8e), eventually split into a southern African branch, and a northern branch which included Eurasian populations. The southern branch experienced considerable climatic stress in southern Africa during Marine Isotope Stages (MIS) 11 and 9, which may have accelerated either a gradual or cladogenetic transition into *Homo sapiens*³⁵. The northern branch may have further bifurcated around 430 ka, possibly giving rise to Denisovans, which populated parts of central and eastern Asia. Inside central Europe, *Homo heidelbergensis*, which experienced strong local climatic stress due to the eccentricity modulated ice-age cycles (Fig. 3b), gradually evolved into Neanderthals between 350-300 ka. Side branches to northwestern Africa, back propagation, multiple dispersals³⁶, interbreeding³⁷ and subsequent speciation³⁸ may have further complicated the picture.”

On succession here and in the text, I am somewhat surprised by the authors citation of *H. sapiens* as in Southern Africa at 300 ka.

More to the point, Kathu Pan 1 layer 3 and Florisbad are both close to 300 ka (including age uncertainty), and both in South Africa. Please see:

Grün, R., Brink, J. S., Spooner, N. A., Taylor, L., Stringer, C. B., Franciscus, R. G., & Murray, A. S. (1996). Direct dating of Florisbad hominid. *Nature*, 382(6591), 500.

Porat, N., Chazan, M., Grün, R., Aubert, M., Eisenmann, V., & Horwitz, L. K. (2010). New radiometric ages for the Fauresmith industry from Kathu Pan, southern Africa: Implications for the Earlier to Middle Stone Age transition. *Journal of Archaeological Science*, 37(2), 269-283

We are aware that several choices in our compilation could not meet the opinions of everyone, exactly because there is no consensus in the literature on several taxonomic issues. This is almost unavoidable with any large compilation, such as ours, which includes 2572 datapoints and species assignments, relying on the vast archaeological and anthropological literature. Yet, our tests by leaving out several sites, show robust patterns of habitat suitability that we report, which depends only little on the inclusion or exclusion of a few individual sites or the classifications. This appears to be the great advantage of our large-scale approach which projects the majority of archaeological/fossil sites for each species onto their own climate envelope – an advantage of the Mahalanobis distance which assigns cumulative probability values to the dataset, using a data-based co-variance metric. Individual fossils e.g. Jebel Irhoud with ambivalent classifications of either *H. heidelbergensis* (adopted here), “archaic” *Homo sapiens*, or something else do not carry much weight for the habitat value calculation (which uses e.g. 269 data points for *Homo heidelbergensis* and 796 sites for *Homo sapiens*) and the results in Fig. 2,3.

They seem to be unaware of key literature in the last few years on African multi-regionalism, which is not evaluated or cited. See, for example, Scerri et al. in TRENDS, 2018, which is fast becoming a dominant model and heavily cited, with follow up literature.

We are very much aware of the Scerri paper and the African multi-regionalism discussion, as one of us is a co-author and provided input into their climatic discussion.

In fact, our paper is consistent with a general multi-regional point of view. In key time intervals, we can see the regional distribution of different hominin species and also the regionally distributed patchwork of habitat suitability among members of one species (which in fact does not necessarily imply multi-regional *origins*). The overlap and potential interaction between species is a key point in our study and it is quantified in Fig. 1, 2, 3, Extended Data Fig. 8, 9. We argue that this spatio-temporal habitat overlap, which previously was difficult to see with only few archaeological and fossil sites, provides key information on potential species interactions and successions.

In fact, the climate envelope model-generated habitat suitability data we provide in our study for the first time could become extremely useful for testing more specific aspects of a multi-regional/multi-species framework, rather than a broad-stroke no-single origin statement. We show that by combining climate data with archaeological/fossil sites, we can gain a much deeper insight into where different groups might have lived at specific times, which allows us to identify potential origins in a multi-regional and multi-species environment more quantitatively. This spatio-temporal structure, revealed here for the first time, gives a more refined view on the potential “when-and-where” of different groups, than just looking at the relatively sparse individual sites in an age context. We think that our study therefore extends the Scerri et al. (2017) study, which focuses only on *Homo sapiens* and the its potential predecessors, by providing a quantitative framework to look for interactions and potential successions between multiple species, including *Homo heidelbergensis*, *Homo neanderthalensis*, *Homo sapiens*. We further include a brief discussion on a potential Denisova split from *Homo heidelbergensis*, which may have occurred in the middle East.

Irrespective of the multi-regionalism debate, it is still valid to ask the key question, who the likely common ancestors were of *Homo sapiens* and *H. neanderthalensis* and Denisovans.

Studying only on 5 species (all existing well-dated sites of fossils and archaeological artefacts in our database are assigned to one of these clades), we focus here on the most likely candidate *Homo heidelbergensis*, which itself likely emerged from *Homo ergaster*, sometimes referred to as African *Homo erectus*. Previous discussions of this fundamental question on the origin of Late Pleistocene species were based either on genetic data and fossil evidence and the provenance of different industries. By infusing well-validated climate information into this discussion, we can get a more refined and quantitative view. Based on this analysis we conclude that *Homo heidelbergensis* (as defined by their fossil/climate-derived climate niche) lived at the right time at the right place and shared similar habitats to explain some of the earliest emergence of Neanderthals (400-300 ka Europe) and *Homo sapiens* (300-200 ka in Southern Africa). This is a new perspective that we are sharing now in Fig. 3 and our updated schematic Fig.4

We are now citing the Scerri et al paper and mention the issue of multi-regionalism, following the reviewer's helpful suggestion. Unfortunately, we do not have enough space to embark on a detailed discussion of multi-regionalism versus multi-regional origins ideas. Moreover, we also wanted to emphasize that our extensive CEM dataset (4 Gigabytes) is available for further analysis by the community and more research could go into understanding the connectivity of refugia in our CEM data.

The revised paragraph now reads

"Our extensive hominin database and the CEM capture regionally distributed patchworks of habitable areas in agreement with a general multi-regional perspective (Scerri et al. 2017) (Fig. 1). According to our CEM, southern and eastern Africa, as well as the region north of the Intertropical Convergence zone emerge as potential long-term refugia for archaic humans. As climate changes on orbital timescales, these refugia shift geographically, creating more complex spatial population structures. Further analysis of the pan-African refugia connectivity in our CEM dataset (e.g. inlay in Fig. 4, see data availability statement) will increase our understanding of hominin dispersal, interbreeding and cladogenic transitions as well as potential cultural exchanges."

In ED Fig. 7, these problems are further demonstrated. Here for the first time the authors use "artefacts" instead of archaeological, so there must be sites in the Raia et al. database.

Yes. The database contains sites with reasonably certain attribution and sites where the human presence is represented by artefacts.

This makes me wonder if archaeological sites are then related to species.

Yes.

I also have concerns about the accuracy of these "site" distributions over the last 2 Ma and I would like to see additional back up information

The original database (Raia et al. 2020) along with the metadata is now provided for review purposes. The actual database used here is a slightly updated one (see web-link above), which was peer-reviewed previously and published as Mondanaro et al. (2020).

In addition, the figure is really hard to see with the small size and the colours used in the symbols and background colours, so this really should be improved.

We have now removed the previous Extended Data Fig. 7, because one other reviewer also pointed out issues in the interpretation of the result.

As the authors can see from my commentary, so much hinges on the nature of the database. But without one, I can not say much more. I would strongly urge that a database be supplied in the supplementary information. I would also urge the authors to conduct their own evaluation of the integrity of the database (site and locality spatial distributions, type of site, dating methods and issues, etc), including commentary, supplementation perhaps, and tabled evaluation data. If the authors can provide more confidence with include a hominin fossil / archaeological site database, I for one would be much happier evaluating the points presented in the Results and Discussion and Synthesis sections.

We agree with the reviewer, that the species assignment needs careful consideration, both of fossils and archaeological sites. We spent a considerable amount of time reviewing the literature, and produced an 82 pages long explanation piece accompanying the database. The database has been used already in peer-reviewed 5 studies and has undergone multiple review processes (see below).

Carotenuto, F., Tsikaridze, N., Rook, L., Lordkipanidze, D., Longo, L., Condemi, S., & Raia, P. (2016). Venturing out safely: The biogeography of *Homo erectus* dispersal out of Africa. *Journal of human evolution*, 95, 1-12.

Carotenuto, F., Di Febbraro, M., Melchionna, M., Mondanaro, A., Castiglione, S., Serio, C., ... & Raia, P. (2018). The well-behaved killer: Late Pleistocene humans in Eurasia were significantly associated with living megafauna only. *Palaeogeography, Palaeoclimatology, Palaeoecology*, 500, 24-32.

Melchionna, M., Di Febbraro, M., Carotenuto, F., Rook, L., Mondanaro, A., Castiglione, S., ... & Diniz-Filho, J. A. F. (2018). Fragmentation of Neanderthals' pre-extinction distribution by climate change. *Palaeogeography, palaeoclimatology, palaeoecology*, 496, 146-154.

Mondanaro, A., Melchionna, M., Di Febbraro, M., Castiglione, S., Holden, P. B., Edwards, N. R., ... & Raia, P. (2020). A major change in rate of climate niche envelope evolution during hominid history. *Iscience*, 23(11), 101693.

P Raia, A Mondanaro, M Melchionna, M Di Febbraro, JAF Diniz-Filho, ...(2020) Past extinctions of Homo species coincided with increased vulnerability to climatic change
One Earth 3 (4), 480-490

We are confident that the database (which can be directly accessed through <https://ars.els-cdn.com/content/image/1-s2.0-S2589004220308853-mmc2.xlsx> and whose original metadata (Raia et al. 2020) we provide as part of this submission to facilitate the review) provides the most parsimonious and to this date most comprehensive compilation of well-dated and species-classified fossil and archaeological sites of our genus homo.

Moreover, we would like to re-iterate here that the nature of the climate envelope model is such that it is not very sensitive to individual misclassifications; but it would certainly be sensitive if entire clusters of datapoints were incorrectly assigned to the wrong species, which is highly unlikely, because the datasets are largely from independent data sources and authors.

Referee #2 (Remarks to the Author):

The Timmermann et al. manuscript uses a paleo-climate model run for the last 2 million years to simulate changes in the regional environment in which early humans evolved. They compare their results to a compilation of both the fossil and archaeological records. They are testing the hypothesis that changes in astronomic forcing significantly altered the environments of Africa causing the evolution, extinction and migrations of hominins. The model output seems to suggest that temperature, rainfall and vegetation (in terms of productivity) did in fact have an impact on human evolution.

This study builds on earlier work by this group and others. First the rich literature using proxy data to look at local and regional climate variability linking it to orbital variations – particularly eccentricity modulated precession. Second modelling work starting with the impacts of East African uplift on the climate and vegetation of East Africa in the 1990s and worth of mention is the excellent modelling work on the environment associated with Neanderthals by the authors of this current study in QSR.

We could all go into great detail and suggest different ways of modelling, different models to use, different scaling, but I am happy that the authors have chosen an appropriate set of models and applied them in the very best way they can. Because modelling is just away to test hypothesis in a temporal-spatial way that is not possible with proxy data.

We are grateful for the reviewer's helpful comments and suggestions to improve the manuscript.

There is, however, one issue which is difficult to deal with. The model outputs are downscaled to $1^\circ \times 1^\circ = 110 \text{ km}$ which is extremely good.

We agree.

But we know that there are huge variations in East Africa over the very short distances due to the mountainous topography. In Ethiopia you can go from cloud rainforest to arid desert in 50 km. East Africa is also dominated by lakes which are threshold mechanisms – if rainfall increases by a certain amount suddenly they fill up. There is also not a one to one relationship between rainfall, temperature and what vegetation exists. This is because of bifurcations in the climate-vegetation relation due to moisture feedback created by vegetation, particularly forests. No idea how you get round this micro/niche environment problem.

We appreciate all of the general issues raised by the reviewer, but many of them would go far beyond our current computational capabilities. One day, we certainly would love to do a 2-million-year transient climate simulation with a 10 km climate resolution with a realistic lake and hydrological model and a dynamic vegetation model. But this seems to be years away - not only for us, but for any research team in the community.

To be more specific:

Eventually, there is only real solution to get around the combined resolution / vegetation niche problem in a dynamically and physically consistent way, and that it through higher resolution modeling at a scale of about 10 km in the atmosphere (about a factor 1000 more computing time than used for our current simulation). Since our current simulation already ran for 6 months non-stop using our extremely generous supercomputing resources of about 1.2 PFlops (comprising a

total of 200 million CPU hours), we would need about 1000 supercomputers or wait for 500 years to get a 2 Ma transient simulation at the adequate resolution. Even with clever increased acceleration techniques, this would not be possible at this stage, not at any time in the near future.

Moreover, to fully capture the bifurcations of the climate-vegetation system, one would need a dynamic fully coupled vegetation model. Even most IPCC/CMIP6 coupled general circulation models (including the CESM 1.2 used here) do not use fully dynamic vegetation models, but simpler versions that have less dynamical freedom and likely under-represent bifurcation points. We think it is fair to say that even the coupled earth system models, which include fully dynamic vegetation components, which allow for the gradual regrowth of different vegetation types, interactive fire modules. and in principle also mammal-based seed-spreading mechanisms etc, are still in an experimental stage for transient paleo climate modeling and future climate change simulations. Again, we agree that this would be extremely desirable, but unfortunately, we cannot resolve the dynamic vegetation part in our long transient simulations. The same applies to lake models in CGCMs (e.g. CESM), which are extremely crude and don't allow for long-term changes in lake level or even drying out of lakes. All of these hydrological issues are best addressed with regional models and offline-hydrological models; but this would not be very helpful in the context of the global approach that we adopt in our study. But it is certainly worth exploring in future.

We greatly appreciate the reviewer's comment about resolution and would like to give at least one reason, why we think a 1x1 degree grid appropriate and that "downscaling" from the original T31 resolution (3.8 degree) to 1x1 degree gives a much higher gain in terms of resolving important climatically relevant features, than going from 1x1 degree to even 1/10 x 1/10 degree. This is illustrated in the figure below, which shows a snapshot of the simulated NETPP using the original T31 grid, and altitude-corrected/down-scaled to 1x1 degree, 0.5x0.5 degree and 0.1x0.1 degree. Clearly the qualitative gain from T31 to 1x1 is considerably higher than for other refinements. This includes the resolution of key topographic features in NETPP such as the Alps, the curved structure of the Himalayas, the Ethiopian highlands and of the Caucasus, which are all captured already in 1x1 resolution. We therefore keep our 1x1 target resolution in our study.

To address the resolution issue in our revised manuscript, we have added the following text as figure caption for a new extended Data Fig. 6 (see below) "Extended Data Fig. 6 Illustration of late-Holocene altitude-corrected downscaling of net primary production: Left: Simulated NPP (gC/m²/year) for original T31 atmosphere resolution, obtained from empirical NPP model using 1000-year late Holocene average of total precipitation and surface temperatures simulated by the CESM1.2 2Ma experiment; middle: same as left, but using altitude-corrections for temperature and precipitation as downscaling onto a 1x1 degree target grid, showing the emergence of key topographic features in Africa. This resolution was chosen in the calculations of the climate envelope model; right: for illustration, same as middle but for a 0.25x0.25 degree horizontal target grid. The qualitative gain in terms of regional details from T31 to 1x1 degree resolution outweighs the additional gain going from 1x1 degree resolution to 0.25x0.25 degree grid".

I do like that the authors have used randomised climate outputs to test their conclusions. This adds depth and makes it clearer their key findings.

We are happy to hear this.

I am also interested in the author's observation that 'largest density of hominin fossils is found in high NPP regions >200 gC/m²/year, with the exception of northeastern'. I think this needs unpacking and I am not sure I agree with their observation. First average NPP over 2 million years seems to go against the argument of the paper that variability or changing climate is important. Second hominins live in very specific places in East Africa – by very productive lakes so the mean may not capture this mean. Third there is also an upper limit – as early humans do not like highly productive regions such as, dense forest see Extended Data Fig. 7. What is interesting is that current fossil data still suggests all the early hominin species evolved in East Africa – the exception to the rule the authors have suggested.

We agree with the reviewer that this discussion was confusing. We have therefore deleted the paragraph and the corresponding Extended Data Figure. Instead, we discuss the NPP/habitat relationship and the optimal NPP range how it evolves in time in more detail in the context of Figure 2c.

I am not convinced by the *Homo heidelbergensis* discussion and suggest a comparison with recently published pan-African data may help here – by Kaboth-Bahr et al. (2021) in PNAS.

Thank you for the suggestion. We only became aware of the Kaboth-Bahr (KB21) 2021 study, after the first submission of our manuscript. Indeed, it is very helpful, and we are now referring to it in our revised manuscript. The KB21 paper also raises a number of interesting points, which I will discuss below:

First, we note that several of the input timeseries of KB21, don't have sufficient temporal resolution in their original version to capture the precessional cycle for the entire 600 ka. Interpolating them a posteriori onto a common higher-resolution time resolution does not resolve this issue. This is particularly true for MAL, MAG, CON, NAM. As a result, the principal components, which we downloaded from the PNAS Supplementary Material, do not have a pronounced precessional signal,

which is at odds with many other paleo-climate datasets in Africa, and of course, most notably the sapropel layers in the Mediterranean.

Moreover, we calculated the correlations between the KB21 principal components 1 and 2 (PC) with the DeltaSST(808-846) index (characterizing the zonal equatorial Pacific temperature gradient as a proxy for El Niño-like or La Niña-like conditions). The correlation values amount to only 0.19 and 0.15, respectively, which is very low. Even the fact that both PCs have (weak) correlations with the “ENSO” index which are in the same order of magnitude (0.1-0.2), even though the PCs themselves have a correlation of exactly zero, suggests only a very weak linkage between the chosen African hydroclimate proxy records and the index for the zonal equatorial Pacific SST gradient. We therefore wonder whether the notion of “paleo-ENSO-driven humidity changes” is justified given the low correlations.

A much more straightforward way to identify the pattern of African hydroclimate change using KB21 data with “ENSO” (rather than to calculate hydroclimate EOFs first and then search for ENSO linkages) is to calculate simply the regression between the KB21’s “ENSO” index and the KB21’s regional normalized hydroclimate records. This is what we did using the data in KB21 and the result is shown below.

The pattern is in fact not very meaningful, and it certainly looks totally different from the present-day analogue (Fig. 3, B in KB21) and the EOF1 and EOF2 in KB21. Because of the very low correlations between ENSO and the African hydroclimate PCs and the otherwise un-informative direct “ENSO”/hydroclimate regression pattern (see above), we are sorry to say this, but we have some reservations about the statistical inferences and interpretations of the KB21 paper.

Nevertheless, we would like to use this discussion as an opportunity to compare our climate model simulations with the KB21 data. We therefore conducted the same EOF analysis on the hydroclimate records at the KB21 sampling locations (marine records shifted on land, similar to PNAS paper). According to our analysis, the first and second mode of African hydroclimate variability on the subsampled 11 locations show the clear eccentricity modulated precessional cycles, in disagreement with the KB21 analysis, which shows only very little precessional variability. The third PC shows a more complicated signal characterized by a superposition of different orbital components and CO₂ forcing. The EOF patterns in KB21 and in our 2Ma simulation are also very different from each other. Whereas KB21 emphasizes an east-west dipole pattern associated with EOF1, the zonal asymmetries are much less clear in the CESM 1.2 simulation, as shown in the figure below.

EOF result from proxy data (Kaboth-Bahr et al. 2021)

EOF result from 2Ma simulation 11 point data

Overall paleo-data/ model intercomparison remains inconclusive, partly because the KB21 dataset lacks precessional variability, which is understandable because some of the original datasets used for the analysis don't resolve precessional-scale variability across the entire 600 ka.

Even though this comparison and extensive discussion maybe quite insightful, including the above-mentioned details and our concerns with some of the statistical interpretations would extend the scope of our study beyond our original intention, which was to elucidate the linkage between climate and the observed presence of hominins.

However, to highlight the novel aspect of remote (e.g. Pacific) forcings, which may play an important role in driving rainfall variability inside Africa (one key aspect of KB21), we have now added the following paragraph in our revised manuscript

“It must be noted here that regional climate- and resulting habitat changes were not only driven by the interplay of local forcings, but also by remote effects, such as eastern equatorial Pacific temperature changes, as has been illustrated recently by synthesizing African hydroclimate proxy records and tropical sea surface temperature reconstructions (Kaboth-Bahr et al. 2021).”

I also do not really understand the reasoning that early species were limited by temporal changes in climate while later species were spatially limited by climate change. The fundamental control on evolution is ‘does an individual reproduce successfully or not’ – do those traits get passed on to the next generation and provide some advantage or not. Hence why I am not sure how the temporal and spatial can be separated but do understand this may be due to the condensed writing style required by Nature.

The point is well taken. The discussion of temporal versus spatial adaptation was indeed too condensed and hard to follow. In our revised text, we try to de-compress this issue a bit further (given the word count limitations) because it is important and represents a perspective shift in the way we

look at adaptation to environmental changes. Most importantly, the variability selection hypothesis (which strictly speaking is a “temporal variability” selection hypothesis) ignores extended mobility of early hominins. What we show here by simply comparing the locations of hominin sites with the simulated net primary production at these sites (Fig. 2) is that this assumption appears to be well justified during the early Pleistocene and for *Homo ergaster* and *habilis*. However, the situation changes drastically with the emergence of *Homo heidelbergensis* after the mid-Pleistocene transition (MPT). They become global wanderers, which meant that by moving around they encountered much larger range of climatic conditions, than just staying put and experiencing the local temporal climatic variations. This encounter with large spatial climatic gradients likely boosted our genus’ capability for climatic adaptation, which in turn also extended their range. This second post-MPT phase is characterized by spatio-temporal climatic adaptation, which is different from the stationary temporal adaptation of our early ancestors.

The previous version (160 words)

*“Moreover, from 2–1 Ma the simulations show a gradual trend towards increasing habitat suitability in Africa with median values straddling between 200–400 gC/m²/year and an increase in the range of net primary productivity occurring at the key central African fossil sites (Fig. 2c). Subsequently, as other hominin groups, dispersed into Eurasia they adapted to a wider range of net primary production (Fig. 2c) – most notably for *Homo heidelbergensis* from 0.9–0.4 Ma. We therefore suggest that initially early Pleistocene hominins in Africa had to cope mostly with temporal climatic variability. This pre-adaptation phase which lasted from 2–1 Ma may have primed our genus to become “wanderers” and eventually habitat generalists. After the mid-Pleistocene transition, the geographic range of hominins expanded dramatically, thereby exposing migrating groups to larger spatial climatic variability (Fig. 2c), which likely resulted in genotype changes. To understand hominin speciation during the Pleistocene, we need to account for the full spatial and temporal complexity of the climate signal.”*

Is now modified to (291 words)

*“Moreover, during the early Pleistocene (2–1 Ma), *Homo ergaster* and *Homo habilis* occupied two main habitats – one in central eastern and one in southern Africa (Fig. 2b). On average, these two hominin groups preferred geographic regions which were characterized by a relatively stable NPP straddling between 200–380 gC/m²/year (Fig. 2c). However, there were several intervals during the Early Pleistocene (around 1.42, 1.1 and 0.95 Ma), when central and southern African hominins encountered higher regional NPP values of up to 520 gC/m²/year. However, in the absence of any clear geographic expansion, these early groups mostly adapted to local orbital-scale variations in climatic and NPP (e.g. Fig. 1 g–i, Extended Data Figs. 3,4,6), as reflected also in their habitat suitability (Fig. 1). After the mid-Pleistocene transition and with the emergence of the first *Homo heidelbergensis* ~850 ka, the dynamics changed dramatically: hominins began to migrate into Eurasia and other regions, encountering along their journey a much wider spatial range of NPP from extremely low values of 20 gC/m²/year to values beyond 600 gC/m²/year (Fig. 2c). These migrating groups crossed large spatial gradients in climate and NPP, that by far exceeded the temporal ranges in NPP experienced by their more stationary Early Pleistocene ancestors. This transition to global wanderers must have required *Homo heidelbergensis*, to acquire new adaptation skills, which in turn also boosted their ability to further expand their geographic range, thereby providing a strong positive migration/climate adaptation feedback. Therefore, to understand hominin evolution during the Pleistocene, one needs to account for the full spatial and temporal complexity of the climate signal. Summarizing, we find that in the early Pleistocene – except for *Homo erectus* – early hominins adapted mostly to temporal shifts in climate, whereas after the mid-Pleistocene transition our ancestors become global wanderers.”*

Overall this is an interesting and stimulating piece of science. The authors try to test the hypothesis posed by many colleagues that orbital forcing changed the local/regional environment of Africa which created opportunities for evolution and extinction of hominin species. The modelling is excellent as is the comparison with the fossil record.

We are pleased to hear that the reviewer is satisfied with the modeling approach and the data/model comparison.

The issues are with model resolution – but this is true of all model based studies.

We hope to have addressed by including a new Extended Data Fig. 6, which clearly shows that going to higher resolution (0.25 degree) won't reveal many new qualitative features, that would be relevant for our large-scale climate envelope model. Going further beyond that would require supercomputing resources, which are currently not available to anyone– unfortunately.

There is an issue with many assumptions being made in the modelling all of which sound reasonable but not create the realistic output wanted (the compounded error/assumption idea).

Unfortunately, this is intrinsic to any modeling approach that tries to quantify the effect of past climate change on hominin evolution and dispersal. There is no model without assumptions. In any case, we tried to test the robustness of our key results by accounting for the age uncertainties of the fossil and archaeological sites. Our climate envelope model is also relatively stable to inclusion or omission of individual sites. This appears to be the great advantage of our large-scale approach which projects archaeological/fossil sites for each species onto their own climate envelope – an advantage of the Mahalanobis distance which assigns cumulative probability values to the dataset, using a data-based co-variance metric.

I have struggled with the interpretation of the results which seems to jump beyond the model results and tries to explain what caused the evolution of key species.

In our revised manuscript, we have now streamlined the text, swapped the sequence of some paragraphs, added subsection headings to increase clarity and logical flow.

Referee #3 (Remarks to the Author):

Review of 2021-07-10738 By Chris Brierley, UCL

This paper represents a substantial body of work. The ability to create suitability maps varying in time for 5 different hominin species appears a wonderful result. My expertise doesn't cover all the disciplines included in this manuscript. So I can't comment on whether this is a substantial advance on existing work, but the depth of the climate modelling contribution to it certainly is.

We appreciate the reviewer's constructive comments.

To my knowledge, no one has yet undertaken a transient simulation with the model a sophisticated CCSM3 for anything like this length of simulation. I would be very interested to see they evolution of the climate within these simulations, but I accept it isn't the focus of this particular manuscript and I hope it will come out in another paper later.

Indeed, there is subsequent paper in preparation that will focus on the physical aspects of our 2-million-year simulation, namely the energy transport in the climate system and how it changed from Early to Late Pleistocene conditions.

The creation of a habitability metric seems to be have been done using a suitable and appropriate method – although I confess I am not an expert in this kind of model.

Items to be addressed/clarified:

I was surprised that there was no justification of the choice of the 4 input variables to this model. The CESM run allows for many interesting variables to be investigated (for example, relative strength of a 1 in 10 year drought on soil moisture). While I don't think that all such options should be investigated, the reasoning behind picking the 4 that are used should be explained.

The point is well taken. We have in fact run the hominin habitat model for a variety of variables also including the amplitude of the seasonal cycle of precipitation and temperature. Interestingly, the results were quite similar for the resulting hominin habitat values. The reason is that – in particular in the tropics - there are strong correlations between annual mean precipitation (used in our final selection) and seasonal cycle changes of precipitation (due to the fact that precipitation is a positive definite variable and a change in variability also rectifies into a mean state). First and second moments are coupled. The seasonal cycle of temperature contains very similar spatio-temporal information as the two selected variables of annual mean temperature and minimum temperature. Moreover, some of the relevant climatic variables are correlated on orbital timescales, either because their variability is dominated by the CO₂ forcing (annual mean temperature or minimum temperature), or the precessional cycle (e.g. annual mean precipitation and seasonal amplitude of precipitation). This implies that by including other variables or replacing annual mean with seasonal amplitude, not much new and independent spatio-temporal information is gained for the climate-envelope model.

We have finally decided to use annual mean temperature, annual mean precipitation, minimum yearly precipitation and Net primary productivity, because each of them plays an important role in

the physiological constraints on hominin survival (T, PPmin) and availability of food resources (NPP).

Each of the climatic parameters used for the climate envelope model represents a time mean over 1000-years and the habitat suitability maps are also 1000-year means. We have therefore filtered out the effect of interannual to centennial-scale temperature and hydroclimate variability. The proposed idea to use shorter-term drought indices is interesting, but we think that even though the occurrence of 20-year drought may impact the habitat suitability temporarily, it won't affect our 1000-year mean view of habitats. But it may be worth exploring in subsequent studies, how shorter-term hydroclimate variability may influence this first order picture.

To better explain our choice of input variables we have now added the following text in the Methods section:

*“To derive a climate envelope model (CEM) which best characterizes the habitable conditions for hominins, we choose 4 key climatic variables: annual mean temperature and precipitation (T^*_{am} , P^*_{am}), minimum precipitation (P^*_{min}) and terrestrial net primary production (N). Obtained as 1000-year downscaled averages ($1^\circ \times 1^\circ$ horizontal resolution), these variables, which are related to hominin survival (T^*_{am} , P^*_{min}) and the availability of food resources (P^*_{am} , N), are combined as a 4 dimensional climate environment vector $C(t)=(T^*_{am}, P^*_{am}, P^*_{min}, N)$, with 2000 values in time (t), which correspond to 1000-year (400-year) orbital (model) means of the 2-million year simulation.”*

And

“The stability of the CEM has been tested by using different combinations of climate parameters (annual mean and seasonal range of temperature and precipitation and annual mean and minimum values of temperature and precipitation) and dimensionalities. The key conclusions of our study remain basically unchanged, because of the high temporal correlations of some of these variables on orbital timescales.”

The artificial inflating of the CO₂ forcing by 1.5x in the GCM simulation seems unjustified. The climate sensitivity of CCSM4 at T31 resolution is 2.90C according to Shields et al (2012), so CESM1.2 should be similar.

The equilibrium climate sensitivity of our model version under present-day conditions without CO₂ amplification is only 2.4°C – not 2.9°C.

This is pretty-much exactly the central estimate according to the new IPCC AR6 report.

2.4°C and even 2.9°C are very different from the CMIP6 average value of 3.7 +/- 1.2°C – see Meehl et al. 2020 (PNAS). However, some CMIP6 models have difficulties in reproducing the observed temperature ranges. A more suitable estimate might be the IPCC AR6 consolidated estimate of 3°C, with a very likely range of 2°C to 5°C and a likely range of 2.5°C to 4°C (AR6, Summary for Policymakers). In this regard, the 2.4°C ECS from our CESM1.2 version would be in the lower range of the IPCC very likely estimate and outside the IPCC AR6 likely estimate. Other important reasons to enhance the radiative forcing in our model are: i) the lack of explicit N₂O, CH₄, dust and vegetation forcings – each of which is expected to correlate with the CO₂ on glacial-interglacial timescales; ii) the fact that only by increasing the radiative forcing in our model, can we reproduce the 6°C LGM/late Holocene temperature range estimated from Tierney et al (2020), and the

glacial/interglacial range estimated from SST proxy data in Nature and Friedrich and Timmermann
et al. (2016) Science Advances.

In fact, the value of 4.5°C reached by Friedrich and Timmerman (2020) is even outside the likely range from the IPCC.

Correct; but... it is still within the very likely range of IPCC, and within the range (even though upper bound) of the CMIP6 models (3.7 +/- 1.2 °C). Using the data from Meehl et al. 2020 “Context for interpreting equilibrium climate sensitivity and transient climate response from the CMIP6 Earth system models” PNAS, we can see that even the warm climate Equilibrium Climate Sensitivity (ECS) in Friedrich and Timmermann (2016) of 4.5°C is smaller compared to that of 14 out of 39 other CMIP6 models used in Meehl et al. 2020. Given that the mean ECS in CMIP6 is 3.7°C with a standard deviation 1.2°C, we consider the warm climate estimate of Friedrich and Timmermann still within the likely range of CMIP6 earth system models; but outside the likely range of the older CMIP5 version and outside the IPCC, AR6, SPM likely range. Moreover, in the context of our study here, it may be more relevant to use the ECS estimate of 3.22 °C as a benchmark, which is representative of the entire 784 ka history in Friedrich and Timmermann (2016), and not just the interglacials. Clearly the discussion is complicated and given the limited space we have decided to keep it to a minimum in our revised Methods section.

Clearly, there could be some non-linearity in the Pleistocene value due to the ice-sheets, but that sits more happily into a earth system sensitivity framework. There needs to be a much better justification of this, as otherwise the CESM simulation would artificially inflates the (greenhouse gas forced) climate variations seen over past 2 Ma. I note that the timeseries in EDFig 2 suggest otherwise, but how can you discount these biases as arising from downscaling (something you argue is vital) or deficiencies in the ice-sheet forcings.

This is indeed an interesting point. Without the CO₂ factor 1.5 amplification, the amplitude of the model-simulated global mean, tropical and Antarctic temperature variations would not be in good agreement with the corresponding paleo-proxy reconstructions (Fig. Extended Data Fig. 2a, Extended Data Fig. 1a, Extended Data Fig. 2b, respectively). This agreement cannot be easily achieved by keeping the standard CESM1.2 ECS and accounting for uncertainties only in ice-sheet and sea-level forcings, which contribute 18.5% to the total forcing (see supplementary Figure S7 in Friedrich and Timmermann).

To capture the results from the discussion above, and the consider three main reasons for inflating our radiative forcing (low ECS of CESM1.2, missing non-CO₂ radiative forcings, matching the paleo glacial/interglacial range), we have revised the Methods paragraph to

“The model uses the bathymetry of the Last Glacial Maximum and time-varying forcings of greenhouse gases¹⁵, ice-sheets¹⁵ and astronomical insolation conditions¹⁶. Our CESM1.2 version has a relatively low equilibrium climate sensitivity (ECS) of 2.4°C per CO₂ doubling, which lies outside the likely range of estimates⁴⁴ (3.7°C +/-1.2°C) obtained with other climate model simulations conducted as part of the Coupled Model Intercomparison Project, phase 6 (CMIP6), but within the lower range of recent estimates compiled by the IPCC 6th assessment report⁴⁵ of Working Group 1, which identify a very likely ECS range of 2°C to 5°C. To have a more realistic response to past longwave radiative forcings in our paleo-climate model simulation and to implicitly capture radiative effects of other CO₂-correlated forcings from dust, vegetation, N₂O or CH₄, we therefore scaled the range of the applied CO₂ forcing¹⁵ by a factor of 1.5. The resulting effective equilibrium climate sensitivity (which includes non-CO₂ greenhouse gas forcings) amounts in our case to ~3.8°C, in reasonable agreement with the CMIP6

estimate and previous paleo climate estimates^{18,19} of 3.2°C, which were obtained from reconstructions of the global mean surface temperature and radiative forcings covering the last 784 ka. The amplification of the CO₂ forcing in CESM1.2 leads to a realistic representation of the amplitude of global mean, tropical and Antarctic temperature changes (Extended Data Figs. 1b, 2a, b) and to a simulated temperature range between Last Glacial Maximum and Late-Holocene conditions of ~5.9°C, in close agreement with recent paleo-proxy based estimates⁴⁶.”

It is a shame that the CESM1.2 simulation did not deploy either interactive vegetation, which would have avoided the need for the NPP approach. Isotopes would have been really useful for subsequent studies too."

Isotopes would have certainly been useful for subsequent paleo studies, but they are not crucial in the context of our submitted hominin succession paper.

I am confused by the multiple downscaling steps (as shown in ExtD Fig. 5). This figure shows that there are 3 interim steps to go from the CLIMBER simulation to the species distribution. As I understand it, the CESM1.2 acts as an initial dynamical downscaling, which is then followed by additional statistical downscaling and an empirical NPP model. This raises some questions:

1. What is the added benefit of the CESM downscaling step? It comes at a considerable computational cost, but are the conclusions the same as just doing a statistical downscaling from the CLIMBER variables?

The CLIMBER => CESM step should not be considered a “downscaling” step. We simply use the ice-sheet and CO₂ evolution from CLIMBER as a forcing for CESM1.2. However, none of the other physical variables (rain, temperature etc.) from CLIMBER are used in the CESM1.2 simulations, nor in any of the subsequent steps of our analysis.

CLIMBER is a very coarse 2.5-dimensional model with 51°x10° atmospheric resolution in longitude and latitude. However, for the ice-sheet model CLIMBER uses a higher resolution of 1.5°x0.7°, which in fact is even higher than the original horizontal resolution of CESM1.2 (3.85 degrees).

CLIMBER's strength is its computational efficiency and the fact that it includes both carbon cycle and ice-sheet models. However, its representation of trade wind systems, the Intertropical Convergence, modes of climate variability and other regional features is inadequate to be used for a standard dynamical downscaling to a 1x1 degree grid. In fact, one would have to develop new downscaling techniques to deal with a refinement factor of 51x10. This would be far outside the scope of this paper.

We have therefore applied a different, much more suitable method, in which CLIMBER higher-resolution ice-sheet and CO₂ forcings are applied to a full earth system model. The CESM1.2 model generates realistic atmospheric and ocean circulations and large-scale rainfall patterns and monsoon systems are well resolved, as demonstrated by a plethora of CCSM4 and CESM1 studies. What a 3.8-degree resolution spectral atmospheric model does not capture well are smaller-scale topographic features, such as the Ethiopian Highlands, the Caucasus, Carpathians etc. which may have all played an important role on hominin dispersal and settlement. In particular the effect of these topographic features on temperature and rainfall is not well represented in coarse resolution earth system models. To account for altitude effects on climate, we apply an additional step to the CESM1.2 output. Even this step should not be considered as a *sensu-stricto* “statistical downscaling”

approach, because it simply uses the atmospheric lapse rate for temperature and the Clausius Clapeyron relationship for moisture to the surface height difference between Etopo5 and CESM1.2, which is then interpolated back onto a 1x1 degree.

The only benefit of this step is to account for topographic barriers more accurately in our hominin habitat estimates. We realize that the previous description of the 3 steps was not very clear. We have therefore revised the explanation accordingly.

2. Why is 10 degree resolution sufficient, but T₃₁ isn't? Does this relate to the spatial range roamed by hominins present at the fossil sites? Or are the topographic differences more important – but I'd guess that the altitudinal differences between the individual sites and ETOPO₁ are greater than the altitudinal differences between ETOPO₁ and the T₃₁ model. I don't see any discussion of these points in the paper.

We appreciate the reviewer's comment about resolution, which was also raised by reviewer 2. We would like to give at least one clear indication, why we think a 1x1 degree grid appropriate and that "downscaling" from the original T₃₁ resolution (3.8 degree) to 1x1 degree gives a much higher gain in terms of resolving climatically relevant features, than going from 1x1 degree to even 1/10 x 1/10 degree. This is illustrated in the figure below, which shows a snapshot of the simulated NETPP using the original T₃₁ grid, and altitude-corrected/down-scaled to 1x1 degree, 0.5x0.5 degree and 0.1x0.1 degree. Clearly the qualitative gain from T₃₁ to 1x1 is considerably higher than for other refinements. This includes the resolution of key topographic features in NETPP such as the Alps, the curved structure of the Himalayas, the Ethiopian highlands and of the Caucasus, which are all captured already in 1x1 resolution. We therefore keep our 1x1 target resolution in our study.

To address the resolution issue in our revised manuscript, we have now added a new Extended Data Fig. 6 (see below with the following figure caption):

“Extended Data Fig. 6 Illustration of late-Holocene altitude-corrected downscaling of net primary production: Left: Simulated NPP ($\text{gC}/\text{m}^2/\text{year}$) for original T31 atmosphere resolution, obtained from empirical NPP model using 1000-year late Holocene average of total precipitation and surface temperatures simulated by the CESM1.2 2Ma experiment; middle: same as left, but using altitude-corrections for temperature and precipitation as downscaling onto a 1x1 degree target grid, showing the emergence of key topographic features in Africa. This resolution was chosen in the calculations of the climate envelope model; right: for illustration, same as middle but for a 0.25x0.25 degree horizontal target grid. The qualitative gain in terms of regional details from T31 to 1x1 degree resolution outweighs the additional gain going from 1x1 degree resolution to 0.25x0.25 degree grid”.

- Does the empirical NPP model really do a better job at capturing the temporal changes in NPP simulated by CESM1.2? I understand the argument about the prescribed PFTs, but why not just look at the NPP for the individual PFTs and if necessary recombine them. This NPP step feels really crude.

The CESM1.2 model was run with the carbon and nitrate option turned off, which means that except for the areas where the ice-sheet changes, the plant functional types did not change. We therefore did not simulate changes in NPP in our CESM1.2 simulation. To overcome this limitation, we use a posteriori an empirical parameterization for NPP which uses the elevation-downscaled temperature and precipitation data.

The fidelity of the Del Grosso (2008) model is further tested here, by using the temperature and precipitation output from a pre-industrial simulation conducted with the Community Earth System Model, version 2 in 1x1 degree resolution and applying it to the empirical model Del Grosso model to calculate NPP. The comparison between the original NETPP (calculated from the complex CLM5 model, left side figure below) simulated by CESM2 and the Del Grosso model (a handful of diagnostic equations, right side figure below) using the CESM2 precipitation and temperature annual mean data shows an overall good agreement (pattern correlation of 0.8). Obviously, a simple diagnostic model does not capture the full complexity of CESM2/CLM5 (which includes more complex soil processes, fire, seasonality, sunlight, etc.), leading to discrepancies e.g. over central US, northeastern Siberia or central India, but the general agreement in key regions for hominin dispersal considered in our study strongly supports the viability of our a-posteriori NPP approach.

The fidelity of the NPP parameterization for present-day conditions has been further documented in the original Del Grosso et al. (2008) model

https://www.academia.edu/download/39883108/Global_potential_net_primary_production_2015_1110-14429-1k764bx.pdf

- 1 Raia, P., Mondanaro, A., Melchionna, M., Di Febbraro, M. & Diniz-Filho, J. A. F. Past Extinctions of Homo Species coincided with increased vulnerability to Climatic Change. *One Earth* 3, 1-11 (2020).
- 2 Etherington, T. R. Mahalanobis distances and ecological niche modelling: correcting a chi-squared probability error. *PeerJ* 7, doi:10.7717/peerj.6678 (2019).
- 3 Farber, O. & Kadmon, R. Assessment of alternative approaches for bioclimatic modeling with special emphasis on the Mahalanobis distance. *Ecological Modelling* 160, 115-130, doi:10.1016/s0304-3800(02)00327-7 (2003).
- 4 Mondanaro, A. *et al.* A Major Change in Rate of Climate Niche Envelope Evolution during Hominid History. *Iscience* 23, doi:10.1016/j.isci.2020.101693 (2020).
- 5 Grove, M. Speciation, diversity, and Mode 1 technologies: The impact of variability selection. *Journal of Human Evolution* 61, 306-319, doi:10.1016/j.jhevol.2011.04.005 (2011).
- 6 Potts, R. Variability selection in hominid evolution. *Evolutionary Anthropology* 7, 81-96, doi:10.1002/(sici)1520-6505(1998)7:3<81::aid-evan3>3.3.co;2-1 (1998).
- 7 Meehl, G. A. *et al.* Context for interpreting equilibrium climate sensitivity and transient climate response from the CMIP6 Earth system models. *Science Advances* 6, doi:10.1126/sciadv.aba1981 (2020).
- 8 Willeit, M., Ganopolski, A., Calov, R. & Brovkin, V. Mid-Pleistocene transition in glacial cycles explained by declining CO₂ and regolith removal. *Science Advances* 5, doi:10.1126/sciadv.aav7337 (2019).
- 9 Friedrich, T., Timmermann, A., Tigchelaar, M., Timm, O. E. & Ganopolski, A. Nonlinear climate sensitivity and its implications for future greenhouse warming. *Science Advances* 2, doi:10.1126/sciadv.1501923 (2016).
- 10 Friedrich, T. & Timmermann, A. Using Late Pleistocene sea surface temperature reconstructions to constrain future greenhouse warming. *Earth and Planetary Science Letters* 530, doi:10.1016/j.epsl.2019.115911 (2020).

Author Rebuttals to First Revision:

Referees' comments:

Referee #1 (Remarks to the Author):

I complement the authors in their attempt to examine long term climate effects on hominins. With respect to orientation, this is the type of big picture synthesis deserving of publication in Nature.

I reviewed a previous version of this paper as Reviewer 1. In that review I noted considerable frustration at not being able to access the hominin and archaeological data set behind the paper. The authors responded in a revised version of the paper by supplying us with the archaeological data set. I assume that this data set would be freely available and accessible if this paper is eventually published?

The database provided by the authors consists of 2572 entries for hominin fossil and archaeological sites in the Old World between 2 MA and 30 ka. As other reviewers are experts on modelling, I center my comments exclusively on the nature of the database and its implications.

From the outset, I must unfortunately air my reservations about the quality of the data set and the 'species attribution variable'. I do agree with the reviewers that this is a decent data set, but at the same time, I believe it should be considerably improved for increased accuracy. (Because I am not confident in the data set, I have not made comments on the conclusions drawn in the main text of the paper).

According to their rebuttal, Figure 1 is the main map of all sites based on the database entries. With the authors caveats that Figure 1 does not necessarily portray all sites (one dot may represent more than one site once you zoom in), I am struck by a couple of things. One point is that the large majority of the sites for all hominin species attributions are located in Europe (naturally with the exception of ergaster + habilis). I well recognize that the majority of Palaeolithic research has been, and continues to be, conducted in Europe. This, of course, introduces serious geographic biases, which the authors should probably comment specifically on, and how it may influence the modelled evidence and conclusions. That said, another significant concern is that the data sets outside of Europe do not appear to be representative of published data. In some cases, I believe this is serious and should be tackled. For instance, key data is missing, resulting in substantial geographic gaps in all insets of Figure 1A to D – where there should be available, published data.

As I was concerned about the missing data portrayed in the main text, I did a quality check of the database, examining the representation of sites in several regions (and certainly not all). I wish to make it clear that my survey of the data set is by no means comprehensive, and I am sorry if I make some mistakes below through my quick survey - but I think the authors will get my point, as we proceed in going through some example regions.

Arabia. Only 4 Pleistocene sites are listed in the database. These range between 127 to 38 ka. This is of immediate concern, with significant omissions for the cultural history of Arabia. This pertains to

both older dates and younger dates, easily retrievable in the literature. For example, two papers led by Scerri et al. (2018, 2021) are in Scientific Reports, and a recent one is by Groucutt et al. (2021) in Nature; these sites and ages are missing in the database. This is important because these publications report on a series of ages exceeding 127 ka, and down to 400 ka, so a prehistory extending for 275 ka is absent in the paper. These older sites are presumably manufactured by archaic Homo. The missing Acheulean and the Middle Palaeolithic site assemblages are inferred to be manufactured by archaic hominins and H. sapiens in the papers. In addition, key publications which report on other key Middle Palaeolithic sites are also missing in the dataset. For instance, three Middle Palaeolithic sites, each dated to ca. 55 ka, are missing – these are located in Saudi Arabia (Jennings et al., 2016; Groucutt et al. 2021) and Yemen (Delagnes et al., 2012). These 55 ka year old sites are inferred to be manufactured by both H. sapiens and the Neanderthals in publications. What is the overall effect? Arabia appears as a blank spot on the maps for key time frames - see Figure 1, which only has archaeological and fossil sites depicted as Homo sapiens on the inset. Furthermore, this means that other hominin species were occupying habitats that were in semi-arid and pluvial intervals in Arabia. This was not just confined to H. sapiens per se. [NB: please correct the database entry for Jebel Faya. It is in the UAE, not Saudi Arabia]

Iranian plateau. Only 3 Pleistocene sites are listed, ranging to between 31 to 39 ka. A readily available synthesis of all ages is available for Iran, see Shoaee et al. 2021, JAA, Table 1. In that synthesis the authors will also see that Middle Palaeolithic site ages extend back to 50 ka (at Mirak), beyond the 39 ka mark presented in the data set. Given that Neanderthals are present in the Zagros (Shanidar), I assume the authors would classify Mirak as Neanderthals – though no attribution of species has ever been made for this site. Missing from the data set is also an important new Neanderthal fossil discovery at 43-41 ka at Bawa Yawan (see Heydari-Guran et al., 2021, PLOS ONE). I would think that these ages would enhance the Neanderthal and H. sapiens data and the maps presented in the paper – with a wider range of hominin species in various habitats.

South Asia. The authors list approximately 18 Palaeolithic sites ranging from ca. 1.8 Ma to ca. 40 ka. While they have captured some of the literature, there are a number of problems with this data set, some of which I highlight. Firstly, I am concerned about the way in which the archaeological sites are typed into species, as no practitioner working in South Asia has even tried to do so.

In the paper, the archaeological sites and ages ranging between 1.8 Ma and 236 ka are typed as Homo erectus; those between 385 to 268 ka are H. heidelbergensis; those between 150 – 40 ka are H. sapiens; and a unusual category of H. sapiens/H. heidelbergensis are between 210 and 125 ka. From a fossil perspective, there is only one fossil of an archaic hominin at Hathnora (which the authors categorise as Homo erectus and dated to 236 ka – both of which are contentious and debated in the literature as to species and the cited date – other chronometric ages have been published). There are other hominin fossils - H. sapiens after 48 ka - but data on Sri Lanka is not presented in this paper (see below).

With respect to the oldest ages (1.8 Ma to 236 ka), there are significant issues. Though 7 sites are presented, 5 of these have been significantly disputed by multiple authors – the ages are not widely accepted at all and only adhered to by a small number of Indian scholars who first reported the ages. For the ages between 385 and 268, I am unclear why these are H. heidelbergensis sites (if Hathnora

is 236 and said to be *H. erectus*, why are the ages in this category *heidelbergensis*?). For the *H. sapiens* sites, the authors push back our species to 150 ka in India – hugely contentious, of course, in the absence of fossils. For the *H. sapiens*/*H. heidelbergensis* category – I do not understand why these sites are placed in a hybrid list (/) as opposed to the other sites in the *heidelbergensis* and *sapiens* categories. Some of these sites and ages are certainly valid, so I can not follow the logic of classification. A substantive issue for the Indian data set is that a number of valid ages are absent – for example, other localities and ages at Jwalapuram, the Dhaba site, etc.

For Sri Lanka, no archaeological or fossil sites are presented in the data set. This is so despite its widely published ages between 48 ka and 30 ka. Consequently, there is no data and these sites do not appear on Figure 1. The importance of the Sri Lankan sites is that they are situated in rainforests – some of the very best evidence for the adaptation of hominins in a rainforest ecosystem - this is relevant to the ecological and environmental arguments presented in the paper.

May I suggest that the authors consult and review synthesis papers that present dated archaeological sites of South Asia – for example, Blinkhorn and Petraglia (2017), *Curr. Anthrop.* In publications such as these, the authors will also see how the data are treated – e.g., sites earlier than 100 ka are typically associated with Acheulean toolkits, and thus thought to be the product of archaic hominins.

Mongolia. Only 1 site is presented, Tsagaan Agui, ranging to between 56-30 ka, and classed as Neanderthal/*sapiens*. The authors do not seem to be aware of the growing number of well-dated sites in Mongolia (Tolbor, for example) – again data are tabled with respect to sites types and dates. There is also the key fossil *H. sapiens* at Salkhit (34 ka). It seems to me that places like Mongolia are very important to show expansion events, and climate/ecosystems, that *H. sapiens* was inhabiting.

China. A larger set of fossil and archaeological sites are available for China, and the authors present a pretty good list. However, I did a brief check of the data base, and I do see some missing sites. For instance, why is the 2.1 Ma year old site complex at Shangchen (Zhu et al. 2018, *Nature*), not in the database? I recognize that this is perhaps due to the fact that the Timmermann et al. paper concentrates on 2 Ma and younger, but younger sites are present at Shangchen, and these do not appear to be tabulated. With respect to Lower to late Middle Pleistocene sites, some key sites and dates are not present – please refer to the comprehensive tabled data in Yang et al., 2020, in *QSR* – a cross-check between the data set and published synthesis papers like these would give us more confidence.

As I expressed for the South Asian data set, I have the same reservations with assigning ages and sites to species - I would think that this is likely to be greeted with much skepticism. I also do not understand the hybrid categories (/) and how they are used in this paper for the analysis, i.e., *H. heidelbergensis*/*H. erectus*; *H. neanderthalensis*/*H. heidelbergensis*. Is the data in these hybrid cases treated in the same way as the data in single-species categories or is this data excluded? I am drawn to the *H. sapiens* category - the authors must recognize that placing some of the early sites into *H. sapiens* will be contentious. Moreover, plugging sites such as Zhiren cave into *H. sapiens* will be treated with some skepticism as the mix of morphological attributes has long been noted, defying easy classification.

Other regions. With respect to other regions, I am wondering why data from some parts of SE Asia are included in the data set (Java), but not other key regions with fossils and Pleistocene fossils in both mainland and island contexts? Is there a reason for excluding these key data sets, but only including Java? Please explain.

I also wonder about the exclusion of Australia, as data is readily available, and presumably good information about climate and environments would emerge by inclusion of this continent.

Species Assignments. I will not repeat my reservations, as I did above, about placing archaeological sites into simple species categories. I would like to query the authors here as to why some species are entirely excluded in the data set. There is no category for Denisovans, for instance, and this fossil and archaeological data is presumably ignored. I suppose the authors intend not to include other species – such as *H. floresiensis* and *H. luzonensis* – an explanation on why these species and regions are excluded from analysis would be useful.

One comment and suggestion about species attributions. Perhaps it would be best to rethink how species is being categorised in this paper. I think it would be reasonable if the authors, for example, only use species for sites that they are confident about. For archaeological sites without any sign of species, I believe it would be much more logical to devise categories, e.g., *Homo* sp. A, *Homo* sp. B., and examine these relative to time slices. Inferences in a discussion still could be made about possible species. I hope that the authors understand my point, as I believe that the categorization of archaeological sites into simple species taxonomies gives a false impression of human evolutionary history.

Referee #3 (Remarks to the Author):

I believe that the authors have responded to my comments in a satisfactory fashion, and see no further obstacle to publication w.r.t. the climate modelling aspects. I appreciate the greater discussion and justification in the revised manuscript of the approaches adopted.

Personally I wouldn't have used the CMIP6 ensemble's ECS as a benchmark to aim at, especially with NCAR having made an alternate version of CESM2 specifically because their LGM simulations rule the normal version out. But I accept that other members of the community don't share my disquiet, so see no need for the authors to further revise their text. The non-CO2 forcings is also something that I hadn't appreciated earlier.

Chris Brierley (UCL Geography)

Author Rebuttals to First Revision:

----- Original manuscript number (2021-07-10738A) -----

Response to Reviewer 1 (our response in **blue**)

Referee #1 (Remarks to the Author):

I complement the authors in their attempt to examine long term climate effects on hominins. With respect to orientation, this is the type of big picture synthesis deserving of publication in Nature.

R: We agree and would like to thank the reviewer for the constructive comments, which we addressed in detail below.

In our revised manuscript we extended our original fossil and archaeological data compilation by 20% and ran several sensitivity experiments with our habitat suitability model (addressing also species attribution uncertainty). The main conclusions from our previous manuscript version remain unchanged.

I reviewed a previous version of this paper as Reviewer 1. In that review I noted considerable frustration at not being able to access the hominin and archaeological data set behind the paper. The authors responded in a revised version of the paper by supplying us with the archaeological data set. I assume that this data set would be freely available and accessible if this paper is eventually published?

R: The entire dataset will be made available as part of this submission (supplementary data), especially so since it is no longer the original data, but an extended version, which includes 491 more entries, including those suggested by reviewer 1.

The database provided by the authors consists of 2572 entries for hominin fossil and archaeological sites in the Old World between 2 Ma and 30 ka. As other reviewers are experts on modelling, I center my comments exclusively on the nature of the database and its implications.

From the outset, I must unfortunately air my reservations about the quality of the data set and the 'species attribution variable'. I do agree with the reviewers that this is a decent data set, but at the same time, I believe it should be considerably improved for increased accuracy. (Because I am not confident in the data set, I have not made comments on the conclusions drawn in the main text of the paper).

R: To address the reviewer's concerns we have reviewed all the original data, and extended the original Raia et al. 2020 data by an additional 491 records including those suggested by the

reviewer. This was a major effort, which, we believe, deserves an official new release of our extensive data compilation, which we will include as part of our submission of the revised manuscript through additional supplementary data. With this, the research community will now be given open access to this most comprehensive compilation of radiometric species-stratified archaeological and fossil data (excluding single fossil species, Australia and the Americas -see comment below). Furthermore, we made an additional attempt to address uncertainties in species attribution and age uncertainties (see our response further below).

We have included a more detailed description of our dataset in the Methods section. The references for each data entry are provided in the supplementary excel sheet. Our revised methods section (which addresses several key points of reviewer 1) now summarizes the main differences to the Raia et al. 2020 dataset and the overall approach to treat the data. The relevant section reads now:

*“The species distribution model (SDM) of Homo derives from a recent compilation of archaeological and fossil data⁵⁵. The original data compilation published in 2020 (ref.55) includes 2754 radiometric age estimates for fossil hominin occurrences spanning nearly 2.8 million years, each accompanied by the confidence intervals around the estimate, the fossil site name, the archaeological layer within the site (where available) from which the dated sample derives, the geographical coordinates of the site, and the possible attribution to either one or more than one Homo species. Confident attributions to a single species generated a core record, where instances with multiple attributions formed an extended record. Six different species, *H. habilis*, *H. ergaster*, *H. erectus*, *H. heidelbergensis*, *H. neanderthalensis*, and *H. sapiens* were recognized. The record is updated here (Supplementary Data 1), restricted to the temporal age interval spanning from 2 Ma to 30 ka and excluding Australia and the Americas, including now 3232 data entries. We further decided to collapse *H. habilis* and *H. ergaster* into a single, African Oldowan toolmaker species. Each occurrence is attributed to a given species, depending on the presence of unambiguous anatomical remains, either singly or in connection to a specific lithic tool industry which helps guiding the identification if not otherwise possible from the bones/teeth alone (398 entries, 12%), the age limits of the individual species (e.g. an occurrence in Africa older than the first appearance of *H. sapiens* at Jebel Irhoud⁵⁶ yet younger than the first appearance of *H. heidelbergensis* at Melka Kunture⁵⁷ is attributed to *H. heidelbergensis*), and the stone tool industry (e.g., French Mousterian stone tools are unambiguously assigned to *H. neanderthalensis*, Aurignacian tools were attributed to *H. sapiens*). By applying these criteria, the core record includes 94.5% of the attributions, 48.5% of which refer to *H. neanderthalensis*, and 37.5% to *H. sapiens*. Where neither of these criteria are met, as in the original compilation the SDM acknowledges attribution uncertainty, which we account for in our analysis by testing the stability of our results with respect to different versions of the SDM (Extended Data Fig. 4). For instance, transitional industries (e.g., Levantine Mousterian or Lincombian-Ranisian-Jerzmanowician) receive multiple attribution as they can fit either *H. sapiens* or *H. neanderthalensis* in terms of toolmaker identity^{58,59}. A second source of uncertainty stems from dating. Although some 50% of the entries refer to the 14C method (>90% of which implemented with AMS dating), other dating methods such as ESR (14% of the sample), thermoluminescence (12%), and OSL (12%)*

are less precise than radiocarbon dates. Still, multiple datings are present for individual fossil sites even within a single stratigraphic layer within the site.”

According to their rebuttal, Figure 1 is the main map of all sites based on the database entries. With the authors caveats that Figure 1 does not necessarily portray all sites (one dot may represent more than one site once you zoom in), I am struck by a couple of things. One point is that the large majority of the sites for all hominin species attributions are located in Europe (naturally with the exception of ergaster + habilis). I well recognize that the majority of Palaeolithic research has been, and continues to be, conducted in Europe. This, of course, introduces serious geographic biases, which the authors should probably comment specifically on, and how it may influence the modelled evidence and conclusions. That said, another significant concern is that the data sets outside of Europe do not appear to be representative of published data. In some cases, I believe this is serious and should be tackled. For instance, key data is missing, resulting in substantial geographic gaps in all insets of Figure 1A to D – where there should be available, published data.

R: the referee is certainly correct, there is huge geographic sampling bias with Europe clearly being super-sampled as compared to any other place. Yet, there are at least two good reasons this is not an issue here. First, climatic variation occurs in time and space. Europe is the smallest land area considered here, and the shortest in terms of occupation. Thus, the several more data coming from Europe add what in statistical terms is pseudo-replication, that is more data and no increase in variance (variance is what matters). Secondly, since data in our climate envelope model are collapsed onto geographic cells (1x1 degree resolution), sampling a cell multiple times in the same time interval adds zero variance again. In other words, even though Europe is much more sampled geographically than any other place, the reconstructed Mahalanobis-distance based climate envelope model is only weakly affected. In fact, using only half of the Neanderthal sites in Europe (excluding the other half randomly) generates a very similar climate envelope model to the original one, as illustrated in the figure below for the time-average Neanderthal habitat suitability for the Last Glacial period calculated for 50% of the Neanderthal sites (left, 740 sites) and the full Neanderthal dataset (right, 1480 entries).

This analysis further documents the robustness of our approach with respect to adding or removing individual sites or even large clusters of sites, that are already well samples, such as

for *Homo sapiens* and *Homo neanderthalensis*. The habitat model is largely determined by the overall statistics, rather than individual data points.

As I was concerned about the missing data portrayed in the main text, I did a quality check of the database, examining the representation of sites in several regions (and certainly not all). I wish to make it clear that my survey of the data set is by no means comprehensive, and I am sorry if I make some mistakes below through my quick survey - but I think the authors will get my point, as we proceed in going through some example regions.

Arabia. Only 4 Pleistocene sites are listed in the database. These range between 127 to 38 ka. This is of immediate concern, with significant omissions for the cultural history of Arabia. This pertains to both older dates and younger dates, easily retrievable in the literature. For example, two papers led by Scerri et al. (2018, 2021) are in Scientific Reports, and a recent one is by Groucutt et al. (2021) in Nature; these sites and ages are missing in the database. This is important because these publications report on a series of ages exceeding 127 ka, and down to 400 ka, so a prehistory extending for 275 ka is absent in the paper. These older sites are presumably manufactured by archaic Homo. The missing Acheulean and the Middle Palaeolithic site assemblages are inferred to be manufactured by archaic hominins and *H. sapiens* in the papers. In addition, key publications which report on other key Middle Palaeolithic sites are also missing in the dataset. For instance, three

Middle Palaeolithic sites, each dated to ca. 55 ka, are missing – these are located in Saudi Arabia (Jennings et al., 2016; Groucutt et al. 2021) and Yemen (Delagnes et al., 2012). These 55 ka year old sites are inferred to be manufactured by both *H. sapiens* and the Neanderthals in publications. What is the overall effect? Arabia appears as a blank spot on the maps for key time frames - see Figure 1, which only has archaeological and fossil sites depicted as *Homo sapiens* on the inset. Furthermore, this means that other hominin species were occupying habitats that were in semi-arid and pluvial intervals in Arabia. This was not just confined to *H. sapiens* per se. [NB: please correct the database entry for Jebel Faya. It is in the UAE, not Saudi Arabia]

R: the reply to these concerns is twofold:

1. some papers the reviewer cites went out in 2021, simply put it, they were published after we had prepared the manuscript. Most of this material is now included, such as for instance Saffaqah, Al Marrat 3, Shi'bat Dihya, Mirak (which is given *the Homo neanderthalensis/Homo sapiens* double attribution), Bawa Yawan (published in 2021), Jwalapuram, Dhaba, Tolbor, Nanshanbian, Baigu, Fangniushan and Putaoyuan (published in Yang et al., 2020) and several others. For what concerns the region of interest to this particular point (Arabia), we now include Saffaqah, Al Marrat 3, and Shi'bat Dihya among others. Saffaqah shows a long stratigraphic sequence with multiple anthropological layers. Time range ranges between 188 and 305 ka and it chronologically overlaps with Middle Pleistocene Acheulo-Yabrudian of Levant and the more advanced Middle Palaeolithic cultures. Surprisingly, the lithic assemblage of Saffaqah could be related to an advanced form of Acheulean culture typically associated to earlier

hominins such as *H. heidelbergensis*/*H. erectus*. Saffaqah represents the youngest yet documented Acheulean in southwest Asia (Scerri et al. 2018). Pending further anthropological evidence, we tentatively ascribed Saffaqah to *H. heidelbergensis* considering age and technological affinity as the sole available evidence. Please consider, once more, that we refer to ‘heidelbergensis’ as a pre-sapiens, pre-neanderthal stage of human evolution living during the Middle Pleistocene in the Old World. Whether or not ‘heidelbergensis’ is given taxonomic credit, it is of no impact on the analyses.

For the remaining sites, we attributed to *H. sapiens* all the anthropological evidence associated to MIS5 following the human dispersal scenario described in Groucutt et al. 2018, 2021. Jebel Faya, Al Wusta, Jebel Qattar, and Alathar paleolake deposit fall in this context. However, we cannot use the same consideration for fossil human presence evidence during the MIS3. Levant (and possibly Arabia) could represent the geographic area of Neanderthals-*H.sapiens* admixture. Climate-driven human dispersals, perhaps including Neanderthals from the north, occurred during the partial climatic amelioration of early MIS 3 (around 59–50 ka). During this time frame, there is a large number of anthropological sites and several human remains related to Neanderthals or *H. sapiens* otherwise. Furthermore, no sound technological markers exist for distinguishing the two human species under these peculiar circumstances.

One of these sites, Al Marrat 3 was recently dated between 56.2 and 53.9 ka by using OSL. The lithic assemblage at Al Marrat shows Levallois reduction methods with clear affinities to that of the Levantine Late Middle Palaeolithic, but broadly similar to other technologies found elsewhere, such as in Somalia and certain MIS 5 contexts associated with *H. sapiens* (e.g. unit XV of Qafzeh Cave) (Jennings et al. 2016). We used the double attribution *H.sapiens*/*H.neanderthalensis* because it is currently unclear whether the human groups at Al Marrat 3 were Neanderthals or *H. sapiens*.

Something similar can be argued for Shi’bat Dihya 1 (Wadi Surdud basin). Multiple OSL dates given a weighted mean age at some 55 ka (Delagnes et al. 2012). The core reduction strategies of lithics are typically Middle Paleolithic. However, no clear affinities appear between the Wadi Surdud assemblage and the East African MSA while it was difficult to compare the assemblage to Levantine Mousterian given the variability of the debitage modes proper to these two entities (Delagnes et al. 2012). Consequently, we used the double attribution *H.sapiens*/*H.neanderthalensis* also for this site.

Scerri, E. M., Shipton, C., Clark-Balzan, L., Frouin, M., Schwenninger, J. L., Groucutt, H. S., ... & Petraglia, M. D. (2018). The expansion of later Acheulean hominins into the Arabian Peninsula. *Scientific reports*, 8(1), 1-9.

Groucutt, H. S. et al. Homo sapiens in Arabia by 85,000 years ago. *Nature Eco. Evo.* 2, 800–809 (2018).

Groucutt, H. S., White, T. S., Scerri, E. M., Andrieux, E., Clark-Wilson, R., Breeze, P. S., ... & Petraglia, M. D. (2021). Multiple hominin dispersals into Southwest Asia over the past 400,000 years. *Nature*, 597(7876), 376-380.

Jennings, R. P., Parton, A., Clark-Balzan, L., White, T. S., Groucutt, H. S., Breeze, P. S., ... & Petraglia, M. D. (2016). Human occupation of the northern Arabian interior during early Marine Isotope Stage 3. *Journal of Quaternary Science*, 31(8), 953-966.

Delagnes, A., Tribolo, C., Bertran, P., Brenet, M., Crassard, R., Jaubert, J., ... & Macchiarelli, R. (2012). Inland human settlement in southern Arabia 55,000 years ago. New evidence from the Wadi Surdud Middle Paleolithic site complex, western Yemen. *Journal of Human Evolution*, 63(3), 452-474.

2. The other reason is that we initially stick to exactly what the reviewer suggests, that is that we limit ourselves to what we were confident with. In addition, we now make it explicitly clear why we used multiple attributions for some single sites, and how we deal with taxonomic and dating uncertainties (which were not implemented in the former version of this manuscript).

Missing from the data set is also an important new Neanderthal fossil discovery at 43-41 ka at Bawa Yawan (see Heydari-Guran et al., 2021, PLOS ONE). I would think that these ages would enhance the Neanderthal and *H. sapiens* data and the maps presented in the paper – with a wider range of hominin species in various habitats.

R: Bawa is now part of the data

South Asia. The authors list approximately 18 Palaeolithic sites ranging from ca. 1.8 Ma to ca. 40 ka. While they have captured some of the literature, there are a number of problems with this data set, some of which I highlight. Firstly, I am concerned about the way in which the archaeological sites are typed into species, as no practitioner working in South Asia has even tried to do so.

In the paper, the archaeological sites and ages ranging between 1.8 Ma and 236 ka are typed as *Homo erectus*; those between 385 to 268 ka are *H. heidelbergensis*; those between 150 – 40 ka are *H. sapiens*; and a unusual category of *H. sapiens/H. heidelbergensis* are between 210 and 125 ka. From a fossil perspective, there is only one fossil of an archaic hominin at Hathnora (which the authors categorise as *Homo erectus* and dated to 236 ka – both of which are contentious and debated in the literature as to species and the cited date – other chronometric ages have been published). There are other hominin fossils - *H. sapiens* after 48 ka - but data on Sri Lanka is not presented in this paper (see below).

With respect to the oldest ages (1.8 Ma to 236 ka), there are significant issues. Though 7 sites are presented, 5 of these have been significantly disputed by multiple authors – the ages are not widely accepted at all and only adhered to by a small number of Indian scholars who first reported the ages. For the ages between 385 and 268, I am unclear why these are *H. heidelbergensis* sites (if Hathnora is 236 and said to be *H. erectus*, why are the ages in this category *heidelbergensis*?). For the *H. sapiens* sites, the authors push back our species to 150 ka in India – hugely contentious, of course, in the absence of fossils. For the *H. sapiens/H. heidelbergensis* category – I do not understand why these sites are placed in a hybrid list (/) as opposed to the other sites in the *heidelbergensis* and *sapiens* categories. Some of these sites and ages are certainly valid, so I can not follow the logic of classification. A substantive issue for the Indian data set is that a number of valid ages are absent – for example, other localities and ages at Jwalapuram, the Dhaba site, etc.

R. We wish to point out the paleoanthropological evidence indicates that South Asia is the key area to interpret the human dispersals during the Middle Pleistocene. Unfortunately, the abundance of fossil evidence is not as rich it would be desirable. There is only a semi-complete hominin skull recovered from Narmada Valley in India. Consequently, we are often forced to rely on stone tool technology and occurrence datings to define a possible human attribution. The reviewer must have misunderstood our attribution of the Hathnora skull. We did not attribute it to *H. erectus*, but rather to *H. heidelbergensis*, as described in the Supplementary Information of Raia et al. 2020, following the attribution provided by several colleagues (Rightmire, 1998; Athreya, 2007; Sankhyan, 2010).

As for the Indian Peninsula, just a few anthropological sites are well temporal-constrained. Pappu et al. (2011) documented the evidence of Acheulean stone artefacts in Attirampakkam (India) dated at some 1.5 Ma. In a subsequent paper, Akhilesh and colleagues focused on the upper archeological levels at Attirampakkam. They tentatively reported the first evidence of Middle Paleolithic culture to 385 ± 64 ka and provided a new set of dating (Akhilesh et al., 2018). Dates from layers 5-4 (385 and 268 ka, respectively) are both old and well beyond (i.e. older than) the temporal range of any *H. sapiens* remains elsewhere in Eurasia. Thus, we associated these occurrences to *H. erectus/H. heidelbergensis*. In contrast, Layer 2 is dated at some 170 ka and authors attested Acheulean tools were progressively replaced from Levallois blade technology with a definitively absence of biface use. Dates from layer 2 are in temporal overlap with *H. sapiens* maxilla recovered from Misliya cave (Hershkovitz et al. 2018). Therefore, we cannot exclude a possible attribution to *H. sapiens* and yet, we took into account the dates from Hathnora and the more parsimonious southern route dispersal of *H. sapiens* towards India during MIS 5. Consequently, we associated the dates from layers 3-2 to *H. heidelbergensis/H. sapiens*, whereas layer 1 dated to some 74 ka is more secure evidence for the presence of our own species at site. Indeed, many works linked the advent of *H. sapiens* in India with Toba eruption at 74 ka or earlier (Clarkson et al. 2020; Blinkhorn et al. 2019).

We hope our choices are clearer now, and wish to remark the same reasoning and presentation of the rationales for our choices is present in the One Earth paper supplementary material we have provided.

Rightmire, G. P. (1998). Human evolution in the Middle Pleistocene: the role of *Homo heidelbergensis*. *Evolutionary Anthropology: Issues, News, and Reviews: Issues, News, and Reviews*, 6(6), 218-227.

Athreya, S. (2007). Was *Homo heidelbergensis* in South Asia? A test using the Narmada fossil from central India. In *The evolution and history of human populations in South Asia* (pp. 137-170). Springer, Dordrecht

Sankhyan, A. R. (2010). Pleistocene hominins and associated findings from central Narmada valley bearing on evolution of man in south Asia.

Akhilesh, K., Pappu, S., Rajapara, H. M., Gunnell, Y., Shukla, A. D., & Singhvi, A. K. (2018). Early Middle Palaeolithic culture in India around 385–172 ka reframes Out of Africa models. *Nature*, 554(7690), 97-101.

Hershkovitz, I., Weber, G. W., Quam, R., Duval, M., Grün, R., Kinsley, L., ... & Weinstein-Evron, M. (2018). The earliest modern humans outside Africa. *Science*, 359(6374), 456-459.

Blinkhorn, J., Ajithprasad, P., Mukherjee, A., Kumar, P., Durcan, J. A., & Roberts, P. (2019). The first directly dated evidence for Palaeolithic occupation on the Indian coast at Sandhav, Kachchh. *Quaternary Science Reviews*, 224, 105975.

Clarkson, C., Harris, C., Li, B., Neudorf, C. M., Roberts, R. G., Lane, C., ... & Petraglia, M. (2020). Human occupation of northern India spans the Toba super-eruption~ 74,000 years ago. *Nature communications*, 11(1), 1-10.

For Sri Lanka, no archaeological or fossil sites are presented in the data set. This is so despite its widely published ages between 48 ka and 30 ka. Consequently, there is no data and these sites do not appear on Figure 1. The importance of the Sri Lankan sites is that they are situated in rainforests – some of the very best evidence for the adaptation of hominins in a rainforest ecosystem - this is relevant to the ecological and environmental arguments presented in the paper.

R: the omission of Sri Lanka was on purpose. In this study we focused on climatic variability within occupied land areas, where the species could ‘make for’ the suitable places to live. The colonization of Sri Lanka by *Homo sapiens* around 50 ka is a well-recognized archaeological fact. Yet, survival on the rainforest there does not imply *H. sapiens* made any special preference for the habitat conditions found there, neither it means the rainforest was there at human arrival to begin with, and the counterevidence is impossible to grasp because there is no way to get easily out of an island once trapped there by rising sea levels. As a general approach, we avoided the (very few) insular records for exactly this reason. The situation of *H. erectus* on Java is entirely different since Java is a landbridge island often connected to the land during glacial periods and could easily be abandoned when the eustatic conditions allow for (Flores man and nearly one million year persistence of *H. erectus* on Java standing as proofs). Sri Lanka was loosely connected to India until 1480 when Adam’s bridge was wiped out by a cyclone. Whether or not the ‘bridge’ was there during the late Pleistocene, though, is an entirely different issue, and there is clearly no consensus on this. What is less contentious is that humans made it there during a semi-arid phase, which is inconsistent with the presence of rainforests (Ratnayake, 2016). Conversely, we have in the database fossil *H. sapiens* occurrence in forested areas at the time of their existence at Tham Lod rockshelter and Tam Pà Ling in Thailand (Chitkament et al. 2016, Milano et al. 2018).

Chitkament, T., Gaillard, C., & Shoocongdej, R. (2016). Tham Lod rockshelter (Pang Mapha district, north-western Thailand): Evolution of the lithic assemblages during the late Pleistocene. *Quaternary International*, 416, 151-161.

Milano, S., Demeter, F., Hublin, J. J., Durringer, P., Patole-Edoumba, E., Ponche, J. L., ... & Bacon, A. M. (2018). Environmental conditions framing the first evidence of modern humans at Tam Pà Ling, Laos: A stable isotope record from terrestrial gastropod carbonates. *Palaeogeography, Palaeoclimatology, Palaeoecology*, 511, 352-363.

Ratnayake, A.S. (2016) Links between paleoclimate and prehistorical human dispersal in Sri Lanka: a critical view. *Interdisciplinary Environmental Review*, Vol. 17, 3/4, 249–260.

May I suggest that the authors consult and review synthesis papers that present dated archaeological sites of South Asia – for example, Blinkhorn and Petraglia (2017), *Curr. Anthropol.* In publications such as these, the authors will also see how the data are treated – e.g.,

sites earlier than 100 ka are typically associated with Acheulean toolkits, and thus thought to be the product of archaic hominins.

RE: we did, thank you

Mongolia. Only 1 site is presented, Tsagaan Agui, ranging to between 56-30 ka, and classed as Neanderthal/sapiens. The authors do not seem to be aware of the growing number of well-dated sites in Mongolia (Tolbor, for example) – again data are tabled with respect to sites types and dates. There is also the key fossil *H. sapiens* at Salkhit (34 ka). It seems to me that places like Mongolia are very important to show expansion events, and climate/ecosystems, that *H. sapiens* was inhabiting.

R: In the new version of human database, we added the anthropological evidence and datings from Tolbor 15 (Derevianko et al 2013), and Tolbor 21 (Rybin et al. 2020). Furthermore, we changed the attribution of stone tools recovered from Tsagaan Agui layer 3 from *H. neanderthalensis/H. sapiens* to *H. sapiens*. We believe this replacement is necessary according to new anthropological evidence which predates the presence of *H. sapiens* in Eastern Europe and Central Asia at some 45 ka (Zwyns et al. 2019; Hublin et al. 2020).

As regards the human skull cap recovered at Salkhit, the sample was recently re-dated at some 34 ka by applying a new AMS technique (Devièse et al. 2019) and using OxCal 4.3 program and the INTCAL13 calibration curve. The previous date (ca. 22Ka) was much younger and therefore not present in our data according to the selection criteria applied, yet it has now been added.

Derevianko, A. P., Rybin, E. P., Gladyshev, S. A., Gunchinsuren, B., Tsybankov, A. A., & Olsen, J. W. (2013). Early upper paleolithic stone tool technologies of northern Mongolia: the case of Tolbor-4 and Tolbor-15. *Archaeology, Ethnology and Anthropology of Eurasia*, 41(4), 21-37.

Rybin, E. P., Paine, C. H., Khatsenovich, A. M., Tsedendorj, B., Talamo, S., Marchenko, D. V., ... & Gunchinsuren, B. (2020). A new Upper Paleolithic occupation at the site of Tolbor-21 (Mongolia): Site formation, human behavior and implications for the regional sequence. *Quaternary International*.

Zwyns, N., Paine, C. H., Tsedendorj, B., Talamo, S., Fitzsimmons, K. E., Gantumur, A., ... & Hublin, J. J. (2019). The northern Route for Human dispersal in central and northeast Asia: new evidence from the site of Tolbor-16, Mongolia. *Scientific reports*, 9(1), 1-10.

Hublin, J. J., Sirakov, N., Aldeias, V., Bailey, S., Bard, E., Delvigne, V., ... & Tsanova, T. (2020). Initial Upper Palaeolithic *Homo sapiens* from Bacho Kiro Cave, Bulgaria. *Nature*, 581(7808), 299-302.

Devièse, T., Massilani, D., Yi, S., Comeskey, D., Nagel, S., Nickel, B., ... & Higham, T. (2019). Compound-specific radiocarbon dating and mitochondrial DNA analysis of the Pleistocene hominin from Salkhit Mongolia. *Nature communications*, 10(1), 1-7.

China. A larger set of fossil and archaeological sites are available for China, and the authors present a pretty good list. However, I did a brief check of the data base, and I do see some

missing sites. For instance, why is the 2.1 Ma year old site complex at Shangchen (Zhu et al. 2018, Nature), not in the database? I recognize that this is perhaps due to the fact that the Timmermann et al. paper concentrates on 2 Ma and younger, but younger sites are present at Shangchen, and these do not appear to be tabulated. With respect to Lower to late Middle Pleistocene sites, some key sites and dates are not present – please refer to the comprehensive tabled data in Yang et al., 2020, in QSR – a cross-check between the data set and published synthesis papers like these would give us more confidence.

As I expressed for the South Asian data set, I have the same reservations with assigning ages and sites to species - I would think that this is likely to be greeted with much skepticism. I also do not understand the hybrid categories (/) and how they are used in this paper for the analysis, i.e., *H. heidelbergensis/H. erectus*; *H. neanderthalensis/H. heidelbergensis*. Is the data in these hybrid cases treated in the same way as the data in single-species categories or is this data excluded? I am drawn to the *H. sapiens* category - the authors must recognize that placing some of the early sites into *H. sapiens* will be contentious. Moreover, plugging sites such as Zhiren cave into *H. sapiens* will be treated with some skepticism as the mix of morphological attributes has long been noted, defying easy classification.

R: Coincidentally the Yang et. 2020 and Raia et al. came online in the same month. Unsurprisingly, we could not consider human occurrences reported in Yang's paper as part of our previous OneEarth publication and the original data compilation (Raia et al. 2020). For our revised manuscript, we conducted a thorough review of Chinese hominin occurrences that were published during the last two years and these data are now included in our analysis and the supplemental data.

Specifically, we included the anthropological occurrences documented at Feiliang, Madigou, Huojiadi, Shangbaichun, Longgansi 3, and Baigu. All this evidence can be confidently ascribed to *H. erectus*. Indeed, they documented the use of an archaic core-flake technology. Moreover, their estimated chronological range predates the Melka Kunture site (e. g. the oldest *H. heidelbergensis* occurrence ever published), and the Mid-Pleistocene climate transition (MPT). It is a crucial point for our taxonomical attribution criteria. Many authors suggested MPT driven human dispersal in China and allowed multiple waves of migration by new hominin populations (Yang et al. 2020). Guo et al. (2019) argued the transition between *H. erectus* and more modern forms occurs during the 600-400 ka interval. To corroborate this, technological innovations, including the development of Large Cutting Tools typical and rare evidence of Acheulean tools, occur from the MPT and afterwards.

Consequently, all the sites dated post-MPT (e.g. Hougou, Dongpo, Motialing, Longyadong and so on), cannot securely associated with *H. erectus* only. We used the double attribution *H. erectus/H. heidelbergensis* for them. We decided to use a unique attribution only when explicitly suggested by other authors. For instance, human skulls of Dali and Jinniushan are considered *H. heidelbergensis* as indicated by Rightmire (1998, 2004), Manzi (2016), Zhoukoudian Locality 4 (New Cave) and 15 were associated to *H. heidelbergensis* from the authors who described the material (Shen et al. 2016).

Eventually, we did not consider the possibility of *H. sapiens* presence in China before 120 ka. Sun et al. 2021 combined ancient DNA analysis with a multimethod geological dating strategy to analyze most of samples which attested the presence of *H. sapiens* in China starting from 170 ka. Surprisingly, they demonstrated human samples from Fuyan and Yangjiapo caves dated at late Holocene (Sun et al. 2021) contradicting the first dates based on flowstones calcites. Although some authors criticized the approach of Sun and colleagues, these new findings seem to be very consistent with the human dispersal scenarios illustrated for southern Asia (Hublin 2021).

In the light of the new evidence, the Lingjing (Xuchang) cranium, Zhiren mandible, and Homo remains from Xujiayao represent some exceptions. Although these human samples are dated at some 120 ka or just before, they show too derived morphologies to be considered *H. erectus* or *H. heidelbergensis* (Li et al. 2017; Ao et al. 2017).

Guo, Y., Sun, C., Luo, L., Yang, L., Han, F., Tu, H., ... & Granger, D. (2019). 26 Al/10 Be Burial Dating of the Middle Pleistocene Yiyuan Hominin Fossil Site, Shandong Province, Northern China. *Scientific reports*, 9(1), 6961.

Rightmire, G. P. (2004). Brain size and encephalization in early to mid-Pleistocene Homo. *American Journal of Physical Anthropology: The Official Publication of the American Association of Physical Anthropologists*, 124(2), 109-123.

Manzi, G. (2016). Humans of the Middle Pleistocene: The controversial calvarium from Ceprano (Italy) and its significance for the origin and variability of Homo heidelbergensis. *Quaternary international*, 411, 254-261.

Shen, C., Zhang, X., & Gao, X. (2016). Zhoukoudian in transition: research history, lithic technologies, and transformation of Chinese Palaeolithic archaeology. *Quaternary International*, 400, 4-13.

Yang, S. X., Yue, J. P., Zhou, X., Storozum, M., Huan, F. X., Deng, C. L., & Petraglia, M. D. (2020). Hominin site distributions and behaviours across the Mid-Pleistocene climate transition in China. *Quaternary Science Reviews*, 248, 106614.

Sun, X. F., Wen, S. Q., Lu, C. Q., Zhou, B. Y., Curnoe, D., Lu, H. Y., ... & Li, H. (2021). Ancient DNA and multimethod dating confirm the late arrival of anatomically modern humans in southern China. *Proceedings of the National Academy of Sciences*, 118(8).

Hublin, J. J. (2021). How old are the oldest Homo sapiens in Far East Asia?. *Proceedings of the National Academy of Sciences*, 118(10).

Li, Z. Y., Wu, X. J., Zhou, L. P., Liu, W., Gao, X., Nian, X. M., & Trinkaus, E. (2017). Late Pleistocene archaic human crania from Xuchang, China. *Science*, 355(6328), 969-972.

Ao, H., Liu, C. R., Roberts, A. P., Zhang, P., & Xu, X. (2017). An updated age for the Xujiayao hominin from the Nihewan Basin, North China: Implications for Middle Pleistocene human evolution in East Asia. *Journal of human evolution*, 106, 54-65.

We further explain below why we believe referring to any particular species makes no impact on the analyses, and added to the manuscript the sentence:

*“We acknowledge that i) our species subdivisions may be controversial, ii) our species subdivisions do not necessarily require constancy of morphology, habitat and behaviour, least so genetic isolation. However, we remark that, whereas some species attributions could well be questioned (e.g. *Homo heidelbergensis*), we remain confident that the majority of the record (i.e. 86% of the core data belong to the well-defined, widely accepted *H. neanderthalensis* or *H. sapiens* record and tool-making traditions) presents little challenge, and that whether or not some species might be considered valid, there are widely accepted constraints. Clearly, 500 ka old remains in Africa can neither belong to *H. sapiens* nor *H. habilis*⁶⁰, to the best of the current knowledge, irrespective of whether the name *H. heidelbergensis* is considered appropriate. Still, we did our best to evaluate the sources of uncertainty and pitted our main findings against their potential impact, testing four alternative scenarios (Extended Data Fig. 4) as per species attribution and datings”*

Other regions. With respect to other regions, I am wondering why data from some parts of SE Asia are included in the data set (Java), but not other key regions with fossils and Pleistocene fossils in both mainland and island contexts? Is there a reason for excluding these key data sets, but only including Java? Please explain.

RE: same as above, we updated the compilation accordingly.

I also wonder about the exclusion of Australia, as data is readily available, and presumably good information about climate and environments would emerge by inclusion of this continent.

R: Our paper focuses on identifying the climate driven overlap of different species as an indicator for species successions. Since evidence from Australia and the Americas indicates that only a single species (*Homo sapiens*) has been there we neither included Australia, nor the Americas in our paper and data compilation. We would like to note here, that our data compilation only includes regions, times and hominin groups that are needed to address the main objective of our paper. It was not our main objective of our paper to generate an all-encompass archaeological compilation.

Species Assignments. I will not repeat my reservations, as I did above, about placing archaeological sites into simple species categories. I would like to query the authors here as to why some species are entirely excluded in the data set. There is no category for Denisovans, for instance, and this fossil and archaeological data is presumably ignored. I suppose the authors intend not to include other species – such as *H. floresiensis* and *H. luzonensis* – an explanation on why these species and regions are excluded from analysis would be useful.

R: the reason for excluding some species is that our study focuses on climatic-driven habitat variability. A habitat and climate envelope cannot be adequately defined through one or two locations and less than a handful of timepoints. Therefore, Denisovans and *H. floresiensis* and *H. luzonensis* are not included in our manuscript because there is no sufficient data. To make this clear, we now added to the manuscript:

“...we [excluded] uncertain and poorly-dated specimens (e.g. *H. floresiensis*, *H. naledi*, *H. bodoensis*, *H. longi*, Denisovans) which are either restricted to too few fossil sites for which no climatic variability can possibly be ascertained, or are not currently definite as potentially including any other locality/remain in their definition.”

One comment and suggestion about species attributions. Perhaps it would be best to rethink how species is being categorised in this paper. I think it would be reasonable if the authors, for example, only use species for sites that they are confident about. For archaeological sites without any sign of species, I believe it would be much more logical to devise categories, e.g., Homo sp. A, Homo sp. B., and examine these relative to time slices. Inferences in a discussion still could be made about possible species. I hope that the authors understand my point, as I believe that the categorization of archaeological sites into simple species taxonomies gives a false impression of human evolutionary history.

R: We thank the reviewer for this suggestion. To address the issue of species attribution uncertainties and assess the robustness of our original approach, we have now developed 2 different versions of the dataset – 1) our default dataset, in which we have eliminated all data entries with ambiguous species attributions (tier 1), 2) a second dataset, which replaces all ambiguous entries by a randomly selected species from the possible candidates (tier 2).

Moreover, even though not explicitly requested by any of the reviewers, we have also tested age uncertainties in the fossil/archaeological record by creating a multi-age and single-age version of the dataset (described now in the methods section, Extended Data Fig. 4)

We found that not only the main findings and conclusions stood the proof almost untouched but we would like to emphasize that it could hardly be otherwise, despite the errors and shortcomings that certainly are still part of our effort – the reason being that the habitat suitability model is created from a statistical interpolation of a large number of data-points. Individual mis-identifications of species or age uncertainties of individual records therefore do not affect the overall results.

The methods section now includes the following description

“To account for species attribution uncertainties and age uncertainties, we therefore decided to run our analyses here according to four different approaches:

1. multi date-tier 1 - Only the core record (which excludes entries with ambiguous species attributions) and all the age estimates available for each archaeological layer are used. Multiple age estimates per layer are possible and the age uncertainty for each of them is included in our analysis. This subdivision includes 3060 data entries. The main analysis in our paper is based on this case, but we have to consider possible sampling biases due to the higher weights given to archaeological layers with multiple dates.

2. *multi date-tier 2 - The extended record (ambiguous species attributions are treated by randomly choosing among the possible candidate species) is used along with multiple age estimates (including uncertainties) per layer. This subdivision includes 3245 (all) data entries.*

3. *single date-tier 1 - When there are multiple age estimates for a single archaeological layer, we combine those for this approach to provide a minimum and maximum age for the layer. Each archaeological layer has only one entry, thereby eliminating possible sampling biases in the estimation of our Climate Envelope Model. This subdivision includes 1567 data entries.*

4. *single date-tier 2 – Age estimates for archaeological layers are treated as in approach (3) except the extended, rather than the core record is used. This subdivision includes 1662 data entries.*

*We acknowledge that i) our species subdivisions may be controversial, ii) our species subdivisions do not necessarily require constancy of morphology, habitat and behaviour, least so genetic isolation. However, we remark that, whereas some species attributions could well be questioned (e.g. *Homo heidelbergensis*), we remain confident that the majority of the record (i.e. 86% of the core data belong to the well-defined, widely accepted *H. neanderthalensis* or *H. sapiens* record and tool-making traditions) presents little challenge, and that whether or not some species might be considered valid, there are widely accepted constraints. Clearly, 500 ka old remains in Africa can neither belong to *H. sapiens* nor *H. habilis*, to the best of the current knowledge, irrespective of whether the name *H. heidelbergensis* is considered appropriate. Still, we did our best to evaluate the sources of uncertainty and pitted our main findings against their potential impact, testing four alternative scenarios (Extended Data Fig. 4) as per species attribution and datings, and excluding uncertain and poorly-dated specimen (e.g. *H. floresiensis*, *H. naledi*, *H. bodoensis*, *H. longi*, *H. denisova*) which are either restricted to too few fossil sites for which no climatic variability can possibly be ascertained, or are not currently definite as potentially including any other locality/remain in their definition.”*

Summarizing,

- **we added the suggested data sites from reviewer 1- no major changes in our main conclusions**
- **we generated an updated hominin compilation (~20% larger than in previous submission) and reran the analysis - no major changes in our main conclusions**
- **we will make our updated archaeological and fossil site compilation available to the community as supplementary data**
- **we included uncertainty estimates for age and species attributions - no major changes in our main conclusions (Extended Data Fig. 4)**
- **Focusing on species overlap, Australia and the Americas are not part of this paper (2Ma -30 ka); we therefore did not include them in our compilation and analysis**

Reviewer Reports on the Second Revision:

Referees' comments:

Referee #1 (Remarks to the Author):

This is my third round of review on the paper by Timmermann and colleagues. In the two previous rounds of review, I expressed concern about access to the site/fossil database as well as concerns about the way in which the hominin taxa were being represented in the paper. I have now read the newly revised paper and the rebuttal. Two significant concerns, the weakness of the database, and access to the database, have now been effectively handled by the authors. I am pleased to see that the data set has been expanded, increased by 20% as the investigators note. I have done a spot check of the data base, and I found each site that I was looking for as part of the search. All data for the sites that I found appears to be accurate. It is possible, of course, that some sites are missing in this expanded data set, but as the authors indicate, even with the expanded data set, their main conclusions do not change, so I am satisfied with this explanation. I also expressed concern that the database would not be available to others once this paper is published. The authors indicate that the database will be published as a supplementary file, so this allays my concerns on this point. Others can now verify their findings and they can expand on this database for other analyses. My other main concern in the previous review was that the way in which the authors were handling hominin taxonomy, and the way in which archaeological sites were identified relative to species. The authors improve their descriptions on this topic in the revised paper and they add important explanations about species attributions and classifications and they add caveats to the paper. I find this approach agreeable and now their classifications and arguments are easier to follow. The authors also provide good explanations for the other concerns that I raised. In their rebuttal, their explanations are satisfactory (e.g., why certain regions were not included in this paper). All in all, the paper has been much improved over the three rounds, and I believe that the paper makes a valuable scientific contribution to human evolutionary studies.

A couple of minor errors:

Check that all genera and species are italicised. For example, I saw an instance of *Homo* without italics on Line 253. Check throughout.

The paper mostly has British spelling. I saw an instance of American style on Line 227, paleo instead of palaeo. Check spelling conventions throughout.

Figure 2 caption reads Levante in two spots. I assume this should be Levant.

Author Rebuttals to Second Revision:

RESPONSE TO REMAINING REFEREE'S COMMENTS

=====

Referees' comments:

Referee #1 (Remarks to the Author):

This is my third round of review on the paper by Timmermann and colleagues. In the two previous rounds of review, I expressed concern about access to the site/fossil database as well as concerns about the way in which the hominin taxa were being represented in the paper. I have now read the newly revised paper and the rebuttal. Two significant concerns, the weakness of the database, and access to the database, have now been effectively handled by the authors. I am pleased to see that the data set has been expanded, increased by 20% as the investigators note. I have done a spot check of the data base, and I found each site that I was looking for as part of the search. All data for the sites that I found appears to be accurate. It is possible, of course, that some sites are missing in this expanded data set, but as the authors indicate, even with the expanded data set, their main conclusions do not change, so I am satisfied with this explanation. I also expressed concern that the database would not be available to others once this paper is published. The authors indicate that the database will be published as a supplementary file, so this allays my concerns on this point. Others can now verify their findings and they can expand on this database for other analyses. My other main concern in the previous review was that the way in which the authors were handling hominin taxonomy, and the way in which archaeological sites were identified relative to species. The authors improve their descriptions on this topic in the revised paper and they add important explanations about species attributions and classifications and they add caveats to the paper. I find this approach agreeable and now their classifications and arguments are easier to follow. The authors also provide good explanations for the other concerns that I raised. In their rebuttal, their explanations are satisfactory (e.g., why certain regions were not included in this paper). All in all, the paper has been much improved over the three rounds, and I believe that the paper makes a valuable scientific contribution to human evolutionary studies.

We thank the reviewer for the thorough review.

A couple of minor errors:

Check that all genera and species are italicised. For example, I saw an instance of Homo without italics on Line 253. Check throughout.

Changed accordingly.

The paper mostly has British spelling. I saw an instance of American style on Line 227, paleo instead of palaeo. Check spelling conventions throughout.

Changed accordingly

Figure 2 caption reads Levante in two spots. I assume this should be Levant.

Changed accordingly

Referee #3

'The editor has asked me to especially focus on whether the comments of Reviewer #2 on the original draft have been addressed. That review raised several concerns about (1) model resolution, (2) the quality of the discussion around NPP, (3) the discussion of *H. Heidelbergensis*, (4) the impact of temporal vs spatial climate limitations and (5) over-extension beyond the model results. I feel that the authors rebuttal adequately addressed the question of insufficient resolution (Q1). This is at the frontier of what current resources permit and the inclusion of Extended Data Fig. 6 allows readers to judge for themselves whether this is valid.'

We thank referee 3 for assessing our response to referee 2's comments. No further action has been taken.